# PROBABILISTIC LANGUAGE-IMAGE PRE-TRAINING

**Sanghyuk Chun**      **Wonjae Kim**      **Song Park**      **Sangdoo Yun**
NAVER AI Lab

## ABSTRACT

Vision-language models (VLMs) embed aligned image-text pairs into a joint space but often rely on deterministic embeddings, assuming a one-to-one correspondence between images and texts. This oversimplifies real-world relationships, which are inherently many-to-many, with multiple captions describing a single image and vice versa. We introduce Probabilistic Language-Image Pre-training (Pro-LIP), the first probabilistic VLM pre-trained on a billion-scale image-text dataset using only probabilistic objectives, achieving a strong zero-shot capability (*e.g.*, 74.6% ImageNet zero-shot accuracy with ViT-B/16). ProLIP efficiently estimates uncertainty by an "uncertainty token" without extra parameters. We also introduce a novel inclusion loss that enforces distributional inclusion relationships between image-text pairs and between original and masked inputs. Experiments demonstrate that, by leveraging uncertainty estimates, ProLIP benefits downstream tasks and aligns with intuitive notions of uncertainty, *e.g.*, shorter texts being more uncertain and more general inputs including specific ones. Utilizing text uncertainties, we further improve ImageNet accuracy from 74.6% to 75.8% (under a few-shot setting), supporting the practical advantages of our probabilistic approach. The code is available at https://github.com/naver-ai/prolip.

## 1 INTRODUCTION

Vision-language models (VLMs) aim for a joint vision-language embedding space, and have become a cornerstone in the recent advance of machine learning (Radford et al., 2021; Jia et al., 2021; Li et al., 2022; Zhai et al., 2023). For training, VLMs map an aligned image-text pair (*e.g.*, an image and its corresponding caption) into the same space using contrastive learning. Their rich joint representations learned from large-scale image-text aligned datasets have achieved significant success in various downstream tasks, such as zero-shot classification (by treating class labels as a templated text, *e.g.*, a photo of {·}) or image-text cross-modal retrieval.

Despite their great success, most VLMs encode representations into a deterministic Euclidean space. This assumes a one-to-one correspondence between images and texts, which oversimplifies the complex nature of real-world relationships. In practice, image-text matching is inherently many-to-many. Multiple captions can accurately describe an image, each highlighting different aspects of the visual content. For example, a train image can be described by multiple captions, *e.g.*, "a train", "train station" or "train parked next to a station". Conversely, a caption may correspond to several images describing similar scenes or objects, *e.g.*, "a train" can be matched to all the train images. However, as shown in Figure 1 (b), a deterministic model (*e.g.*, CLIP (Radford et al., 2021)) fails to capture the multiplicity, *e.g.*, "Train Station" embedding is located to an irrelevant point to the other train images and captions. This is because the CLIP loss forces the positive pairs close and random negative pairs far away, which has no stable solution when we map them onto a point in Euclidean space.

Instead of representing an input to a deterministic point vector, we aim to map an input to a random variable. As shown in Figure 1 (a), our probabilistic VLM (PrVLM) approach can handle the multiplicity, *e.g.*, the distribution of "Train Station" covers all the train image distributions. This paper introduces Probabilistic Language-Image Pre-training (ProLIP), the first PrVLM pre-trained on billion-scale image-text pairs only using probabilistic objectives. Compared to previous PrVLM works (Chun et al., 2021; Ji et al., 2023; Upadhyay et al., 2023; Chun, 2024), ProLIP has several advantages. First, while the previous methods need a dedicated module to predict the uncertainty, ProLIP estimates uncertainty very efficiently simply by adding an "uncertainty token" ([UNC]) to input without other additional parameters. Second, we introduce a novel inclusion loss, which en-

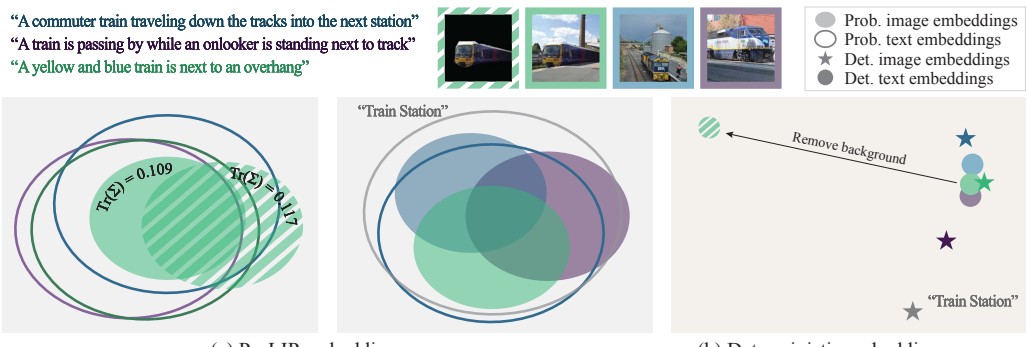

Figure 1: **Comparison of ProLIP and deterministic embedding spaces.** We visualize images and captions from MS-COCO Caption (Chen et al., 2015) using models trained on DataComp 1B (Gadre et al., 2024) with 1.28B seen samples (See Appendix A.2 for more details of visualization method). ProLIP can capture multiplicity of image-text matching (*e.g.*, the text embedding of "Train station" covers all three train images), while deterministic embeddings fail to capture the ambiguity. Furthermore, when we synthetically remove the background, ProLIP maps the new embedding near the original embedding but with a larger uncertainty value ($0.109 \rightarrow 0.117$, while the deterministic model maps the new embedding very far from the original embedding.

forces the distributional inclusion relationship between an image-text pair and between the original input data and the masked one. Our new objective helps embeddings be more interpretable by humans. Third, ProLIP can be trained from scratch without needing any pre-trained models and achieve state-of-the-art zero-shot capability without fine-tuning. Furthermore, ProLIP achieves strong zero-shot capability, *e.g.*, 74.6% ImageNet zero-shot accuracy with the ViT-B/16 backbone, where the CLIP model with the same number of seen samples achieves 73.5% (Ilharco et al., 2021).

In the experiments, ProLIP slightly outperforms the deterministic CLIP model in zero-shot classification (ZSC) tasks (*e.g.*, CLIP shows 67.2 ImageNet ZSC accuracy, while ProLIP shows 67.6). We also show the benefits of using uncertainty estimates for image-text tasks. First, we observe that our intuition and the learned uncertainty are aligned well. For example, (1) texts generally "include" images (*i.e.*, texts are more uncertain than images), (2) shorter texts tend to be more uncertain, (3) more general texts/images tend to be more uncertain and include more specific ones (*e.g.*, masked image and "Train Station" in Figure 1). Furthermore, we show two applications when a proper uncertainty estimate is helpful; Bayesian Prompt Re-Weighting (BPRW), a fully Bayesian approach to seek better ImageNet zero-shot prompts which improves accuracy from $74.6\% \rightarrow 75.8\%$), and the uncertainty-based dataset traversal, which provides a better understanding of dataset hierarchy.

## 2 PRELIMINARY

### 2.1 INHERENT AMBIGUITY INDUCED BY THE MULTIPLICITY OF IMAGE-TEXT PAIRS

The nature of image-text matching is many-to-many. Unfortunately, in practice, this multiplicity is not fully annotated in VL datasets; we only treat one corresponding caption as "positive" caption, while the others are considered as "negative". For example, in COCO Caption (Chen et al., 2015), more than 80% of positive correspondences are labeled as "negative" (Chun et al., 2022). As observed by Chun (2024), this hidden multiplicity inherently causes ambiguity for VL datasets. For example, assume we have the caption "a train is next to a train station" and three semantically similar images showing a train next to a train station. Here, there will be only one positive image for the caption due to the construction protocol of VL datasets. If we approximate three image embeddings as the same image embedding, the correspondence between this approximated embedding, and the caption will be uncertain (*i.e.*, either positive or negative). Suppose we use deterministic matching loss, such as contrastive loss used by CLIP (Radford et al., 2021). In this case, the best deterministic mapping will map the caption embedding not very close to the image embeddings but "properly" far away from them. CLIP does not have enough capability to capture multiplicity and ambiguity. We aim to achieve an embedding space that can represent the inherent uncertainty of the input (also known as "aleatoric uncertainty") for a more interpretable and understandable embedding space. We include more detailed discussion in Appendix A.1.

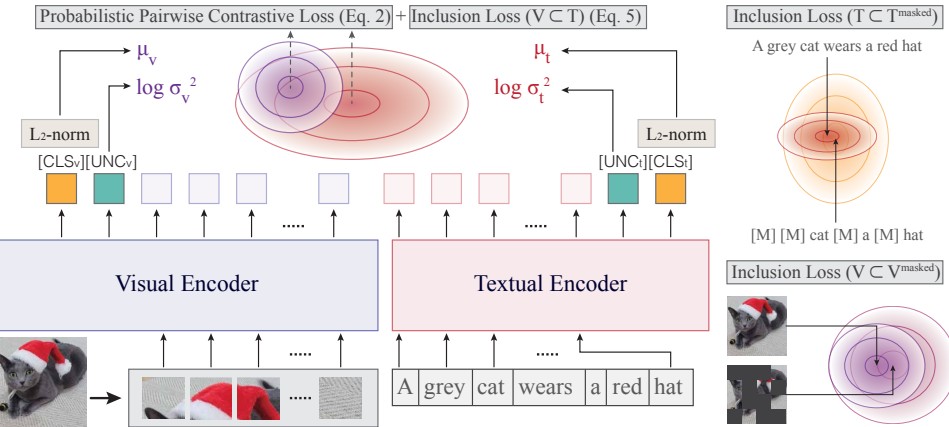

Figure 2: **Overview of ProLIP.** [CLS] and [UNC] tokens are used for $\mu$ and $\log \sigma^2$, respectively.

## 2.2 PROBABILISTIC IMAGE-TEXT REPRESENTATIONS

Probabilistic embeddings map each data point as a random variable (*e.g.*, a Gaussian distribution) rather than a fixed vector, capturing the inherent uncertainty and diversity. This approach offers a better understanding of the semantic space by providing an extra axis of uncertainty, *e.g.*, we can quantify the uncertainty of the input by using the estimated uncertainty (*e.g.*, covariance of Gaussian). Recently, Kirchhof et al. (2023) theoretically have shown that a probabilistic representation learning with a proper probabilistic matching loss can recover the correct aleatoric uncertainty. Namely, a probabilistic mapping can capture the ambiguity of the inputs. Probabilistic embeddings have been actively studied for applications with inherent ambiguity, such as word embeddings (Nguyen et al., 2017), image embeddings (Oh et al., 2019), face understanding (Shi & Jain, 2019; Chang et al., 2020), 2D-to-3D pose estimation (Sun et al., 2020), speaker diarization (Silnova et al., 2020), video understanding (Park et al., 2022), and composed image retrieval (Neculai et al., 2022).

As we discussed in Section 2.1 and A.1, VL tasks also suffer from aleatoric uncertainty caused by the inherent multiplicity of image-text matching and sparse annotations. Recently, there have been attempts to tackle the inherent ambiguity of VL tasks with probabilistic embeddings (Chun et al., 2021; Ji et al., 2023; Upadhyay et al., 2023; Chun, 2024). However, these methods have a very limited scale to be used as a generic purpose VLM, such as CLIP. For example, ProbVLM (Upadhyay et al., 2023) is an ad-hoc module top on the frozen pre-trained CLIP, limiting the full exploration of the probabilistic space. Furthermore, ProbVLM is only trained on small image caption datasets, such as CUB (Wah et al., 2011) or COCO caption (Chen et al., 2015), which makes it not applicable to more practical zero-shot classification applications. MAP (Ji et al., 2023) proposes a pre-training method using a cross-attention Transformer. However, it has a limited zero-shot capability, resulting in the need to fine-tune the model for each downstream task. Furthermore, its structure is highly inefficient for retrieval systems; it needs both image and text pair to compute a similarity between them, *i.e.*, we have to compute all possible image-text pairs to get the full similarity. Lastly, PCME++ (Chun, 2024) showed a possibility of pre-trained PrVLM, but their scalability is still limited (*e.g.*, achieving 34% ImageNet zero-shot accuracy). We empirically observe that the objective function of PCME++ shows slow or unstable training under large-scale image-text pairs. Furthermore, all these PrVLMs need heavy additional parameters to estimate uncertainty from data. ProLIP does not need a dedicated module for uncertainty estimate but employs a very efficient strategy using [UNC].

# 3 PROBABILISTIC LANGUAGE-IMAGE PRE-TRAINING

## 3.1 ARCHITECTURE

We model an input as a Gaussian random variable with a diagonal covariance by estimating mean $\mu$ and variance $\sigma^2$ vectors from the input. Similar to CLIP (Radford et al., 2021), ProLIP has separate visual and textual encoders. We use VisionTransformer (ViT) (Dosovitskiy et al., 2021) for the visual encoder and Transformer (Vaswani et al., 2017) for the textual encoder.

Previous Probabilistic VLMs (PrVLMs) introduce additional parameters for estimating uncertainty. For example, PCME++ (Chun, 2024) uses one multi-head self-attention block for this. However, this approach will require additional parameters and computational costs, limiting usability (See Table C.6). Instead, we introduce a new uncertainty token [UNC], along with the class token [CLS] (See Figure 2). Compared to the previous PrVLMs, [UNC] requires almost negligible additional parameters. The visual encoder takes [CLS] and [UNC] at the beginning of the input sequences, while the textual encoder takes [UNC] and [CLS] at the end of the input. This is because the textual encoder of the original CLIP uses the end-of-sentence token rather than [CLS] at the beginning. Note that we assume diagonal covariance for simplicity, namely, [UNC] is the same dimension with [CLS]. We use the L2-normalized [CLS] output as $\mu$ and [UNC] output as $\log \sigma^2$. Similar to [CLS], [UNC] is projected to the final embedding space using a linear layer. We initialize the bias value of this layer to a small value (*e.g.*, $-10$) to initialize the initial $\sigma^2$ scale small (*e.g.*, $\exp(-10) \approx 5 \times 10^{-5}$ for each dimension). This simple trick helps stable training.

## 3.2 PROBABILISTIC PAIRWISE CONTRASTIVE LOSS

In this subsection, we introduce the probabilistic pairwise contrastive loss (PPCL), the main objective function of ProLIP. PPCL is similar to the probabilistic matching loss (PML) of PCME++, but we modify PML for stable training based on the log sigmoid loss by SigLIP (Zhai et al., 2023).

Following PCME++, we use the closed-form sampled distance (CSD) as our probabilistic distance:

$$d_{\text{CSD}}(Z_1, Z_2) = \mathbb{E}_{Z_1, Z_2} \|Z_1 - Z_2\|_2^2 = \|\mu_1 - \mu_2\|_2^2 + tr(\Sigma_1 + \Sigma_2) = \|\mu_1 - \mu_2\|_2^2 + \|\sigma_1 + \sigma_2\|_1, \quad (1)$$

where $Z_1$ and $Z_2$ are Gaussian random variables with diagonal covariances. The probabilistic matching loss by PCME++ uses pairwise binary cross entropy (BCE) by taking $-a \cdot d_{\text{CSD}}(Z_1, Z_2) + b$ as logits, where $a$ and $b$ are learnable scalars. However, we empirically observe that PML fastly converges to a small value, and its gradient is dramatically small, which makes the overall learning procedure slow or unstable (see Appendix A.3 for details). To solve the problem, we employ log sigmoid loss (Zhai et al., 2023). By replacing the squared L2 distance $\|\mu_1 - \mu_2\|_2^2$ to Equation (1) (details are in Appendix A.4), we have a new probabilistic pairwise contrastive loss (PPCL):

$$\mathcal{L}_{\text{PPCL}}(Z_v, Z_t) = -\log \frac{1}{1 + \exp(y_{vt}(-a(\mu_v^\top \mu_t - \frac{1}{2} tr(\Sigma_v + \Sigma_t)) + b))}, \quad (2)$$

where $a$ and $b$ are learnable scalar values and $y_{vt}$ is 1 if $v$ and $t$ are matched otherwise -1.

## 3.3 INCLUSION LOSS

Although PPCL enables to learn probabilistic representations, we empirically observe that the learned uncertainty from data is often counterintuitive to humans. For example, we may expect that text captions with a general meaning (*e.g.*, "photo") has a very large covariance that can cover all the photographic image embeddings, but sometimes it does not. Similarly, we may expect that if a text or an image loses some information (*e.g.*, some tokens are randomly masked), its probability distribution will entail the distribution of the original sample. However, it is not always guaranteed that a model will learn desired uncertainty, especially under noisy image-text correspondences.

To tackle the issue, we introduce a novel objective function enforcing a random variable $Z_1$ to be included in another random variable $Z_2$. Let $p_1$ and $p_2$ be their corresponding probability density function (pdf). If $Z_1$ is included in $Z_2$, then we can presume that the area with high $p_1$ will be overlapped to the area with high $p_2$. From this observation, we propose a novel inclusion measure by emphasizing the area with high $p_1$ and compute expectation of the emphasized $p_1$ under the distribution $p_2$. Specifically, we take the square to $p_1$, and compute $\int p_1(x)^2 p_2(x)dx$. This measure is related to Bhattacharyya distance ($\int \sqrt{p_1 p_2} dx$) or the inner product ($\int p_1 p_2 dx$), but our measure is designed for measuring "inclusion" (it becomes high if $Z_1$ is included in $Z_2$ and otherwise low) while the others are designed for measuring "distance" or "dissimilarity" between distributions.

The log inclusion measure (omitting constants) can be derived as follows:

$$\text{inc}(Z_1, Z_2) = \log \int_{-\infty}^{\infty} p_1^2(x) p_2(x)dx = -2 \log \sigma_1^2 - \log \sigma_2^2 - \frac{1}{2} \log(A) + \frac{B^2}{4A} - C,$$

$$\text{where } A = \frac{1}{\sigma_1^2} + \frac{1}{2\sigma_2^2}, \quad B = \frac{2\mu_1}{\sigma_1^2} + \frac{\mu_2}{\sigma_2^2}, \quad C = \frac{\mu_1^2}{\sigma_1^2} + \frac{\mu_2^2}{2\sigma_2^2}. \quad (3)$$

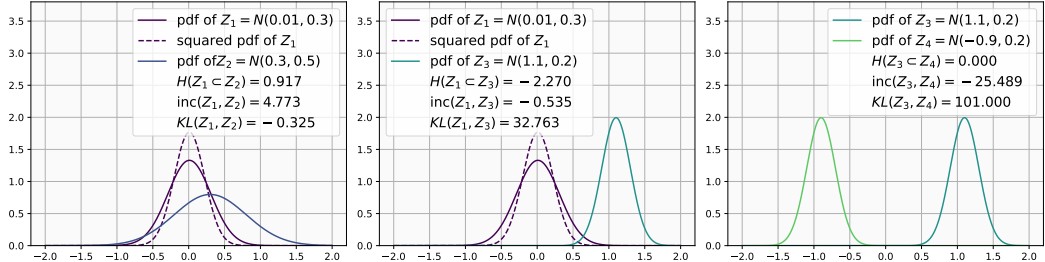

Figure 3: **Visual understanding of the proposed inclusion loss.** We plot probabilistic density functions (pdfs) of three pairs of Gaussian distributions and their inclusion hypothesis, $\mathcal{H}(Z_1 \subset Z_2)$ (Equation (4)), log inclusion (Equation (3)) and KL divergence. The dashed line denotes the squared pdf, *i.e.*, $p^2(x)$. $\mathcal{H}(Z_1 \subset Z_2)$ becomes (a) positive if $Z_1$ is included in $Z_2$ and (b) otherwise negative. (c) If $Z_1$ and $Z_2$ are the same level, then $\mathcal{H}$ will become zero. While log inclusion represents how $Z_1$ is included in $Z_2$, KL measures the "dissimilarity" between distributions (*e.g.*, (c) has the largest KL, but the smallest "inc"). Figure A.2 shows more examples.

The full derivation is in Appendix A.5.

Now, using Equation (3), we introduce a hypothesis test whether $Z_1$ is included in $Z_2$ as follows:

$$\mathcal{H}(Z_1 \subset Z_2) = \log \int_{-\infty}^{\infty} p_1^2(x)p_2(x)dx - \log \int_{-\infty}^{\infty} p_1(x)p_2^2(x)dx. \quad (4)$$

$\mathcal{H}$ is positive if the hypothesis is true and otherwise negative (See Figure 3). It has two distinct properties compared to other probabilistic measures. First, $\mathcal{H}$ is asymmetric. The most probabilistic measures aim to measure the "distance", "overlapping" or "dissimilarity" between two distributions, therefore symmetric (*e.g.*, Wasserstein distance and Bhattacharyya distance). Meanwhile, we measure the level of "inclusion" of two random variables, therefore asymmetric: if $Z_1$ is included in $Z_2$, then $Z_2$ will not be included in $Z_1$, *i.e.*, $\mathcal{H}(Z_1, Z_2) \neq \mathcal{H}(Z_2, Z_1)$. Compared to the asymmetric measure, such as KL divergence, we aim to measure how $Z_1$ is "included" in $Z_2$. In contrast, KL measures the dissimilarity between them based on relative entropy. As shown in Figure 3, even if $Z_1$ and $Z_2$ have the same variance, we have very high KL, while our measure becomes very small.

Similarly to PPCL, we use the log sigmoid loss for stable convergence. We use $\log \int_{-\infty}^{\infty} p_1^2(x)p_2(x)dx - \log \int_{-\infty}^{\infty} p_1(x)p_2^2(x)dx$ as the logit value, where the logit becomes positive if $Z_1$ is included in $Z_2$. Now, we introduce our novel inclusion loss as follows:

$$\mathcal{L}_{\text{inclusion}}(Z_1 \subset Z_2) = -\log \frac{1}{1 + \exp\left(-c\mathcal{H}(Z_1 \subset Z_2)\right)}, \quad (5)$$

where $c$ is a positive scalar. For stability, we fix $c$ as a large value, such as 1000. Like KL divergence, inclusion loss can be volatile if variance values become extremely small. To prevent a loss explosion, during training, we multiply a small $\varepsilon$ to $\frac{1}{\sigma^2}$ for computing $A, B, C$ in Equation (3), mimicking each Gaussian has sufficiently large variances multiplied by $1/\varepsilon$. See Table C.5 for more details.

Using the inclusion loss, we enforce two properties to the model. First, we let text distribution include image distribution, *i.e.*, $\mathcal{L}_{\text{inclusion}}(Z_v \subset Z_t)$. This intuition is from observations from the previous studies showing that "text entails image" (Chun et al., 2021; Desai et al., 2023; Chun, 2024; Kim et al., 2024). Conceptually, when we describe an image, we select the product of the relevant concepts by an arbitrary choice. Therefore, text usually has more general information than images. As another property, we let an embedding of partial information include the embedding of its full information, *i.e.*, $\mathcal{L}_{\text{inclusion}}(Z \subset Z^{\text{partial}})$. For example, we generate a text containing partial information of the original caption by masking out random tokens (Devlin et al., 2018). Similarly, we generate a partial image by masking out the image tokens (He et al., 2022). In practice, we mask out 75% of the input tokens to generate partial information. For text, we replace the input tokens with [MASK], and for image, we drop the input patch tokens for efficient computation (Li et al., 2023).

Finally, we use the VIB loss as a regularization of each Gaussian embedding (*i.e.*, preventing too small $\sigma^2$) following Chun et al. (2021) and Chun (2024) – See Appendix A.7 for the details. Putting

Equation (2) and Equation (5) all together, we have the following learning objective:

$$
\begin{aligned}
\mathcal{L} = &\sum_{Z_v \in \mathcal{V}} \sum_{Z_t \in \mathcal{T}} \mathcal{L}_{\text{PPCL}}(Z_v, Z_t) + \sum_{Z_v \in \mathcal{V}} \left[ \alpha_2 \mathcal{L}_{\text{inclusion}}(Z_v \subset Z_{v^{\text{masked}}}) + \beta \mathcal{L}_{\text{VIB}}(Z_v) \right] \\
&+ \sum_{Z_t \in \mathcal{T}} \left[ \alpha_2 \mathcal{L}_{\text{inclusion}}(Z_t \subset Z_{t^{\text{masked}}}) + \beta \mathcal{L}_{\text{VIB}}(Z_t) \right] + \sum_{(v,t) \in (\mathcal{V}, \mathcal{T})} \alpha_1 \mathcal{L}_{\text{inclusion}}(Z_v \subset Z_t),
\end{aligned}
\tag{6}
$$

where $\alpha_1, \alpha_2, \beta$ are control hyperparameters for each loss function. For computational efficiency, we generate masked samples only for 12.5% samples in the mini-batch and compute inclusion loss using them. VIB loss is computed for all samples. We report the loss ablation study in Appendix C.3.

### 3.4 Prompt Tuning with Uncertainty Estimates

As observed by Chun (2024), the estimated uncertainty is not only beneficial to understanding the input data uncertainty but also effective to zero-shot classification (ZSC). In practice, we use multiple templated text prompts to estimate the textual embedding of class names. For example, the original CLIP paper uses 80 prompts, including "a photo of $\{\cdot\}$". Although this prompt engineering with a mixture of templates significantly improves ZSC performances, it is still unclear which template benefits ZSC. Furthermore, if we carefully explore images of each class, we can conjecture that each class might need different templates. For example, in ImageNet, "ferret" images often co-occur with the ferret's owner. In this case, prompts like "a photo of **my** ferret" can be helpful to estimate a text feature corresponding to the images, compared to using "a origami ferret".

How can we select the most informative text prompts? One possible solution will be filtering out highly uncertain text prompts. For example, for "black-footed ferret" class, "the embroidered $\{\cdot\}$", "the origami $\{\cdot\}$" or "the plastic $\{\cdot\}$" have high uncertainty values, while "a low resolution photo of $\{\cdot\}$" or "a cropped photo of $\{\cdot\}$" have small uncertainty values. Our experiment shows this strategy is moderately effective: it improves ImageNet ZSC accuracy +0.1pp. We presume that it is because the variety of text prompt uncertainties for each class is not significantly large. Also, we presume that the suitability of text prompts is not solely dependent on the text itself; we may need to consider how texts describe the corresponding images well. We propose Bayesian Prompt Re-Weighting (BPRW), a simple probabilistic approach to find the optimal weight of prompts for each class.

Let $\pi_c \in \mathbb{R}^N$ be the weight of each prompt, where $N$ is the number of prompts (*i.e.*, 80 for ImageNet) and $c$ is the class index. Our goal is to find the best $\pi_i$ that the new text embedding $Z_t^{\text{new}} = \sum_i \pi_c^i Z_t^i$ describe the given $M$ image embeddings $Z_v^j$. To achieve this goal, we optimize $\pi_c$ to have the best posterior for $Z_t$ and $Z_v$. First, we sample $K$ point vectors for each $Z_v^j$ and assume they are observations (total $K \times M$ point vectors). Next, we optimize a simple Expectation-Maximization (EM) algorithm to find the best $\pi$ achieving the best log-likelihood. Here, we set a Dirichlet prior for $\pi_c$ using the uncertainty values of each $Z_t$, *i.e.*, a prompt with higher uncertainty has a smaller prior. Due to the page limit, we describe the detailed algorithm in Appendix A.8.

Although our algorithm is theoretically well-founded and flexible by a Bayesian approach (*e.g.*, setting prior assumption using $\sigma_t^2$), it needs the image embeddings corresponding to the target class, which violates the ZSC assumption. We tackle this issue by collecting corresponding image embeddings using KNN for each text class embedding. If we can use some true pairs under a few-shot setting (*e.g.*, 5 true images for each class), we observe a significant performance improvement.

## 4 Experiments

### 4.1 Implementation Details and Experimental Protocol

**Model.** We use ViT-B/16 (Dosovitskiy et al., 2021) as our image encoder and a 12-layer 768-wide Transformer (Vaswani et al., 2017) as our text encoder. We set the embedding dimension to 768 and the context length to 64 tokens, following SigLIP ViT-B/16 (Zhai et al., 2023).

**Optimization.** We implement ProLIP based on `openclip` (Ilharco et al., 2021) and the DataComp-1B dataset (Gadre et al., 2024). We list the optimization hyperparameters in Appendix B.1. We train ProLIP models using 32 NVIDIA H100 GPUs with Bfloat16 precision, taking about one day to train a ViT-B/16 model with 1.28B seen samples. We initialize the bias value of the linear projection top

on [UNC] to $-10$ to initialize $\log \sigma^2$ with a small value. We set the initial scale and bias parameters ($a$ and $b$ in Equation (2)) to 10 and $-10$, following Zhai et al. (2023). We randomly select 12.5% image-text pairs from the mini-batch and masking out their 75% information using [MASK] for texts (Devlin et al., 2018) and token drop for images (He et al., 2022). Fine-tuning details are in Appendix B.2.

**Evaluation.** We evaluate the models on 38 tasks of Datacomp evaluation suite (Gadre et al., 2024) – the full evaluation datasets are listed in Appendix B.3. We report five categories: ImageNet, 6 ImageNet variants with distribution shifts, 13 VTAB tasks, 3 retrieval tasks, and the average of 38 tasks. In addition, we employ the HierarCaps dataset (Alper & Averbuch-Elor, 2024), which provides captions with four different hierarchies (*e.g.*, "water sports" $\Rightarrow$ "kite surfing" $\Rightarrow$ "kite surfer on top of the board" $\Rightarrow$ "kite surfer in the air on top of a red board"). Similarly, we construct new HierarImgs dataset, which provides images with four different hierarchies (See Figure B.2 for examples). We will describe the details of HierarCaps, HierarImgs, and their evaluation in Section 4.4.

## 4.2 MAIN RESULTS

Table 1 shows the main result. We use multiple prompts for each task following the DataComp evaluation suite. Similar to CLIP zero-shot classification (ZSC), ProLIP uses the ensemble of multiple prompts by $Z_t^{\text{mixed}} = \mathcal{N}(\frac{1}{N} \sum_i \mu_i, \frac{1}{N} \sum_i \sigma_i^2)$, where $\mu_i$ and $\sigma_i^2$ denote the mean and variance of $i$-th prompt and $N$ is the number of prompts (*e.g.*, 80 for ImageNet). Note that if we treat this operation as the "average" of $N$ random variables, then the variance should be $\frac{1}{N^2} \sum_i \sigma_i^2$, but we empirically observe that the division decreases the final ZSC performance, *e.g.*, 74.51 where our approach shows 74.58 on ImageNet. We did not use the uncertainty-based ZSC described in Section 3.4 for evaluating 38 tasks. Instead, we use CSD to find the nearest class text embedding.

Table 1 shows that ProLIP outperforms CLIP in all metrics with 1.28B seen samples. Furthermore, when we train ProLIP with 12.8B seen samples, we achieve a high-performing PrVLM. We show more ablation studies in Appendix C.3, including loss design, hyperparameter, and architecture.

Table 1: **Zero-shot classification results.** The full results for each task are in Table C.1. ViT-L/16 and SO400M/14 results are the fine-tuned results from the pre-trained SigLIP models. More results in Table C.2.

|  |  | # Samples Seen | ImageNet | IN dist. shifts | VTAB | Retrieval | Average |
|---|---|---|---|---|---|---|---|
| ViT-B/16 | CLIP | 1.28B | 67.2 | 55.1 | 56.9 | 53.4 | 57.1 |
|  | SigLIP | 1.28B | 67.4 | 55.4 | 55.7 | 53.4 | 56.7 |
|  | ProLIP | 1.28B | 67.8 | 55.3 | 58.5 | 53.0 | 57.9 |
|  | ProLIP | 12.8B | 74.6 | 63.0 | 63.7 | 59.6 | 63.3 |
| ViT-L/16 | ProLIP | 1.28B* | 79.4 | 68.6 | 64.0 | 61.3 | 65.9 |
| ViT-SO400M/14 | ProLIP | 1.28B* | 79.3 | 69.0 | 65.1 | 62.5 | 66.6 |

## 4.3 UNDERSTANDING THE LEARNED UNCERTAINTY

**Uncertain samples visualization.** From Equation (1), we can define the uncertainty of the given input by measuring $tr(\Sigma)$, namely, $\sum_i \sigma_i^2$ (we simply denote $\sigma_v^2$ for image uncertainty and $\sigma_t^2$ similarly). Figure 4 shows the samples with low and high uncertainty values using this value. We extract samples from the 3.5M subset of 12.8B DataComp CommonCrawl small (Gadre et al., 2024) filtered by CLIP similarity and English filtering [1]. We use ProLIP with 12.8B seen samples for the analyses.

Figure 4 shows that the texts with more general meanings have high uncertainty, *e.g.*"Screenshot" or "graphic". This is because a shorter text with more general meaning has more opportunity to be matched to various images. In contrast, certain captions describe a longer and distinct context, such as the exact address or proper nouns, which is unlikely matched to multiple images. We will show that the context length of the text is highly correlated to the uncertainty value (Figure 6). Interestingly, there are captions with high uncertainty despite long context lengths (*e.g.*, more than the specified context length). We empirically observe that such captions have almost no information, showing that ProLIP captures the text uncertainty well. The examples are shown in Appendix C.1.

---

[1] https://huggingface.co/datasets/nielsr/datacomp-small-filtered

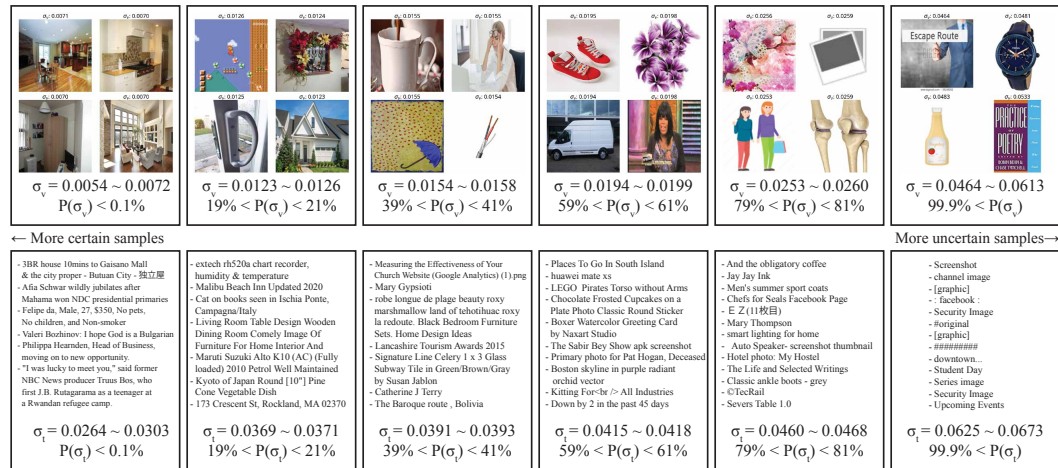

Figure 4: **Uncertain & certain samples.** Visualization from the 3.5M filtered DataComp Small pool.

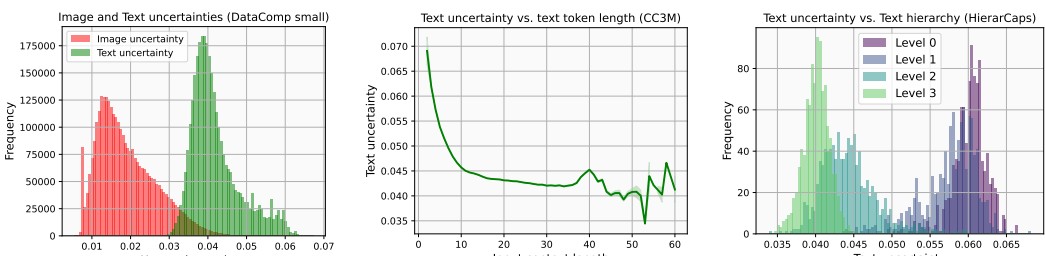

Figure 5: $\sigma_v^2$ vs. $\sigma_t^2$. Generally, texts are more uncertain than images, as shown in Figure 4.

Figure 6: $\sigma_t^2$ **vs. context length.** Shorter texts are generally more uncertain than complex ones.

Figure 7: $\sigma_t^2$ **by Text hierarchy.** More general captions (*i.e.*, lower level) are more uncertain.

We can see certain and uncertain images in the upper row of Figure 4. On the uncertain image side, we can find images solely with an object (*e.g.*, a clock, a book, or a clipart) on a white background. Generally, such images can be matched to multiple possible descriptions, *e.g.*, the name of the product, the detailed explanation of the product, or the written text in the image (*e.g.*, the book title). On the other hand, certain images have more complex visual cues that can be described with more and specific captions. More samples with high and low uncertainty can be found in Figure C.1.

**Statistics of $\sigma_v^2$ and $\sigma_t^2$.** In Figure 4, we also observe that $\sigma_v^2$ is generally smaller than $\sigma_t^2$, would be originated from the inclusion loss $\mathcal{L}_{\text{inclusion}}(Z_v, Z_t)$ in Equation (6). Figure 5 shows that the image embeddings and text embeddings have distinct uncertainty values. In Appendix C.6, we provide a detailed discussion of the relationship between learned uncertainty and human preference.

**What is the source of the uncertainty?** We answer this question by analyzing text context length and data hierarchy. First, we plot the text uncertainty by the context length on the ConceptualCaption 3M (CC3M) captions (Sharma et al., 2018). As shown in Figure 6, a short caption tends to have a large uncertainty value. For example, we observe that the caption *"film series"* has the largest uncertainty value in CC3M, while the caption *"gangsta rap artist told by person @ person I almost died you have to see this!"* is the most certain caption. Namely, more uncertain captions tend to be more logically "general" captions, such as *"dress - sewing pattern"* or *"person – before & after"*, while more certain captions specify a particular situation. From this observation, we explore the relationship between the uncertainty and varying levels of description. We employ the HierarCaps dataset, whose images have four levels of descriptions from the full caption of COCO Caption and its logical entailment hierarchy with three different levels (*e.g.*, "bird" $\Rightarrow$ "blue bird" ...). Examples are shown in Figure 9. Figure 7 shows the relationship between the text uncertainty and text hierarchy levels. Here, Level 0 denotes the most general captions, *e.g.*, "chair" or "bird", and Level 4 represents the original COCO Caption. As shown in the figure, more general captions (lower levels) tend to be more uncertain, while more specific captions (higher levels) tend to be less uncertain.

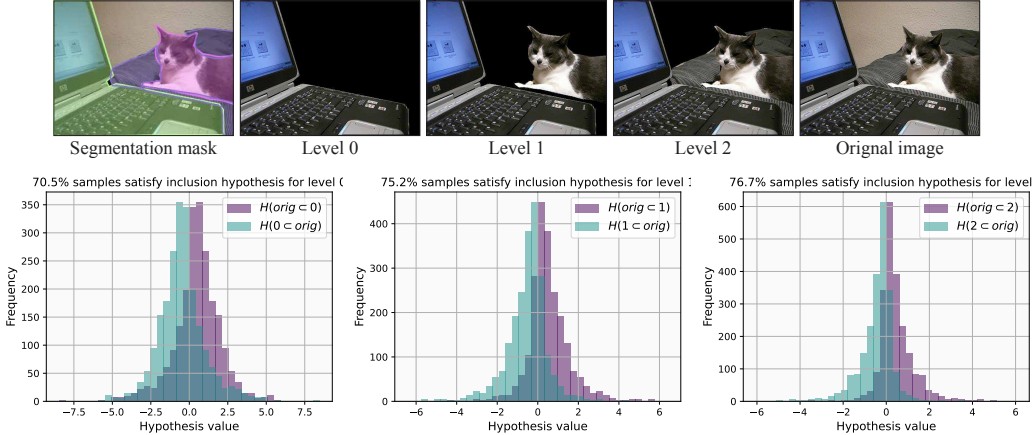

Figure 8: $\sigma_v^2$ **by image hierarchy using HierarImgs.** We tested the inclusion hypothesis of the original image and masked images from level $i$, $\mathcal{H}(\text{orig} \subset i)$, and its inverse hypothesis, $\mathcal{H}(i \subset \text{orig})$. In all tests, more than 70% images are included in their lower-level images (purple histogram bars with positive hypothesis values). The dataset construction of HierarImgs and related discussions can be found in Appendix B.5 and C.5.

We further investigate whether a similar phenomenon happens for the visual modality by constructing a new HierarImgs dataset to represent the logical visual hierarchy. As shown in the upper row of Figure 8, our HierarImgs dataset consists of four levels: Level 4 is the original image and Level 0 is the largest visual segment. The details of the dataset construction can be found in Appendix B.5.

Using the images, we analyze the relationship between the visual uncertainty and varying levels of visual information. We test whether each image becomes more uncertain than the original image by applying the inclusion hypothesis. Namely, we test if a lower-level image includes its original image by Equation (4). As shown in Figure 8, most of the images satisfy the inclusion hypothesis (*e.g.*, more than 70%), implying that ProLIP also captures image hierarchy. In Appendix C.5, we explain more details of the image hierarchy experiments, including the absolute $\sigma_v^2$ value by different levels and possible pitfalls of HierarImgs (*e.g.*, we need more careful filtering for a reliable evaluation).

## 4.4 APPLICATION USING UNCERTAINTY

**Image traversals.** Following Alper & Averbuch-Elor (2024), we first set the [ROOT] embedding. Then, we retrieve the nearest caption of the given image, and interpolate [ROOT] and the text embedding with 50 equally spaced steps. The null text embedding `""` is used as the [ROOT] of the CLIP embedding space. In ProLIP case, we can utilize the additional uncertainty information and the inclusion hypothesis. Hence, we search the root embedding of the given embedding by searching the text embedding that includes the given image embedding most. The other procedure equals to Alper & Averbuch-Elor (2024). We perform traversals on HierarCaps. Details are in Appendix B.4.

We show image traversal results in Figure 9. Interestingly, the most inclusive caption (*i.e.*, [ROOT]) for each image is not always same to the ground truth level 0 caption of HierarCaps. For example, our estimated [ROOT] for the vase picture is "vase", while the GT level 0 caption is "object". From the observation, we can presume that although our retrieval results are plausible, it could lead to inferior HierarCaps retrieval results. To ensure more diversity, we take an average of `""` and the most inclusive text embedding and use it as the root embedding. Using the newly proposed root embedding, we quantitatively measure the performance of our traversal in Table 2. First, our new [ROOT] embedding is more specialized to the inputs, rather than only using `""`; we can achieve higher $R@1^{[ROOT]}$ using our approach. The table shows that the proposed probabilistic image traversal achieves higher recall and precision than deterministic traversal using `""` as [ROOT]. Namely, the probabilistic approach gives more opportunities to get more precise captions during the traversal.

**Uncertainty-based ImageNet prompt enhancement.** As described in Section 3.4, we propose BPRW, a new prompt re-weighting method to find a weight $\pi_c$ for each class. A text embedding weighted by $\pi_c$, *i.e.*, $Z_t^{\text{new}} = \sum_i \pi_c^i Z_t^i$ will be used as a new class embedding for ZSC. Table 3 shows the ImageNet classification results with different strategies. First, solely using text uncertainty and filtering out uncertain texts are not sufficiently effective. They only improve about +0.05pp top-1 ac-

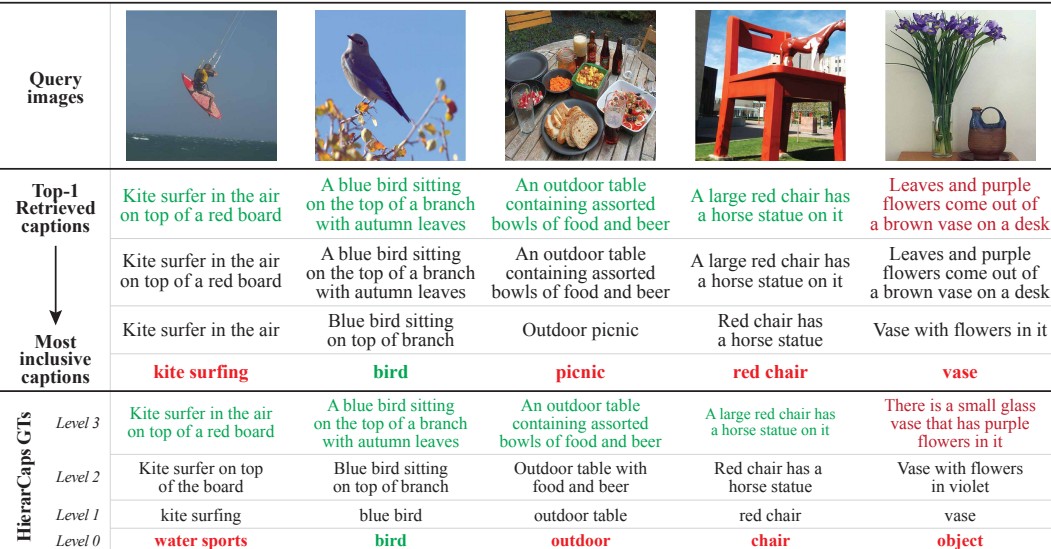

| | | | | | | |
|---|---|---|---|---|---|---|
| **Query images** | | | | | | |
| **Top-1 Retrieved captions** ↓ **Most inclusive captions** | Kite surfer in the air on top of a red board | A blue bird sitting on the top of a branch with autumn leaves | An outdoor table containing assorted bowls of food and beer | A large red chair has a horse statue on it | Leaves and purple flowers come out of a brown vase on a desk | |
| | Kite surfer in the air on top of a red board | A blue bird sitting on the top of a branch with autumn leaves | An outdoor table containing assorted bowls of food and beer | A large red chair has a horse statue on it | Leaves and purple flowers come out of a brown vase on a desk | |
| | Kite surfer in the air | Blue bird sitting on top of branch | Outdoor picnic | Red chair has a horse statue | Vase with flowers in it | |
| | **kite surfing** | **bird** | **picnic** | **red chair** | **vase** | |
| **HierarCaps GTs** | *Level 3* | Kite surfer in the air on top of a red board | A blue bird sitting on the top of a branch with autumn leaves | An outdoor table containing assorted bowls of food and beer | A large red chair has a horse statue on it | There is a small glass vase that has purple flowers in it |
| | *Level 2* | Kite surfer on top of the board | Blue bird sitting on top of branch | Outdoor table with food and beer | Red chair has a horse statue | Vase with flowers in violet |
| | *Level 1* | kite surfing | blue bird | outdoor table | red chair | vase |
| | *Level 0* | **water sports** | **bird** | **outdoor** | **chair** | **object** |

Figure 9: **Image traversals with ProLIP.** For each image, we estimate the `[ROOT]` caption which include the image most using Equation (4). Then, we interpolate the `[ROOT]` caption and the retrieved caption. We compare our interpolation and HierarCaps GTs. Red denotes when the estimated and GT roots are different.

Table 2: **HierarCaps retrieval.** We measure the precision and recall of 50 traversal results. "Prob?" denotes inclusion-based traversal, using the average of `""` and "`[ROOT]`" in Figure 9 as a new `[ROOT]`. R@1$^{[ROOT]}$ is recall of `[ROOT]` embeddings.

| | Prob? | Prec | R@1 | R@1$^{[ROOT]}$ |
|---|---|---|---|---|
| CLIP (1.28B) | ✘ | 25.0 | 63.0 | 0.1 |
| ProLIP (1.28B) | ✘ | 28.4 | 62.6 | 0.1 |
| ProLIP (1.28B) | ✔ | 35.9 | 62.9 | 15.3 |
| ProLIP (12.8B) | ✘ | 31.7 | 67.8 | 0.1 |
| ProLIP (12.8B) | ✔ | 41.1 | 68.0 | 23.3 |

Table 3: **ImageNet prompt tuning by ProLIP.** $K$ denotes the number of few-shot samples for each class (if required). "OpenAI 80 prompts" is the same as "ImageNet" of ProLIP 12.8B in Table 1.

| Prompt strategy | Accuracy | $K$ |
|---|---|---|
| "`a photo of {·}`" | 73.7 | - |
| OpenAI 80 prompts | 74.6 | - |
| Filtering by $\sigma$ stats | 74.6 (+0.03) | - |
| Top-K prompts | 74.7 (+0.07) | - |
| BPRW (proposed) | 74.7 (+0.12) | 0 |
| BPRW (proposed) | 75.6 (+0.99) | 5 |
| BPRW (proposed) | 75.8 (+1.21) | 9 |

curacy. On the other hand, BPRW achieves better accuracy by using additional visual information; we have +0.12pp for ImageNet ZSC. If we can use a few labeled images per class (*e.g.*, five for each class), we can get over 1.2% accuracy improvement by adjusting their weight. In Appendix C.7, we describe more details of BPRW, including hyperparameters and more visualization results of the learned weights by BPRW. Interestingly, the learned weights follow the actual image distributions; if the images are mostly low resolution, "a low resolution" prompt becomes the most significant.

**More examples and discussion.** We include more applications in Appendix C.8, including filtering, dataset understanding, and LongProLIP. Appendix D discusses the diagonal covariance assumption.

## 5 CONCLUSION

In this work, we introduced Probabilistic Language-Image Pre-training (ProLIP), a fully probabilistic vision-language model that addresses the limitations of deterministic embeddings by capturing the inherent multiplicity in image-text relationships. By mapping inputs to random variables and efficiently estimating uncertainty through an "uncertainty token" (`[UNC]`), ProLIP models distributional relationships without additional parameters. The inclusion loss further enhances interpretability by enforcing distributional inclusion between image-text pairs and between original and masked inputs. Our experiments demonstrate that ProLIP is not only beneficial in zero-shot classification tasks but also provides an additional axis of understanding input data by capturing their uncertainty. Our approach highlights the potential of uncertainty modeling in vision-language applications.

## AUTHOR CONTRIBUTIONS

This project is the extension of Chun et al. (2021; 2022); Chun (2024). S Chun led the project; the other authors actively and significantly contributed to the project with advice and feedback. W Kim and S Yun contributed to the baseline openclip implementation and evaluation toolkits from Kim et al. (2024); S Chun developed the main module upon the baseline implementation. The main ideas, such as [UNC], probabilistic pairwise contrastive loss, and inclusion loss, are designed and implemented by S Chun. S Park contributed to constructing the HierarImgs dataset. S Chun wrote the initial version of the manuscript. S Park contributed to a better visualization. All authors contributed to the final manuscript.

## REPRODUCIBILITY STATEMENT

For reproducible research, we use the open-source training dataset (DataComp 1B (Gadre et al., 2024)) and codebase (openclip (Ilharco et al., 2021)). As we clarified in Appendix B.3, we had 1,121,356,767 number of valid URLs among 1.39 billion URLs and 1,118,443,492 number of unique images after de-duplicating URLs (there were 147,676,246 number of duplicated URLs in the DataComp 1B URLs). All the detailed hyperparameters are clarified in Appendix B.1. Our results can be reproducible by our open-source implementation (https://github.com/naver-ai/prolip) and released pre-trained weights in HuggingFace (https://huggingface.co/collections/SanghyukChun/prolip-6712595dfc87fd8597350291); the released weights include ProLIP ViT-B/16 (from-scratch), ViT-L/16, ViT-SO400M/14, ViT-H/14 (fine-tuned), and LongProLIP ViT-B/16 fine-tuned with different datasets.

## ACKNOWLEDGEMENTS

We thank the internal infrastructure team who helped to download the DataComp 1B dataset.

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

# APPENDIX

## A  MORE DETAILS OF PROLIP

### A.1  WHY DO WE NEED PROBABILISTIC REPRESENTATIONS?

Deterministic embeddings (*e.g.*, CLIP (Radford et al., 2021), SigLIP (Zhai et al., 2023)) suffer from the difficulty of representing input data uncertainty (*i.e.*, aleatoric uncertainty). For example, "A person is walking" can be matched to either "A person is walking in the rain" or "A person is walking on sunshine", but in a deterministic embedding space (Figure A.1a), it is challenging to represent the ambiguity of "A person is walking". On the other hand, a probabilistic embedding space (Figure A.1b) can represent this ambiguity by expanding the "area" of the ambiguous embedding. If we assume more ambiguous input, such as "person", a probabilistic embedding space can represent the ambiguous input by assigning a larger uncertainty value to "person". However, a deterministic embedding space will map an input to a specific vector coordinate; one possible choice is to map the uncertain input into the "average" of the possible matched inputs, *e.g.*, let the "person" embedding located to the midpoint of all the person-related text embeddings. However, this approach still cannot capture the complex actual semantic meaning of "person"; it is more complex than the average embedding. This argument is empirically supported by the image traversal experiment in Section 4.4. Using a more proper text embedding as the root embedding [ROOT] performs better than using a native null text embedding. Furthermore, this paper targets the vision-language representation learning scenario, where the ambiguity of inputs is multi-modal; each modality (text and image) can have inherent ambiguity as shown in Figure A.1 and the correspondence between image and text can have ambiguity due to the inherent many-to-many correspondence and abundant false negatives as shown by Chun (2024). Overall, we need to use probabilistic representations rather than deterministic representations to represent the inherent ambiguity of vision-language datasets.

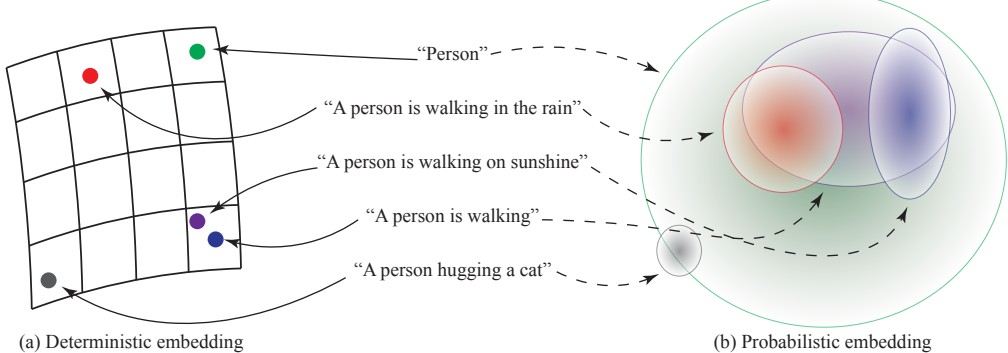

(a) Deterministic embedding          (b) Probabilistic embedding

Figure A.1: **Conceptual comparison between deterministic and probabilistic embedding spaces.** While probabilistic representation space can naturally represent the inherent ambiguity of input data (*i.e.*, aleatoric uncertainty) by estimating the uncertainty of each input, a deterministic embedding space can suffer from mapping complex semantics of ambiguous inputs.

What property should be learned by an ideal PrVLM? An ideal PrVLM should capture three potential input uncertainties: (1) uncertainty from the text modality, (2) uncertainty from the image modality, and (3) uncertainty from the text-image cross-modality. Uni-modal uncertainty is straightforward; if an input has more detailed information (*e.g.*, describing more detailed information in text, or capturing a very detailed and complex scene by photography), then it will have smaller uncertainty, otherwise, it will have larger uncertainty (*e.g.*, providing very high-level caption, such as "person", or only a part object with white background is taken by picture).

The cross-modal uncertainty should capture "how many possible instances can be matched to this input?". We can think this in two different viewpoints: text-to-image and image-to-text. The text-to-image relationship is straightforward. If we have a caption "photo", then it will be matched to all photographs, and if we have a caption with a very detailed description (*e.g.*, the full description of the hotel room), then it will be only matched to specific images. Image-to-text relationships are often determined by the dataset. For example, if we have a caption dataset exactly describing

which objects are in the image, then we can think that an image with less objects will have larger uncertainty. However, if we consider a caption dataset where an image has multiple captions, each caption focuses on completely different objects, then an image with more objects will have larger uncertainty. In other words, unlike text uncertainty originating from a text-to-image relationship, image uncertainty originating from an image-to-text relationship is highly affected by the property of the training dataset.

In practice, because our datasets have a mixed property and their captions are somewhat noisy, our image uncertainty will have a mixed property, namely, unlike text uncertainty, there could be no strong relationship between the absolute uncertainty value and the number of objects (or complexity of images) – Empirically, we have more captions that exactly describing the scene, therefore, a more simple image tends to have larger uncertainty. Also, unlike texts, images always have the same number of pixels; the definition of "information" in images is not as straightforward as that of texts. Since there is no "uni-modal" uncertainty supervision, a PrVLM trained only with image-text relationships will have no guarantee to represent proper image uni-modal uncertainty. To enforce a proper uni-modal image uncertainty, we first propose the uni-modal uncertainty supervision by proposing the inclusion loss, namely, the original image embedding should be included by an image embedding from the masked image.

## A.2 2D VISUALIZATION FOR FIGURE 1

We use a linear projection, such as Principal Component Analysis (PCA), for the visualization. If we use a non-linear projection from a high-dimensional space to a 2-dimensional space, there is no guarantee that the projected Gaussian distribution still follows a Gaussian distribution. Therefore, we project the high-dimensional embeddings to 2-dimensional space using PCA. All models are trained on the DataComp 1B dataset with 1.28B seen samples (*i.e.*, the 1.28B models in Table 1).

For both models, we select ten similar images and their corresponding captions from the Hierar-Caps dataset. Then, for ProLIP, we randomly sample ten embeddings for each embedding using the learned Gaussian distributions. We then apply PCA to the sampled embeddings for visualization.

## A.3 PROBABILISTIC MATCHING LOSS VS. PROBABILISTIC PAIRWISE CONTRASTIVE LOSS

Probabilistic matching loss (PML) by PCME++ (Chun, 2024) and the proposed probabilistic pairwise contrastive loss (PPCL) use almost the same logit based on CSD (Equation (1)). However, they have differences in (1) PML is based on the original CSD, which should compute L2-distance between $\mu$s and (2) PML uses binary cross entropy loss (BCE) while PPCL uses log sigmoid loss.

The first difference can make numerical errors when the difference between $\mu$s is extremely small. It is because L2 distance should compute a "square" operation, *i.e.*, $\sum_{i=1}^{D}(a_i^2 - b_i^2)^2$, where $a_i$ and $b_i$ are scalar value of $i$-th dimension of vector $a$ and $b$. On the other hand, we use logit based on matrix multiplication $\mu_1^\top \mu_2$ (derived in Appendix A.4), showing a more stable and accurate computation in terms of float precision.

The second difference can cause a fast gradient vanishing as already discussed by Chun (2024). Chun (2024) thus employed additional techniques to mitigate the issue, *e.g.*, pseudo positives and mixed data sample augmentation method, such as Mixup (Zhang et al., 2018) and CutMix (Yun et al., 2019). However, in this paper, we omit the techniques because simplicity is important when we train with large-scale training samples. To tackle the issue, we employ log sigmoid loss and a multiplication-based logit, resulting in a fast and stable convergence.

In our ablation study (Appendix C.2), we empirically show that the PML loss used by PCME++ fails to be converged when it is used by a stand-alone way without any deterministic loss (*e.g.*, CLIP loss). On the other hand, PPCL loss converges well even without any additional loss function.

Note that PCME++ and ProLIP use normalized mean for both training and inference to compute cosine similarity as same as CLIP. Also, PCME++ and ProLIP can estimate a Gaussian random variable by the parameterization trick, *i.e.*, $Z = (\mu, \Sigma) = \mu + \Sigma \odot (0, 1)$ (where, $\odot$ denotes the element-wise multiplication). In practice, because we use the closed-form solution for calculating distance (CSD – Equation (1)) and loss functions (inclusion loss, PPCL), no sampling based on a re-parameterization trick is required.

## A.4 Derivation of Equation (2)

We first start from the fact that $d_{L2} = \|\mu_1 - \mu_2\|_2^2 = 2 - 2\mu_1^\top \mu_2 = 2 - 2 \cdot d_{L2cos}$ when $\mu_1$ and $\mu_2$ are L2-normalized (*i.e.*, $\|\mu\|_2^2 = 1$). Namely, a cosine similarity $d_{L2cos} = 1 - \frac{1}{2}d_{L2}$. By replacing $d_{L2}$ to $d_{\text{CSD}}$ (Equation (1)), we can conclude the derivation.

$$d_{CSDcos} = 1 - \frac{1}{2}d_{CSD} = 1 - \frac{1}{2}d_{L2} - \frac{1}{2}tr(\sigma_v^2 + \sigma_t^2) = d_{L2cos} - \frac{1}{2}tr(\Sigma_v + \Sigma_t). \qquad \text{(A.1)}$$

## A.5 Derivation of Equation (3)

Let us assume two Gaussian distributions:

$$p(x) = \frac{1}{\sqrt{2\pi\sigma_1^2}} \exp\left(-\frac{(x-\mu_1)^2}{2\sigma_1^2}\right), \quad q(x) = \frac{1}{\sqrt{2\pi\sigma_2^2}} \exp\left(-\frac{(x-\mu_2)^2}{2\sigma_2^2}\right)$$

First, we take the square of $p(x)$:

$$p(x)^2 = \left(\frac{1}{\sqrt{2\pi\sigma_1^2}} \exp\left(-\frac{(x-\mu_1)^2}{2\sigma_1^2}\right)\right)^2 = \frac{1}{2\pi\sigma_1^2} \exp\left(-\frac{(x-\mu_1)^2}{\sigma_1^2}\right)$$

Now, we compute the integral:

$$\int p(x)^2 q(x) dx = \int_{-\infty}^{\infty} \frac{1}{2\pi\sigma_1^2} \exp\left(-\frac{(x-\mu_1)^2}{\sigma_1^2}\right) \cdot \frac{1}{\sqrt{2\pi\sigma_2^2}} \exp\left(-\frac{(x-\mu_2)^2}{2\sigma_2^2}\right) dx$$

$$= \frac{1}{2\pi\sigma_1^2 \sqrt{2\pi\sigma_2^2}} \int_{-\infty}^{\infty} \exp\left(-\frac{(x-\mu_1)^2}{\sigma_1^2} - \frac{(x-\mu_2)^2}{2\sigma_2^2}\right) dx$$

We expand the terms in the exponent as follows:

$$-\frac{(x-\mu_1)^2}{\sigma_1^2} - \frac{(x-\mu_2)^2}{2\sigma_2^2} = -\left(\frac{1}{\sigma_1^2} + \frac{1}{2\sigma_2^2}\right)x^2 + \left(\frac{2\mu_1}{\sigma_1^2} + \frac{\mu_2}{\sigma_2^2}\right)x - \left(\frac{\mu_1^2}{\sigma_1^2} + \frac{\mu_2^2}{2\sigma_2^2}\right)$$

Let:

$$A = \frac{1}{\sigma_1^2} + \frac{1}{2\sigma_2^2}, \quad B = \frac{2\mu_1}{\sigma_1^2} + \frac{\mu_2}{\sigma_2^2}, \quad C = \frac{\mu_1^2}{\sigma_1^2} + \frac{\mu_2^2}{2\sigma_2^2}$$

The term in the exponent simplifies to:

$$-A\left(x - \frac{B}{2A}\right)^2 + \frac{B^2}{4A} - C$$

Now, the integral becomes:

$$\frac{1}{2\pi\sigma_1^2 \sqrt{2\pi\sigma_2^2}} \exp\left(\frac{B^2}{4A} - C\right) \int_{-\infty}^{\infty} \exp\left(-A\left(x - \frac{B}{2A}\right)^2\right)$$

Using the Gaussian integral formula, we have:

$$\int_{-\infty}^{\infty} \exp\left(-A\left(x - \frac{B}{2A}\right)^2\right) dx = \sqrt{\frac{\pi}{A}}$$

Thus, the integral becomes:

$$\int_{-\infty}^{\infty} p(x)^2 q(x)\, dx = \frac{1}{2\pi\sigma_1^2 \sqrt{2\pi\sigma_2^2}} \exp\left(\frac{B^2}{4A} - C\right) \sqrt{\frac{\pi}{A}}$$

By taking logarithm and omitting constant terms, we have Equation (3).

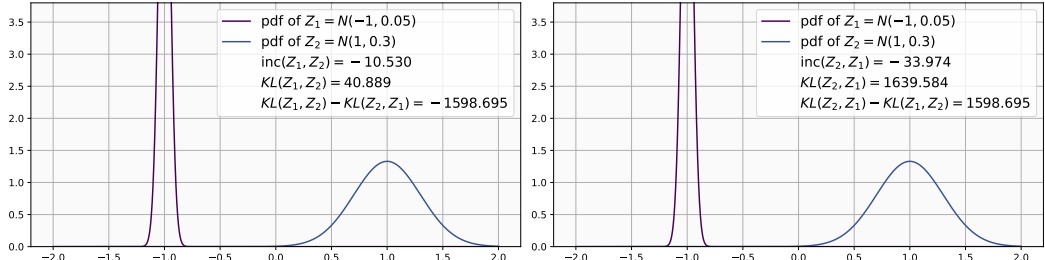

Figure A.2: **More visual examples of the inclusion loss.** The difference of KL and reverse KL (DKL) cannot correctly capture the inclusion relationship.

## A.6 MORE DISCUSSIONS RELATED TO INCLUSION LOSS

Figure A.2 shows more comparisons with the proposed inclusion loss and KL divergence. Specifically, we compare inclusion loss with the difference of KL and reverse KL (DKL), namely, $DKL(Z_1, Z_2) = KL(Z_1, Z_2) - KL(Z_2, Z_1)$. As shown in the figure, the most significant problem with DKL is that it cannot correctly represent the "inclusion" relationship. For example, consider two random variables $Z_1$ and $Z_2$ where they do not include each other as shown in Figure A.2. In this case, although $Z_1$ and $Z_2$ do not include each other, because $DKL(Z_1, Z_2) = -DKL(Z_1, Z_2)$, one of $DKL(Z_1, Z_2)$ or $DKL(Z_2, Z_1)$ will become positive, while the other will become negative. Namely, DKL cannot correctly capture the inclusion relationship. On the other hand, in the same scenario, our inclusion test always returns negative values as shown in the figure, which correctly represents the inclusion relationship.

## A.7 VIB LOSS

VIB loss is a simple regularization term to prevent the collapse of the estimated variance and is widely used by previous probabilistic representation learning methods Oh et al. (2019); Chun et al. (2021); Chun (2024). The VIB loss formulation is as follows:

$$\mathcal{L}_{VIB}(Z) = KL(Z\|\mathcal{N}(0,1)) = -\frac{1}{2}(1 + \log \sigma^2 - \mu^2 - \sigma^2). \tag{A.2}$$

Please refer to Oh et al. (2019) for the full derivation.

## A.8 ALGORITHM OF UNCERTAINTY-AWARE ZERO-SHOT PROMPT SELECTION

**Remark:** Let $\pi_c \in \mathbb{R}^N$ be the weight of each prompt, where $N$ is the number of prompts (*i.e.*, 80 for ImageNet) and $c$ is the class index. Our goal is to find the best $\pi_i$ that the new text embedding $Z_t^{\text{new}} = \sum_i \pi_c^i Z_t^i$ describe the given $M$ image embeddings $Z_v^j$. To achieve this goal, we optimize $\pi_c$ to have the best posterior for $Z_t$ and $Z_v$. For each class $c$, we sample $K$ points from each $Z_v^j$. Namely, we have total $M' = M \times K$ point embeddings as observations for class $c$.

We assume a Dirichlet prior for $\pi_c$ with $\alpha$, where $\alpha$ controls the "uniformity" of the posterior. If we choose large $\alpha$, such as 10, then $\pi$ will be more uniform and if we choose small $\alpha$, such as 0.1, then $\pi$ almost becomes a Direc-delta distribution. We also set the initial $\pi$ with the reversed uncertainty score, *i.e.*, the normalized $1/tr(\Sigma_t)$ for each prompt. We also employ a small trick for stability. After we got the prior value, we add a small $\varepsilon$ to $\Sigma_t$ to make the overall operation stable. Now, we explain the Expectation-Maximization (EM) algorithm for estimating the mixing proportions of a mixture of $N$ Gaussian components, incorporating a Dirichlet prior over the mixing proportions. For simplicity, we omit the class index $c$ for the remaining section.

We place a Dirichlet prior on the mixing proportions for the given $\alpha$:

$$p(\pi) = \text{Dirichlet}(\pi \mid \alpha), \tag{A.3}$$

We have a dataset of $M'$ observations $\{x_j\}_{j=1}^{M'}$ and assume that the data is generated from a mixture of $N$ Gaussian distributions (*i.e.*, the number of prompts).

**E-Step:** Compute the responsibilities $\gamma_{jn}$, which represent the probability that observation $x_j$ was generated by component $n$:

$$\gamma_{jn} = \frac{\pi_n f_n(x_j)}{\sum_{i=1}^{N} \pi_i f_i(x_j)},$$

where $f_n(x_j)$ is the Gaussian probability density function of component $n$ evaluated at $x_j$:

$$f_n(x_j) = \frac{1}{(2\pi)^{D/2}|\Sigma_n|^{1/2}} \exp\left(-\frac{1}{2}(x_j - \mu_n)^\top \Sigma_n^{-1}(x_j - \mu_n)\right).$$

Since $\Sigma_n$ is diagonal, the computation simplifies to:

$$f_n(x_j) = \prod_{d=1}^{D} \frac{1}{\sqrt{2\pi\sigma_{nd}^2}} \exp\left(-\frac{(x_{jd} - \mu_{nd})^2}{2\sigma_{nd}^2}\right).$$

**M-Step:** Compute the effective number of observations assigned to each component:

$$N_n = \sum_{j=1}^{M'} \gamma_{jn}.$$

Then, update the mixing proportions for each prompt $n$:

$$\pi_n = \frac{N_n + \alpha_n - 1}{M' + \sum_{i=1}^{N}(\alpha_i - 1)} = \frac{N_n + \alpha - 1}{M' + N(\alpha - 1)}.$$

Ensure that $\pi_n \geq 0$ and $\sum_{n=1}^{N} \pi_n = 1$. We can simplify the algorithm as follows:

---

**Algorithm 1** Bayesian Prompt Re-Weighting (BPRW)

---

1: **Initialization** ($\alpha$: the Dirichlet prior hyperparameter, $\varepsilon$: the stability hyperparameter)
2: **for** each class $c = 1$ to $C$ **do**
3:   **Collect M' observations** by choosing $M$ number of $Z_v$ and sample $K$ points from each $Z_v$. We can choose $M$ "ground truth" observations where $Z_v$ is the embedding, including the $c$ class. Otherwise, we select $M$ nearest samples from the base text prompt embedding of $c$.
4:   **for** each prompt index $k = 1$ to $K$ **do**
5:     Set initial mixing proportion: $\pi_n \leftarrow \left(\frac{1}{Tr(\Sigma_n)}\right) / \left(\sum_{n=1}^{N} \frac{1}{Tr(\Sigma_n)}\right)$.
6:     Modify covariance matrix for stability: $\Sigma_k \leftarrow \Sigma_k + \varepsilon I$.
7:   **end for**
8:   **repeat**
9:     // E-Step:
10:    **for** each observation $j = 1$ to $M'$ **do**
11:      **for** each prompt index $n = 1$ to $N$ **do**
12:        Compute the responsibility $\gamma_{jn} = \left(\pi_n f_n(x_j)\right) / \left(\sum_{i=1}^{N} \pi_i f_i(x_j)\right)$.
13:      **end for**
14:    **end for**
15:    // M-Step:
16:    **for** each prompt index $n = 1$ to $N$ **do**
17:      Compute $N_n = \sum_{j=1}^{N} \gamma_{jn}$
18:      Update mixing proportion $\pi_n = \frac{N_n + \alpha - 1}{M' + N(\alpha - 1)}$
19:    **end for**
20:  **until** convergence
21: **end for**

---

# B  MORE EXPERIMENTAL DETAILS

## B.1  HYPERPARAMETERS

We use the AdamW (Kingma & Ba, 2015) optimizer following the official `openclip` implementation. We also tried AdamP (Heo et al., 2021), which was the main optimizer of PCME++ (Chun, 2024). We empirically observe that AdamP can improve the overall performances (*e.g.*, average performance from 57.3 to 57.8 under the same setting), but for a fair comparison, we follow the protocol of `openclip` (Ilharco et al., 2021). We use a learning rate of 0.0005, beta1 of 0.9, beta2 of 0.95, weight decay of 0.2, and batch size of 512 for each GPU (*i.e.*, the full batch size is $512 \times 32 = 16384$). We apply 10000 warmup steps and then the learning rate is decayed by cosine learning rate scheduling. We use image augmentations of scaling 0.8 to 1.0, color jittering and grayscale. Among the mini-batch, we select 12.5% image-text pairs and drop 75% tokens of them to compute $\mathcal{L}_{\text{inc}(x \subset x^{\text{masked}})}$. In all experiments, we fix $\alpha_1$ and $\alpha_2$ in Equation (6) as $10^{-7}$ and 0.001, respectively. We tried various combinations of $\alpha$, but we found that the final result is relatively robust to the choice of $\alpha$ unless we choose very large $\alpha$. Similarly, we tested various $\varepsilon$ and $c$ for the inclusion loss (Equation (5)), but as shown in Table C.5, the final result is relatively robust to the choice of the parameters. If not specified, we chose $\varepsilon = -20$ and $c = 1000$, but for future research, we recommend using $\varepsilon = -100$ and $c = 10$, which empirically performs well and stable large-scale training.

## B.2  FINE-TUNING DETAILS

During the project, we indeed aspire to train large models, such as ViT-H, ViT-SO400M, ViT-G or ViT-g. However, as reported by Cherti et al. (2023), ViT-H/14 CLIP model with 34B seen samples (achieving 78.0% ImageNet zero-shot accuracy) task 279 hours with 824 A100s. Using our resource (32 H100s), training the same backbone from scratch takes almost 300 days. For this reason, we tried to fine-tune the pre-trained strong backbones. Note that our goal is not to fine-tune the existing deterministic VLMs, but we report four fine-tuning results, ViT-B/16 (76.0% IN-ZS), ViT-L/16 (80.5% IN-ZS) and ViT-SO400M/14 (82.0% IN-ZS) pre-trained by SigLIP (Zhai et al., 2023) and ViT-H/14 (83.4% IN-ZS) pre-trained by DFN (Fang et al., 2024), for achieving a stronger PrVLM.

For the architectural consistency between all ProLIP models, we remove the attention pooling of pre-trained SigLIP and fine-tune the models using `[CLS]` token-based pooling and fix the image resolution to 224×224. This will lead to slightly worse performance of the fine-tuned models than the original models. Note that we can implement both attention pooling and `[UNC]` token architecture simultaneously, but we did not implement the architecture for simplicity. We again emphasize that we do not aim to achieve the state-of-the-art performance, instead, our goal is to verify whether ProLIP performs better by scaling up the architecture beyond ViT-B/16. For the fine-tuning, we use the learning rate as 5.0e-5, the weight decay as 0.0, and the number of seen samples as 1.28B.

## B.3  TRAINING AND EVALUATION DATASETS

We mainly use the DataComp 1B dataset (Gadre et al., 2024), a filtered version of the LAION-5B dataset (Schuhmann et al., 2022), as our training dataset. We had 1,121,356,767 number of valid URLs among 1.39 billion URLs and 1,118,443,492 number of unique images after de-duplicating URLs (there were 147,676,246 number of duplicated URLs in the DataComp 1B URLs). For the ablation study, we use ConceptualCaption 3M (Sharma et al., 2018), ConceptualCaption 12M (Changpinyo et al., 2021) and RedCaps (Desai et al., 2021) datasets.

We use 38 tasks from the DataComp evaluation suite: ImageNet (Russakovsky et al., 2015), 6 benchmarks for evaluating robustness under ImageNet distribution shifts, including ImageNet-A, ImageNet-O (Hendrycks et al., 2021b), ImageNet-R (Hendrycks et al., 2021a), ImageNet v2 (Recht et al., 2019), ImageNet-Sketch (Wang et al., 2019) and ObjectNet (Barbu et al., 2019), and 13 VTAB task (Zhai et al., 2019), including Caltech-101 (Fei-Fei et al., 2004), CIFAR-100 (Krizhevsky, 2009), CLEVR Counts, CLEVR Distance (Johnson et al., 2017), Describable Textures (Cimpoi et al., 2014), EuroSAT (Helber et al., 2019), KITTI Vehicle Distance (Geiger et al., 2012), Oxford Flowers-102 (Nilsback & Zisserman, 2008), Oxford-IIIT Pet (Parkhi et al., 2012), PatchCamelyon (Veeling et al., 2018), RESISC45 (Cheng et al., 2017), SVHN (Netzer et al., 2011) and SUN397 (Xiao et al., 2010), and 3 retrieval tasks, including Flickr (Young et al., 2014), MS-COCO Caption (Chen et al., 2015) and WinoGAViL (Bitton et al., 2022). There are also 13 additional tasks,

such as CIFAR-10, Country211 (Radford et al., 2021), FGVC Aircraft (Maji et al., 2013), Food-101 (Bossard et al., 2014), GTSRB (Stallkamp et al., 2012), MNIST (LeCun et al., 1998), Pascal VOC (Everingham et al., 2010), Rendered SST2 (Radford et al., 2021), STL-10 (Coates et al., 2011), iWildCam (Beery et al., 2021), FMoW (Christie et al., 2018), Dollar Street (Rojas et al., 2022), and GeoDE (Ramaswamy et al., 2024).

For performing zero-shot classification, we use CSD as the distance between the given image and the pre-extracted text template features representing each class. As CSD uses uncertainty (Equation (1), this protocol uses uncertainty value for the zero-shot classification.

### B.4 HIERARCAPS AND IMAGE TRAVERSAL DETAILS

The HierarCaps dataset (Alper & Averbuch-Elor, 2024) is a human-validated caption dataset where each image has four levels of captions. For example, "water sports" ⇒ "kite surfing" ⇒ "kite surfer on top of the board" ⇒ "kite surfer in the air on top of a red board". These captions are human-validated, namely, the level of the HierarCaps dataset is aligned to the "hierarchical perception" of humans. As shown in Figure 7, the human-validated level of HierarCaps (*i.e.*, human intuitions) is aligned well to the learned uncertainty by ProLIP.

We also showed that the learned uncertainty can improve the image traversal task. The traversal task needs two information: the closed text embedding and [ROOT] embedding for the given image. Once the closed text embedding and [ROOT] embedding are chosen, we interpolate them with 50 equally spaced steps and find the closest text from the database for each interpolated caption. Uncertainty information is used for the traversal task in two perspectives. First, we use CSD to retrieve captions, where CSD uses the uncertainty (Equation (1)). More importantly, we estimate the [ROOT] embedding using uncertainty. Previous approaches use the average text embedding or null text embedding as [ROOT] embedding. Instead of using average or null text embedding, we propose to use uncertainty-based [ROOT] embedding. As clarified in Section 4.4, we first retrieve the most similar caption of the given image (following the common protocol (Alper & Averbuch-Elor, 2024)). Then, we search for the most inclusive caption of the retrieved caption in the database using the inclusion measure that we proposed. Then, we use the most inclusive caption as the [ROOT] embedding, which is a more plausible "root" compared to the average or null embedding.

As shown in Figure B.1, the [ROOT] embedding of CLIP is always closest to the "umbrella", which makes the image traversal by CLIP inaccurate. On the other hand, the [ROOT] embedding of ProLIP correctly estimates the true hierarchy of the given image query.

### B.5 HIERARCHICAL IMAGES DATASET CONSTRUCTION

We construct the hierarchical image dataset by using the validation set of the COCO dataset (Lin et al., 2014), which includes both images and their corresponding segmentation maps. First, the images were filtered to retain only those where the smallest segment was larger than 5000 pixels. Then, the three largest segments were extracted from each image and pasted onto a blank canvas in descending order of size. We illustrate the generated samples in Figure B.2.

## C MORE RESULTS

### C.1 UNCERTAIN AND LONG CAPTIONS

The following single caption shows a very high uncertainty value despite its length (*e.g.*, top 20% uncertain samples in Figure 4). Note that Figure 6 shows that a caption with longer length might have a higher uncertainty, but it is not always true if the caption only has almost useless information. Below, "UNC" denotes the uncertainty value of the corresponding caption, namely $tr(\Sigma_t)$.

UNC: 0.0415 *"Searching for Meaning: Idealism, Bright Minds, Disillusionment, and Hope (Third in a Series of See Jane Win(tm) Books) Cover Image"*
UNC: 0.0415 *"Graphic design profession workdesk monitor printer books lamp pc computer stock illustration"*
UNC: 0.0416 *"Photo ID: 2299331 Views: 14221 UK - Air Force Eurofighter EF-2000 Typhoon FGR4 (ZK306) shot at Fairford (FFD / EGVA) UK - England July 21, 2013 By Michael Brazier-Aviation-Images"*
UNC: 0.0416 *"Vigo 36 inch Farmhouse Apron Single Bowl 16 Gauge Stainless Steel Kitchen Sink with Zurich Chrome Faucet, Grid, Strainer and Soap Dispenser"*
UNC: 0.0418 *"XIAOMI iHealth PT-101B Medical Baby High Sensitivity LED Electric Thermometer Underarm/Oral Soft Head Thermometer Adult BabyTthermometer Sensor"*
UNC: 0.0462 *"Maidofhonortoastus Scenic Sample Invoices Created With Our Online Invoicing Software With Licious Sample Invoice Template With Comely Free Excel Invoice Also Invoice Template For Self Employed In Addition Used Vehicle Invoice And Invoice Payment Reminder As Well As Print Invoice Template Additionally It Services Invoice Template From Invoiceberrycom With Maidofhonortoastus Licious Sample Invoices Created With Our Online Invoicing Software With Comely*

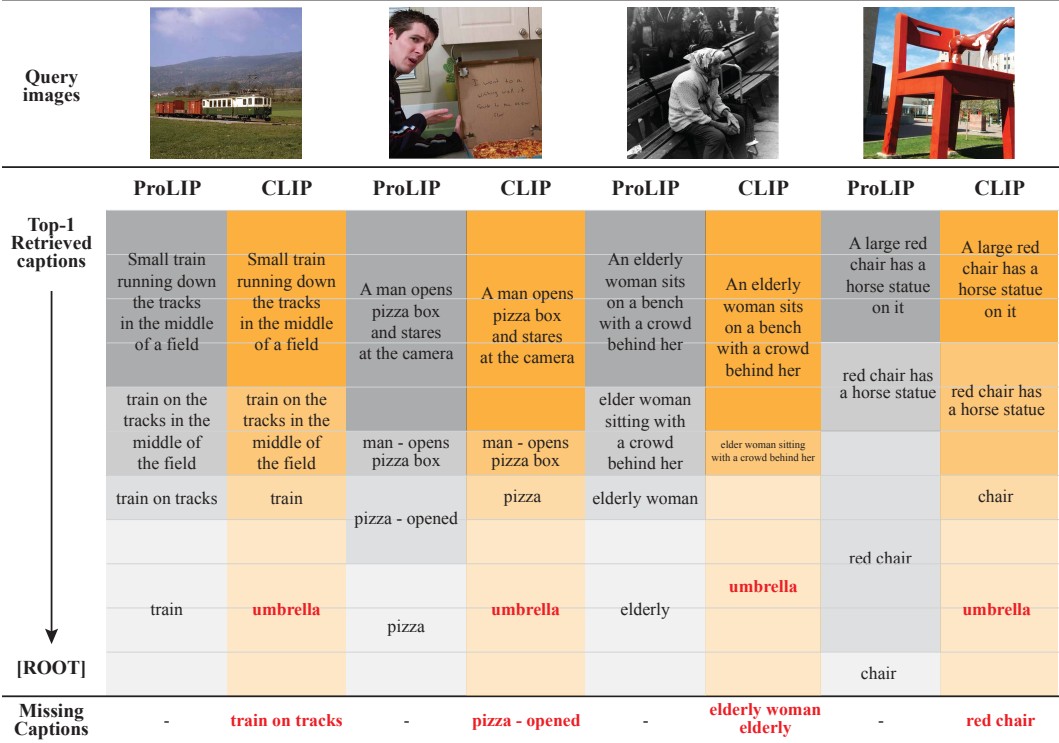

Figure B.1: **Image traversal comparison between ProLIP and CLIP.** The image traversal examples on HierarCaps (Alper & Averbuch-Elor, 2024). The **highlighted captions** denote the wrong captions.

*Sample Invoice Template And Scenic Free Excel Invoice Also Invoice Template For Self Employed In Addition Used Vehicle Invoice From Invoiceberrycom 0."*
`UNC: 0.0462` *"Carterusaus Ravishing Invoice Freewordtemplatesnet With Inspiring Proforma Invoice With Easy On The Eye Chicken Receipts Also I Receipt Notice In Addition Cash Receipts Template And Read Receipt For Gmail As Well As Annual Gross Receipts Additionally Platepass Receipt From Freewordtemplatesnet With Carterusaus Inspiring Invoice Freewordtemplatesnet With Easy On The Eye Proforma Invoice And Ravishing Chicken Receipts Also I Receipt Notice In Addition Cash Receipts Template From Freewordtemplatesnet"*
`UNC: 0.0466` *"Aldiablosus Ravishing Addition Worksheets Dynamically Created Addition Worksheets With Entrancing Addition Worksheets With Appealing Phonics Worksheets Free Also Functions Worksheet Algebra In Addition Cellular Respiration Worksheet Middle School And Beginning Addition Worksheets As Well As Real World Math Worksheets Additionally Chemical Change Worksheet From Mathaidscom With Aldiablosus Entrancing Addition Worksheets Dynamically Created Addition Worksheets With Appealing Addition Worksheets And Ravishing Phonics Worksheets Free Also Functions Worksheet Algebra In Addition Cellular Respiration Worksheet Middle School From Mathaidscom"*

## C.2 MORE ZERO-SHOT CLASSIFICATION EXPERIMENTS

We show the full results of 38 tasks in Table C.1. We can observe that in most benchmarks, ProLIP outperforms CLIP and SigLIP. We also fine-tune the pre-trained SigLIP or CLIP models with our probabilistic training strategy. As we clarified in Appendix B.2, these models were slightly modified due to the architectural difference between the original pre-trained model and our main ProLIP model (*e.g.*, attention pooling vs. `[CLS]` token pooling, and image resolution). Table C.2 shows that the overall performance is improved by increasing the parameter size (from 62.1 to 66.9).

## C.3 ABLATION STUDY

In this subsection, we conduct the ablation study of our design choices. We train the models on Conceptual Caption 3M (Sharma et al., 2018), Conceptual Caption 12M (Changpinyo et al., 2021) and RedCaps (Desai et al., 2021) datasets with 96M seen samples. This efficient setting enables to train a model in 7 hours with 8 A100 NVIDIA GPUs. We measure the effectiveness of our design choice in three categories. First, we measure ImageNet zero-shot top-1 accuracy (IN-Top1). Second, we measure the average $\sigma^2$ values of visual and textual modalities. If a model captures the inherent uncertainty well, we assume that it will have a higher uncertainty for captions, rather than images. Finally, we measure the HierarCaps recall, to measure whether the learned uncertainty captures the hierarchy of captions well.

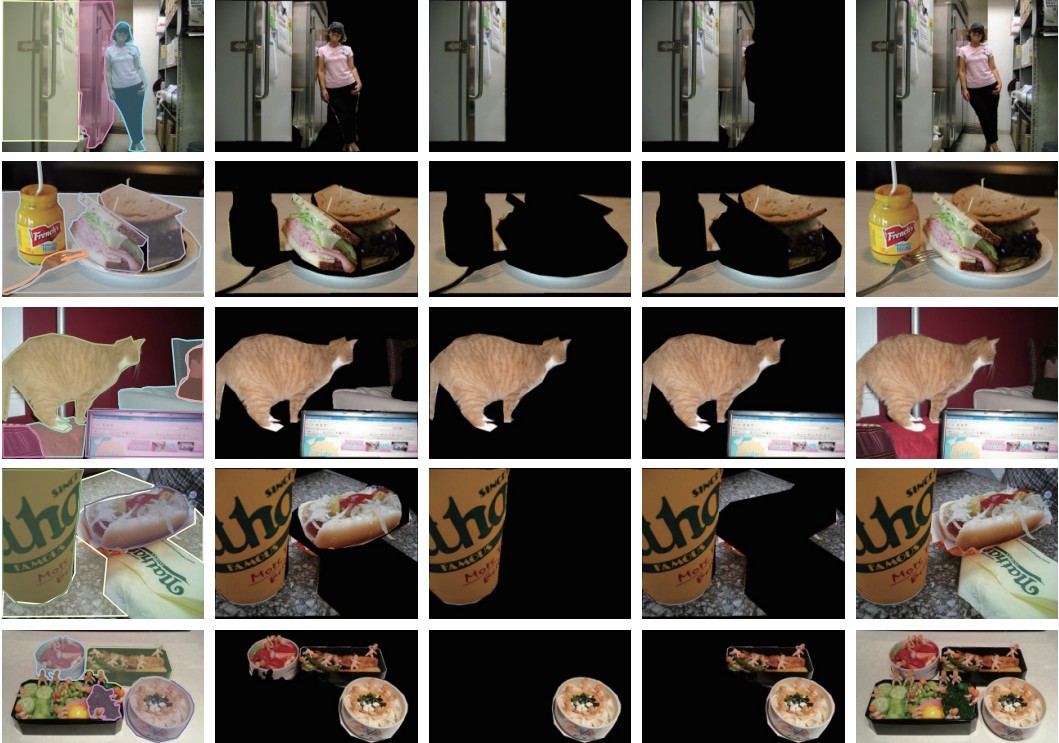

Figure B.2: **Hierarchical COCO samples.**

Table C.3 shows that (1) PCME++ loss is not trainable when we use more limited architecture than multi-head architecture for estimating uncertainty. It supports our assumption discussed in Appendix A.3. (2) PCME++ loss becomes learnable if we use deterministic loss together. (3) $\mathcal{L}_{\text{inc}(v \subset t)}$ enforces images to belong into texts, namely image uncertainty tends to be smaller than text uncertainty with the inclusion loss. (4) $\mathcal{L}_{\text{inc}(x \subset x^{\text{masked}})}$ makes the embedding space can capture the hierarchy of data, namely, improves HierarCaps recall.

Although Table C.3 shows a good intuition for each loss function, we remark that the learned uncertainty might need more training time to be more accurate. From this observation, we also compare ProLIP models by ablating the main losses on DataComp 1B with 1.28B seen samples. Table C.4 shows that using all proposed objectives achieves the best ImageNet and VTAB results. Although it shows slightly worse performance in retrieval, we found that its performance is reasonably good compared to other variants, especially for the baseline (first row) in all measurements.

Table C.5 shows the impact of the choice of the stability parameters for inclusion loss. As we mentioned in Appendix B.1, the final performance is relatively robust to the choice of hyperparameters. In practice, we recommend $\varepsilon = -100$ and $c = 10$ for both stable training and performance.

We also compare the efficiency of the proposed [UNC] token architecture compared to the deterministic baseline and multi-head uncertainty estimate module by PCME++. We compare them with five different architectures (ViT-B/32, ViT-B/16, ViT-B/16 with 768-width, ViT-L/16, and ViT-L/14). Here, we only compare image encoders because image encoders always take the same image token length, which makes it easier to compare the impact of different design choices. We report (1) the input image token sizes for each architecture ("[UNC] token architecture" uses one more), (2) the base parameters of the deterministic one, (3) inference time for 50k ImageNet validation images (lower is faster), and (4) additional parameters compared to the deterministic baseline.

Table C.6 shows three findings. First, even if we increase the input token length, the actual inference speed is not quadratically slow – ViT-B/32 and ViT-B/16 have the same Transformer capability but only input lengths are different (49 vs. 196), but their inference times are 75.18s and 76.68s, which is an almost neglectable change. If we choose a larger model (*e.g.*, ViT-L), the difference

Table C.1: **Zero-shot classification full results.** "FT" denotes the fine-tuned results of the pre-trained models with deterministic objectives. Details of fine-tuning can be found in Appendix B.2.

| | Caltech-101 | CIFAR-10 | CIFAR-100 | CLEVR Counts | CLEVR Distance | Country211 | Describable Textures | EuroSAT | FGVC Aircraft | Food-101 | GTSRB | ImageNet 1k | ImageNet Sketch | ImageNet v2 | ImageNet-A | ImageNet-O | ImageNet-R | KITTI Vehicle Distance | MNIST | ObjectNet |
|---|---|---|---|---|---|---|---|---|---|---|---|---|---|---|---|---|---|---|---|---|
| | | | | | | | | | 1.28B seen samples | | | | | | | | | | | |
| CLIP | 91.75 | 94.34 | 77.85 | 18.01 | 15.79 | 15.63 | 60.43 | 45.09 | 20.77 | 86.55 | 50.27 | 67.24 | 55.70 | 58.95 | 30.73 | 53.70 | 76.35 | 30.80 | 72.96 | 55.39 |
| SigLIP | 92.77 | 94.17 | 78.25 | 20.11 | 15.87 | 15.18 | 60.90 | 39.15 | 21.01 | 85.96 | 40.71 | 67.37 | 55.93 | 59.67 | 30.80 | 54.25 | 76.41 | 18.00 | 73.17 | 55.60 |
| ProLIP | 92.43 | 93.97 | 77.48 | 22.38 | 15.51 | 15.29 | 61.28 | 40.17 | 21.78 | 85.29 | 51.32 | 67.76 | 56.34 | 59.91 | 30.92 | 53.65 | 76.24 | 39.10 | 74.78 | 54.94 |
| ProLIP (ViT-B FT) | 93.69 | 96.39 | 83.13 | 21.03 | 15.17 | 18.35 | 65.32 | 57.28 | 40.31 | 90.55 | 47.39 | 74.60 | 64.78 | 66.03 | 46.81 | 43.55 | 86.63 | 27.29 | 84.33 | 65.44 |
| ProLIP (ViT-L FT) | 94.58 | 98.28 | 87.79 | 23.31 | 16.27 | 26.56 | 70.53 | 71.26 | 49.91 | 94.20 | 52.76 | 79.40 | 70.28 | 72.77 | 67.67 | 33.45 | 92.26 | 19.97 | 85.05 | 75.37 |
| ProLIP (SO400M FT) | 94.29 | 98.17 | 87.92 | 23.02 | 16.69 | 29.75 | 70.27 | 64.96 | 46.11 | 94.27 | 57.22 | 79.26 | 69.74 | 72.55 | 70.05 | 33.15 | 92.32 | 32.91 | 74.22 | 76.00 |
| ProLIP (ViT-H FT) | 95.05 | 98.29 | 87.54 | 23.11 | 22.90 | 30.64 | 70.64 | 59.80 | 49.85 | 94.64 | 61.91 | 79.41 | 69.61 | 72.83 | 67.97 | 33.00 | 92.07 | 16.03 | 87.35 | 74.43 |
| ProLIP (12.8B) | 93.61 | 96.42 | 83.25 | 29.78 | 15.13 | 21.29 | 66.91 | 60.89 | 38.02 | 91.03 | 52.79 | 74.56 | 63.65 | 66.66 | 50.25 | 45.40 | 86.00 | 32.21 | 84.47 | 65.80 |

| | Oxford Flowers-102 | Oxford-IIIT Pet | Pascal VOC 2007 | PatchCamelyon | Rendered SST2 | RESISC45 | Stanford Cars | STL-10 | SUN397 | SVHN | iWildCam | Camelyon17 | FMoW | Flickr | MSCOCO | WinoGAViL | Dollar Street | GeoDE | Average |
|---|---|---|---|---|---|---|---|---|---|---|---|---|---|---|---|---|---|---|---|
| CLIP | 69.67 | 88.57 | 81.38 | 56.67 | 50.08 | 58.57 | 83.09 | 96.78 | 66.43 | 59.68 | 11.16 | 56.03 | 10.57 | 71.09 | 45.49 | 43.46 | 56.43 | 86.97 | 57.12 |
| SigLIP | 68.62 | 88.31 | 81.46 | 56.66 | 49.20 | 55.62 | 83.82 | 96.23 | 67.60 | 61.70 | 10.07 | 63.84 | 11.69 | 71.76 | 45.47 | 43.09 | 57.48 | 87.56 | 56.72 |
| ProLIP | 71.16 | 89.05 | 80.03 | 65.76 | 49.75 | 57.75 | 85.66 | 96.41 | 65.67 | 62.60 | 10.13 | 64.34 | 8.99 | 71.13 | 45.73 | 42.12 | 56.78 | 86.97 | 57.91 |
| ProLIP (ViT-B FT) | 79.76 | 93.49 | 79.37 | 55.87 | 55.68 | 65.30 | 91.21 | 98.26 | 70.26 | 68.15 | 15.28 | 63.47 | 11.15 | 79.00 | 51.71 | 44.32 | 61.80 | 88.81 | 62.13 |
| ProLIP (ViT-L FT) | 83.24 | 95.27 | 81.13 | 58.07 | 57.66 | 71.59 | 93.72 | 99.02 | 73.97 | 66.53 | 16.23 | 66.76 | 18.24 | 82.26 | 55.76 | 45.86 | 67.06 | 91.15 | 65.93 |
| ProLIP (SO400M FT) | 84.18 | 95.55 | 82.58 | 58.54 | 66.78 | 74.78 | 93.20 | 98.84 | 74.25 | 68.60 | 16.66 | 67.29 | 23.18 | 83.46 | 56.61 | 47.35 | 66.36 | 91.02 | 66.63 |
| ProLIP (ViT-H FT) | 82.78 | 95.64 | 82.82 | 64.25 | 59.80 | 72.94 | 94.62 | 99.12 | 75.08 | 71.47 | 13.40 | 79.91 | 20.56 | 81.55 | 55.82 | 47.38 | 66.24 | 91.73 | 66.90 |
| ProLIP (12.8B) | 78.37 | 93.45 | 81.74 | 61.41 | 54.31 | 68.27 | 91.33 | 97.91 | 71.35 | 72.77 | 12.64 | 57.85 | 15.12 | 79.97 | 53.18 | 45.56 | 62.27 | 90.31 | 63.31 |

Table C.2: Comparison of fine-tuned ProLIP with 1.28B seen samples. ViT-H/14 is based on CLIP by DFN (Fang et al., 2024), while the other models are based on SigLIP (Zhai et al., 2023). FLOps are for the base pre-trained models, not modified architecture by ProLIP.

| Backbone | FLOps | # Samples Seen | ImageNet | IN dist. shifts | VTAB | Retrieval | Average |
|---|---|---|---|---|---|---|---|
| ViT-B/16 | 44.44G | 1.28B* | 74.6 | 62.2 | 61.2 | 58.3 | **62.1** |
| ViT-L/16 | 136.41G | 1.28B* | 79.4 | 68.6 | 64.0 | 61.3 | **65.9** |
| ViT-SO400M/14 | 233.54G | 1.28B* | 79.3 | 69.0 | 65.1 | 62.5 | **66.6** |
| ViT-H/14 | 381.68G | 1.28B* | 79.4 | 68.3 | 64.4 | 61.6 | **66.9** |

becomes larger, *e.g.*, 137.77s vs. 176.95s, but it is still not a quadratic order. Second, [UNC] token adds almost neglectable parameters (0.3M for B and 0.6M for L) and inference time compared to the deterministic one. Finally, the multi-head architecture needs a large number of additional parameters (*e.g.*, 20M for B and 40M for L) and shows slower inference time, especially for a larger network (*e.g.*, 176.95s vs. 191.68s for ViT-L/14). Furthermore, in practice, multi-head architecture requires more memory than [UNC] token, which makes it difficult to use a large batch size and scale up to a larger backbone. On the other hand, [UNC] token only needs almost neglectable additional parameters, inference speed, and memory size, which makes it easier to scale up.

## C.4 MORE VISUAL EXAMPLES

We visualize more samples with combinations of various image and text uncertainties. Figure C.1 shows the example image-text pairs with their uncertainty values and similarity score measured by CSD (Equation (1)). The results are similar to Figure 5.

## C.5 MORE RESULTS FOR HIERARIMGS

Unlike text uncertainty (Figure 7), we observe that the visual uncertainty values are not discriminative by the levels in contrast to text uncertainty – See Figure C.2a. Lower-level images tend to have a larger average uncertainty (0.015) than the original images (0.014), but their differences are not

Table C.3: **Ablation study.** All models are trained on Conceptual Caption 3M (Sharma et al., 2018), 12M (Changpinyo et al., 2021) and RedCaps (Desai et al., 2021) where the number of seen samples is 96M.

| Loss | Unc Arch | $\mathcal{L}_{\text{inc}(v \subset t)}$ | $\mathcal{L}_{\text{inc}(x \subset x^{\text{masked}})}$ | IN-Top1 | avg $\sigma_v^2$ | avg $\sigma_t^2$ | H.Cap Recall |
|---|---|---|---|---|---|---|---|
| CLIP | - | - | - | 35.5 | - | - | 48.4 |
| PCME++ + CLIP | multi-head | ✗ | ✗ | 33.6 | 0.0160 | 0.0077 | 55.2 |
| PCME++ | [UNC] | ✗ | ✗ | 0.1 | 0.0354 | 0.0347 | - |
| PCME++ + CLIP | [UNC] | ✗ | ✗ | 36.1 | 0.2552 | 0.2118 | 53.8 |
| ProLIP | [UNC] | ✗ | ✗ | 37.4 | 0.3276 | 0.0745 | 44.8 |
| ProLIP | [UNC] | ✔ | ✗ | 36.8 | 0.0076 | 0.2324 | 46.7 |
| ProLIP | [UNC] | ✗ | ✔ | 37.5 | 0.3319 | 0.0610 | 47.9 |
| ProLIP | [UNC] | ✔ | ✔ | 37.0 | 0.0086 | 0.2254 | 54.8 |

Table C.4: **Large-scale ablation.** All models are ViT-B/16 trained on DataComp 1B with 1.28B seen samples.

| $\mathcal{L}_{\text{inc}(v \subset t)}$ | $\mathcal{L}_{\text{inc}(x \subset x^{\text{masked}})}$ | ImageNet | IN dist. shifts | VTAB | Retrieval | Average |
|---|---|---|---|---|---|---|
| ✗ | ✗ | 67.0 | 54.6 | 56.2 | **53.6** | 56.6 |
| ✔ | ✗ | 67.3 | 54.6 | 56.4 | 53.2 | 57.0 |
| ✗ | ✔ | 67.4 | 54.4 | 56.4 | 53.2 | 56.7 |
| ✔ | ✔ | **67.6** | **55.0** | **57.1** | 53.4 | **57.3** |

Table C.5: **Impact of $\varepsilon$ and $c$ for the inclusion loss.** Details are the same as Table C.4.

| $\varepsilon$ | $c$ | ImageNet | IN dist. shifts | VTAB | Retrieval | Average |
|---|---|---|---|---|---|---|
| -20 | 1000 | 67.6 | 55.0 | 57.1 | 53.4 | 57.3 |
| -10 | 1000 | 67.4 | 55.1 | 57.3 | 53.1 | 57.1 |
| -5 | 1000 | 67.7 | 55.2 | 55.9 | 52.9 | 56.6 |
| -5 | 100 | 67.8 | 55.3 | 56.7 | 53.0 | 57.5 |
| -5 | 10 | 68.0 | 55.5 | 56.8 | 53.6 | 57.4 |
| -10 | 10 | 67.8 | 55.3 | 58.5 | 53.0 | 57.9 |
| -100 | 10 | 67.7 | 55.5 | 56.9 | 53.7 | 57.5 |

significant between levels as texts. Instead of plotting every image in the same histogram, we plot the difference of uncertainty between the original image and its maksed versions in Figure C.2 (b-d).

To understand why some images have reversed image uncertainty by their hierarchy, we visualize the images whose original image is not included by level 0 images. Interestingly, as shown in Figure C.3, we can observe that many images with improper uncertainty estimate by hierarchy actually have wrong visual semantic hierarchy with severely occluded main objects. For example, in the upper row image, the dog in the image appears at the second level, but it only reveals its part rather than the whole body. These results show that proper filtering on the HierarImgs dataset should be required for a more reliable evaluation.

## C.6  MORE DISCUSSIONS FOR HUMAN PREFERENCE AND LEARNED UNCERTAINTY

In Figure 7, we show that the learned uncertainty by ProLIP is well separable by the hierarchy of HierarCaps which is validated by humans. Namely, a HierarCaps caption quadruple from level 0 (the most abstract one) to level 3 (the most detailed one) has an inclusion relationship verified by humans. From this observation, we can conduct a virtual human study to determine whether the learned uncertainty correctly captures human preference.

First, we measure how consistently the numbers in each HierarCaps caption quadruple are ordered decreasingly. Namely, we compute the following metric:

$$\frac{1}{\|\mathcal{T}\|} \sum_{t \in \mathcal{T}} \mathbb{I}_{(tr(\Sigma_t^i) > tr(\Sigma_t^{i+1}))}, \quad \text{(C.1)}$$

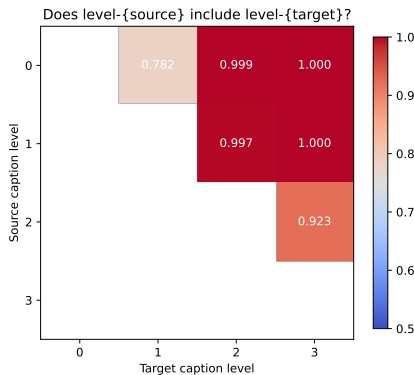

Figure C.4: **The uncertainty value comparison between different levels.**

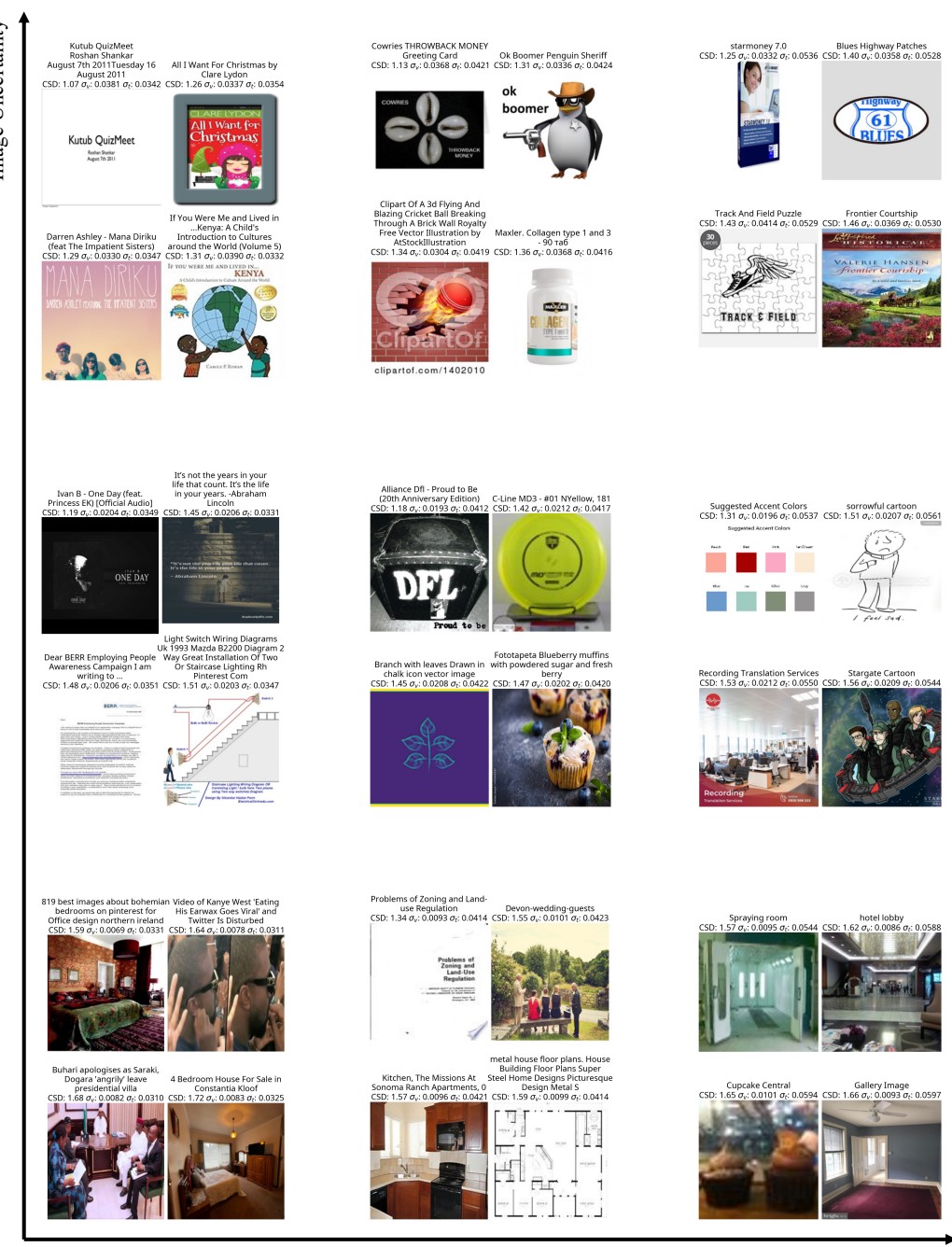

Figure C.1: **Samples with high/low image/text uncertainty.** Samples are drawn from the DataComp small subset. We also report CSD (similarity, lower is closer) between the pair, $\sigma_v^2$, and $\sigma_t^2$ (lower is certain).

where $\mathbb{I}$ denotes the indicator function and $\Sigma_t^i$ denotes the uncertainty value of the level $i$ of the caption $t$. Using ViT-B/16 ProLIP model with 12.8B seen samples, 90.0% of adjacent uncertainties satisfy decreasing order.

Second, as similar to Figure C.2, we show the uncertainty value comparison between different levels in Figure C.4. In the figure, we can observe that most of the level 3 (all) and 2 ($\geq$ 0.997) captions have smaller uncertainty values than their corresponding level 0 and level 1 captions. Also, large number of level 3 captions (0.923) have smaller uncertainty values than their level 2 captions. We

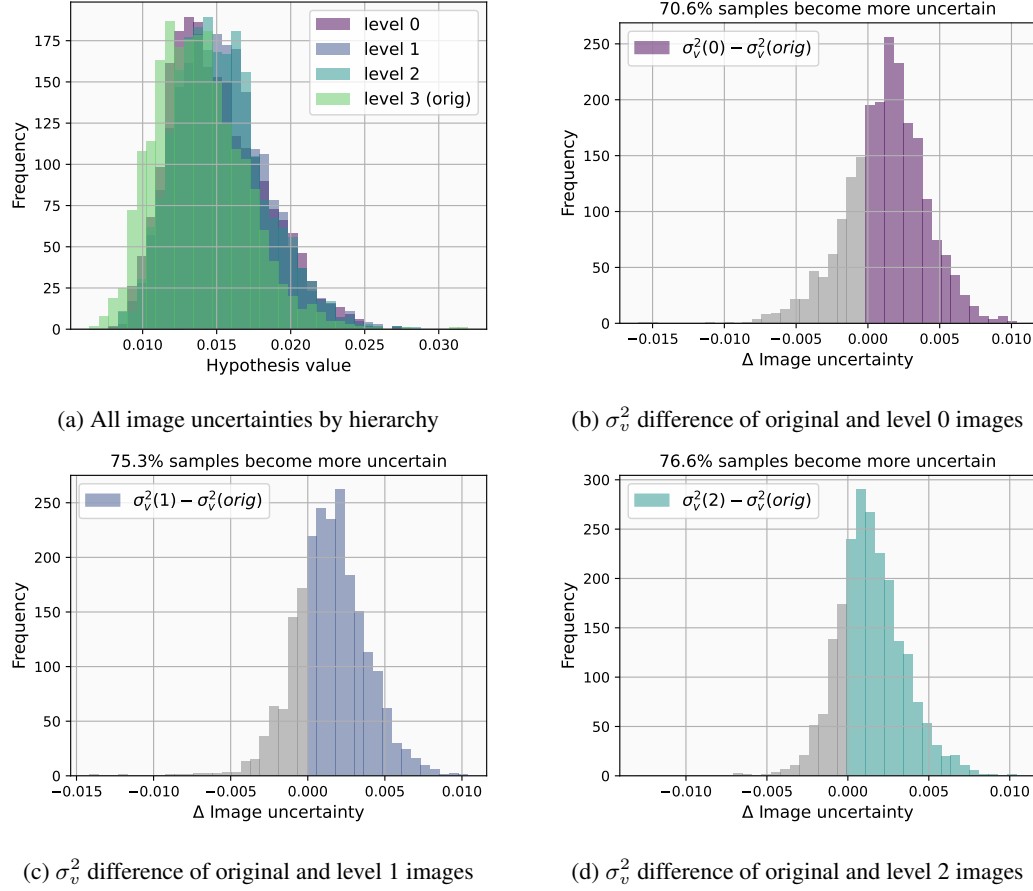

(a) All image uncertainties by hierarchy

(b) $\sigma_v^2$ difference of original and level 0 images

(c) $\sigma_v^2$ difference of original and level 1 images

(d) $\sigma_v^2$ difference of original and level 2 images

Figure C.2: **HierarImgs $\sigma_v^2$ statistics.**

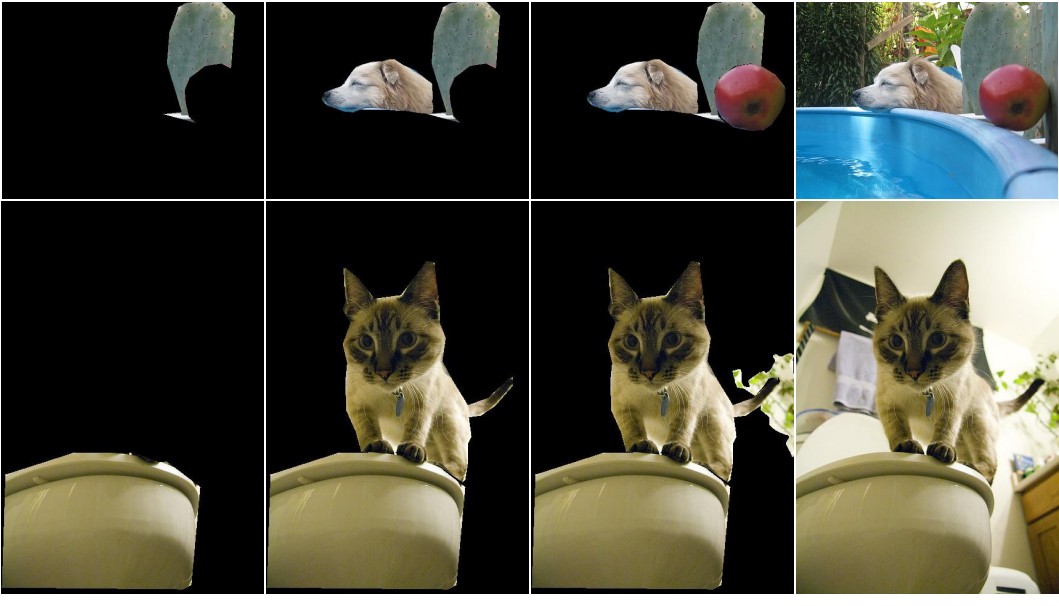

Figure C.3: **Examples of HierarImgs when ProLIP fails to capture $\sigma_v^2$ by hierarchy.**

Table C.6: **Uncertainty architecture comparisons.** The total inference speeds (in seconds) of each architecture for 50k ImageNet validation images are shown (lower is better). The numbers in parentheses denote the number of additional parameters compared to the deterministic baseline model.

|  | ViT-B/32 | ViT-B/16 | ViT-B/16-768 | ViT-L/16 | ViT-L/14 |
|---|---|---|---|---|---|
| # img tokens | 49 | 196 | 196 | 196 | 256 |
| Base param | 151.7M | 150.0M | 197.3M | 428.5M | 428.4M |
| Multi-head | 75.75s (+20.7M) | 78.18s (+20.7M) | 80.08s (+28.9M) | 146.12s (+40.0M) | 191.68s (+40.0M) |
| [UNC] (proposed) | 75.24s (+0.3M) | 76.84s (+0.3M) | 78.61s (+0.6M) | 137.99s (+0.6M) | 177.82s (+0.6M) |
| Deterministic | 75.18s | 76.68s | 78.53s | 137.77s | 176.95s |

found the level 0 and level 1 captions have relatively similar uncertainty values (but still 78.2% level 0 captions have larger uncertainty than their level 1 caption, but as we observed in Figure 9 and B.1, the difference between level 0 and 1 captions are often vague (*e.g.*, water sports vs. kite surfing). Overall, we argue that ProLIP captures human uncertainty preference well as supported by Figure C.4.

## C.7 MORE DISCUSSIONS FOR BAYESIAN PROMPT RE-WEIGHTING (BPRW)

**Hyperparameters.** When ground-truth labels are not accessible (*i.e.*, $K = 0$ in Table 3), we set the $\alpha$ to 5. Then, we select 5 nearest images from each class embedding made by 80 prompts. We then sample 10 samples from the selected image embeddings and get 100 point image embeddings. Now, the sampled embeddings are used as observation of the algorithm (See Appendix A.8). We use $\varepsilon = 0.02$ for a stable convergence. For few-shot settings with $K > 0$, we set $\alpha$ to 2 and select $\frac{100}{K}$ samples (*e.g.*, if $K = 9$, then we sample 11 point vectors from the image embeddings).

**Visualization of the learned weight.** We show examples of $\pi$ and its corresponding images. Figure C.5 shows the examples. We select three classes as examples and show their images and the learned $\pi$. Interestingly, for "black-footed ferret" images, we found that the context "my" has more than 0.5 weight. The actual images of black-footed ferrets are mostly composed of pet images, which makes sense that "my" prompt matches the images the most. Similarly, we observe that "front curtain" images are mostly low resolution due to the insufficient light in theaters, resulting in "a low resolution" or "a dark photo" becoming the most important prompts. Lastly, we see the missile images are mostly in the museums, resulting in "a close-up" prompt becoming the most important, but not significantly (0.1162) as much as the most contributing prompt of black-footed ferret (0.5130) and curtain (0.3921) images.

Using the $\pi$ with a few-shot setting ($K = 5$), we visualize the learned prompt weight. Figure C.6a shows the histogram of the maximum $\pi_c$ for each $c$; an uniform distribution will have 0.01, while larger $\max \pi_c$ denotes that specific prompts specifically selected for the class. In the figure, we can observe that the $\pi$ is generally larger than uniform. In addition, we plot the entropy of $\pi$ in Figure C.6b, where it shows a similar result.

## C.8 MORE APPLICATIONS OF THE LEARNED UNCERTAINTY

**Dataset filtering.** Below, we show the DataComp CLIP filtering small track (filtering 12.8B noisy web-crawled image-text pairs) by using our method and baselines provided by DataComp:

ProLIP uncertainty-aware features help better filtering compared to the other baselines. However, we note that ProLIP is not specifically designed for dataset filtering; proposing a new dataset filtering method using ProLIP will be an interesting future work, but not the scope of the current paper.

**Understanding image dataset.** ProLIP's image uncertainty is not the same as the "image uncertainty" of classification tasks. In classification tasks, an image has a high uncertainty if it can be matched to "multiple classes", while ProLIP assigns a high uncertainty if an image can be described in "multiple different and various captions" (Appendix A.1). For example, assume an image with a white background and a clear overall object shape. For classification, it has low uncertainty because there is no confounder to the classification. However, ProLIP will assign a high uncertainty for this.

From this, we can think two different scenarios: (1) When all images have a homogeneous background and only the quality of the image determines the classification performance (*e.g.*, MNIST), (2) When images are natural images and task is inherently multi-object classification, but the labels are single-labeled (*e.g.*, ImageNet as discussed by (Beyer et al., 2020; Yun et al., 2021))

Class name: Black-footed ferret

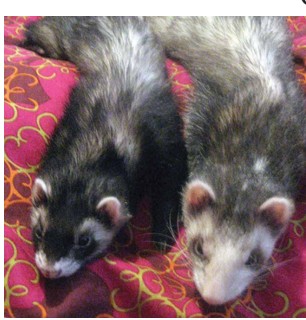 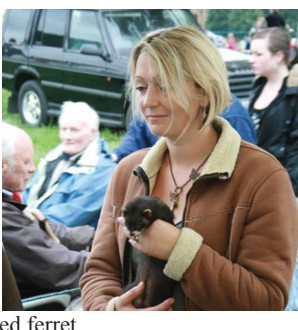 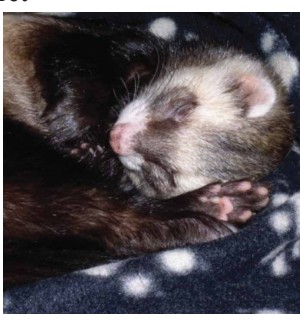

[0.5130] a photo of my black-footed ferret.
[0.0250] itap of my black-footed ferret.

...

[0.0056] a origami black-footed ferret
[0.0056] a sculpture of a black-footed ferret

Class name: Front curtain

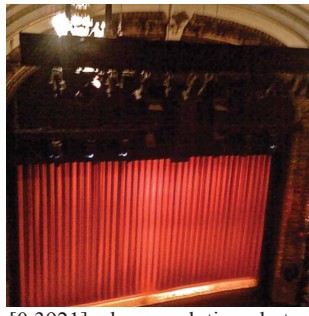 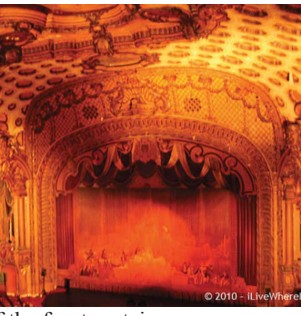 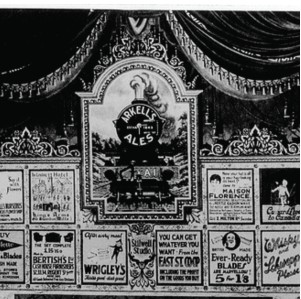

[0.3921] a low resolution photo of the front curtain.
[0.1102] a dark photo of a front curtain.

...

[0.0056] a drawing of a front curtain
[0.0056] a cartoon front curtain

Class name: Missile

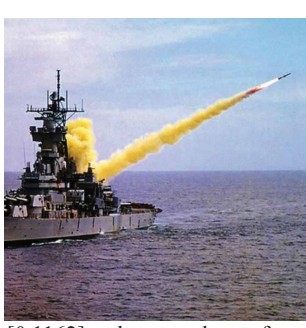 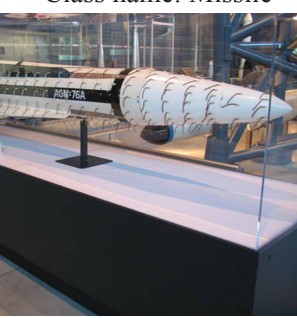 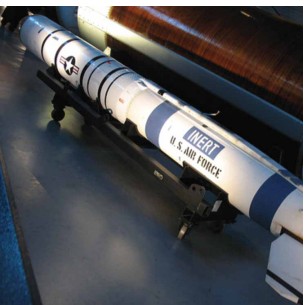

[0.1162] a close-up photo of a missile.
[0.0842] a black and white photo of the missile.

...

[0.0056] the plushie missile
[0.0056] a doodle of the missile

Figure C.5: **Visualization of learned $\pi$ by BPRW for each class.**

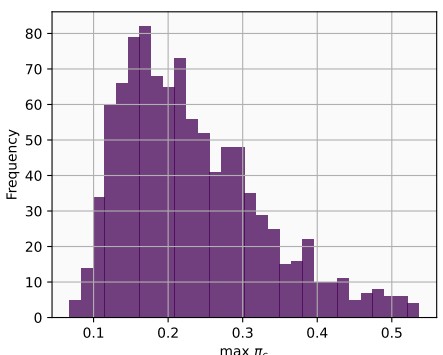

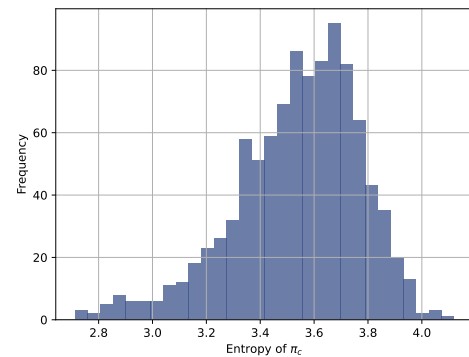

(a) **Histogram of** $\max \pi_c$**.** The uniform distribution will have 0.001 max $\pi$.

(b) **Histogram of entropy of** $\pi_c$**.** The uniform distribution will have 6.9078 entropy value.

Figure C.6: **Statistics of the learned** $\pi$ **by BPRW.** We use $\pi$ obtained from the few-shot setting with $K = 5$.

Table C.7: **Dataset filtering.** Results on DataComp small track (Gadre et al., 2024).

|  | Size | ImageNet | IN dist. | VTAB | Retrieval | Average |
|---|---|---|---|---|---|---|
| No filtering | 12.8M | 0.025 | 0.033 | 0.145 | 0.114 | 0.132 |
| Random subset (25%) | 3.2M | 0.022 | 0.032 | 0.130 | 0.099 | 0.126 |
| LAION-2B filtering | 1.3M | 0.031 | 0.040 | 0.136 | 0.092 | 0.133 |
| English (fasttext), cap length, and img size | 3M | 0.038 | 0.043 | 0.150 | 0.118 | 0.142 |
| Image-based & CLIP score (L/14 30%) | 1.4M | 0.039 | 0.045 | 0.162 | 0.094 | 0.144 |
| CLIP L14 (20%) | 2.6M | 0.042 | 0.051 | 0.165 | 0.100 | 0.151 |
| ProLIP distance (20%) | 2.3M | 0.042 | 0.047 | 0.167 | 0.117 | **0.154** |

As the first example, we choose MNIST. We observe that the learned image uncertainty and the MNIST accuracy show a strong negative correlation, (-0.98), namely, if an image is more uncertain then ProLIP tends to estimate a wrong label. As the second example, we choose ImageNet-1k, which shows a strong positive correlation (+0.98), *i.e.*, a certain image tends to be wrongly classified by ProLIP. This could be counterintuitive in "classification", but it is a correctly estimated value. For example, ImageNet contains various image distributions. Some images are thumbnail images with a white background (high uncertainty) and some images are in-the-wild images with complex background and objects (low uncertainty). In this case, a certain image (more complex images) will be more "difficult" images to be classified, which supports the positive correlation.

Overall, ProLIP's image uncertainty tendency can be used to understand an image dataset. Converting ProLIP's image uncertainty to image classification uncertainty would be an interesting topic.

**Uncertainty by image manipulation.** We additionally show the relationship between image manipulation and uncertainty. We evaluate ImageNet 1k zero-shot accuracy by applying a center occlusion. We applied 0% to 10% occlusion ratio (Table C.8). We also tested optimized noise by the PGD attack (Madry et al., 2018) with sampled ImageNet (Table C.9):

Table C.8: **Occlusion vs. image uncertainty.** Numbers are measured in the ImageNet validation set.

| Occlusion ratio | 0% | 2.5% | 5% | 7.5% | 10% |
|---|---|---|---|---|---|
| ImageNet-1k zero-shot | 74.6 | 74.1 | 73.8 | 73.5 | 73.2 |
| avg($\sigma_v$) | 0.0148 | 0.0149 | 0.0152 | 0.0153 | 0.0153 |

Here, we observe that the image uncertainty is increased by more severe manipulation. Note that as we discussed in "Understanding image dataset", converting ProLIP's image uncertainty to image qualification would be an interesting topic, but we remain this for future work.

Table C.9: **PGD vs. image uncertainty.** $PGD_k$ denotes the PGD attack with $k$ iterations.

|                                      | Clean  | $PGD_1$ | $PGD_5$ | $PGD_{10}$ | $PGD_{40}$ |
|--------------------------------------|--------|---------|---------|------------|------------|
| ImageNet-1k zero-shot (1000 images)  | 72.9   | 20.0    | 3.8     | 2.6        | 2.5        |
| $avg(\sigma_v)$                      | 0.0147 | 0.0149  | 0.0167  | 0.0175     | 0.0190     |

**ProLIP with long context text.** Although ProLIP can capture inherent ambiguity in vision-language tasks, ProLIP is limited to capturing long context text exceeding 64 tokens. To tackle the issue, we fine-tune the pre-trained ProLIP with long context texts, following LongCLIP (Zhang et al., 2024). In our primitive study, we observe that fine-tuning solely with the long text dataset can lead to a significant performance drop for general zero-shot tasks. More detail of our extension, LongProLIP (expanding the text context length from 64 to 256), is out of the scope of this paper; we refer to a technical report for LongProLIP for more interested readers (Chun & Yun, 2025).

## D DISCUSSION AND LIMITATION

As ProLIP is based on a normal distribution with diagonal covariance, ProLIP also shares two concerns discussed by Chun (2024): (1) using diagonal normal distribution would be insufficient to represent compared to the full covariance, (2) if we use different probability distributions (*e.g.*, von Mises–Fisher distribution or Laplacian distribution), the closed-form for PPCL and inclusion loss will not work anymore.

For the first concern, as already discussed by Chun (2024), the diagonal covariance would be insufficient if the dimensionality is too small (*e.g.*, less than ten). In this case, using the full covariance or mixture of Gaussian (MoG) will improve the representation power of the uncertainty. However, in practice, we use a very high dimensionality, *e.g.*, 768 for ViT-B/16, that can sufficiently capture complex semantics. One also can argue that using MoG is more sensible to capture many-to-many correspondences. However, if we have sufficiently large dimensionality, MoG will not be effective. Consider a probabilistic embedding with 2-MoG, namely $Z \sim \frac{1}{2}\mathcal{N}(\mu_1, \Sigma_1)$ with probability 0.5 and $Z \sim \frac{1}{2}\mathcal{N}(\mu_2, \Sigma_2)$ with probability 0.5. We can compute the expected CSD between two $Z$s (where $Z_1$ is parameterized by $\mu_1^1, \mu_2^1, \Sigma_1^1, \Sigma_2^1$ and $Z_2$ is parameterized by $\mu_1^2, \mu_2^2, \Sigma_1^2, \Sigma_2^2$) by computing

$$\frac{1}{4} \left\{ d(\mu_1^1, \mu_1^2, \Sigma_1^1, \Sigma_1^2) + d(\mu_1^1, \mu_2^2, \Sigma_1^1, \Sigma_2^2) + d(\mu_2^1, \mu_1^2, \Sigma_2^1, \Sigma_1^2) + d(\mu_2^1, \mu_2^2, \Sigma_2^1, \Sigma_2^2) \right\}, \quad \text{(D.1)}$$

where $d(\cdot)$ is CSD, *i.e.*, $d(\mu_1, \mu_2, \Sigma_1, \Sigma_2) = \|\mu_1 - \mu_2\|_2^2 + tr(\Sigma_1 + \Sigma_2)$. To simplicity, we omit $\frac{1}{4}$ for the remaining derivation. Now, consider two virtual unimodal Gaussian embeddings $W_1 \sim \mathcal{N}(\mu_1^1 \oplus \mu_1^1 \oplus \mu_2^1 \oplus \mu_2^1, \ \Sigma_1^1 \oplus \Sigma_1^1 \oplus \Sigma_2^1 \oplus \Sigma_2^1)$ and $W_2 \sim \mathcal{N}(\mu_1^2 \oplus \mu_2^2 \oplus \mu_1^2 \oplus \mu_2^2, \ \Sigma_1^2 \oplus \Sigma_2^2 \oplus \Sigma_1^2 \oplus \Sigma_2^2)$, where $\oplus$ denotes the "concatenation" operation, *i.e.*, $W_1$ and $W_2$ have four times larger dimensionality than $Z_1$ and $Z_2$. Interestingly, we can easily show that Equation (D.1) equals CSD between $W_1$ and $W_2$. Note that this derivation is invariant to the diagonal covariance, but also holds for the full covariance. In other words, using MoG is mathematically equivalent to using a larger dimensionality (as much as the square of the number of modes); therefore, if we have a sufficiently large dimensionality that can capture the ambiguity of the dataset, MoG is not a mandatory option.

The second concern can be raised when we use a different probability distribution. As discussed by Chun (2024), all objective functions and computations are distribution-free, but the derived closed-form solutions (*e.g.*, CSD, inclusion loss) will not work anymore if we use different distributions. One exception is MoG with equal mixing coefficients, but it equals simply using larger dimensionality. In practice, if we really need different distributions, we can use a Monte-Carlo approximation as Chun et al. (2021), which is known to be inefficient and inaccurate (Chun, 2024).

