# OpenReview forum: "Probabilistic Language-Image Pre-Training"
_ICLR.cc/2025/Conference — ICLR 2025 Poster_

### Official Review · Reviewer_8Wie · 2024-10-30

**Soundness:** 3
**Presentation:** 3
**Contribution:** 2
**Rating:** 6
**Confidence:** 4

**Summary:**

The paper introduces **Probabilistic Language-Image Pre-training** (ProLIP), an approach to vision-language models that integrates probabilistic modelling to capture the inherent uncertainty in image-text relationships. ProLIP departs from conventional deterministic embeddings by mapping inputs as random variables, estimating uncertainty using an "uncertainty token," and enforcing an inclusion loss to maintain distributional relationships between image-text pairs and between original and masked inputs.

**Strengths:**

-  This paper is well-written.
-  The research problem and motivation are well-defined, highlighting that CLIP-like models tend to oversimplify image-text alignment by assuming a one-to-one relationship.
-  ProLIP is claimed to be the first approach to pre-training probabilistic VLMs on a billion-scale image-text dataset, a feat that requires extensive computational resources.

**Weaknesses:**

-   The improvement over deterministic VLMs like CLIP is limited, with only a marginal gain of around 0.9% in ImageNet accuracy (Table 1) when models are exposed to the same number of samples.
-   While the paper introduces the uncertainty token for efficient uncertainty estimation, it lacks empirical evidence demonstrating the claimed efficiency.
-   Compared to PCME++, ProLIP incorporates an uncertainty token and an inclusion loss, scaling the training dataset to CLIP’s level, yet the observed improvement remains modest, limiting the paper’s overall contribution.
-  ProLIP requires the use of CSD to perform zero-shot prediction, which increases its complexity and makes it less straightforward than CLIP.

**Questions:**

- Although the authors claim ProLIP efficiently estimates uncertainty by incorporating an "uncertainty token" without adding parameters, this approach effectively increases the input length. Given that transformer complexity is quadratic relative to the input length, this addition likely increases the computation required for multi-head attention. How does the "uncertainty token" quantitatively compare to previous probabilistic VLMs in terms of training efficiency and inference speed? For example, the authors can report training time per epoch, inference latency on standard hardware, or FLOPs compared to baseline models.

- Is there any limitation discussed in the paper? (e.g. computational requirements)

---

> ### Author Response · Authors · 2024-11-18
>
> We thank the constructive feedback from the reviewer. In this comment, we address all the raised concerns by the reviewer; the revised paper is uploaded now. We list all the revised content in the “Revision overview” comment. If any concern remains, please let us know.
>
> **[W1] Marginal improvement over CLIP**
>
> First, we want to emphasize that +0.9% ImageNet zero-shot accuracy is not a marginal improvement. As shown in [A], it is almost close to a gap when using a larger backbone (e.g., ViT-H/14 (75.6%) $\Rightarrow$ ViT-g/14 (76.7%) when the number of seen samples is 13B). However, we also want to clarify that our purpose is not to achieve a significantly stronger ImageNet zero-shot performance, but to achieve a better understanding of the input uncertainty (i.e., aleatoric uncertainty).
>
> - [A] Cherti, Mehdi, et al. "Reproducible scaling laws for contrastive language-image learning." Proceedings of the IEEE/CVF Conference on Computer Vision and Pattern Recognition. 2023.
>
> Even if we consider the improvement to be marginal, we argue that learning probabilistic representations comparable to deterministic representations itself is sufficiently valuable. The main purpose of the probabilistic embedding space is to model the inherent uncertainty of inputs (i.e., aleatoric uncertainty), which is not able to be captured by deterministic embeddings regardless of the choice of the objective function. For example, consider four captions, A: “Person”, B: “A person is walking”, C: “A person is walking in the rain” and D: “A person is walking on sunshine”. Conceptually, C and D have completely different semantics hence they should be located far away. However, B can be matched to C and D, and A can be matched to B, C, and D. If we use a deterministic embedding space, we cannot find a stable solution to satisfy the condition (i.e., let d() be a distance function, then d(C, D) is large, but d(B, C) and d(B, D) are small -- similar condition can be derived by A). Furthermore, we can consider a more complicated scenario by considering a new caption E: “A person hugging a cat”; E should have large distances with B, C, and D but a small distance with A, while A should be close to B~E.
>
> On the other hand, as shown in Figure A.1 in the revised paper, a probabilistic embedding space can naturally capture the inherent uncertainty of inputs by representing the uncertainty by using variance; more uncertain inputs will be mapped to a random variable with a larger variance. Namely, if we need to make a positive relationship for (B, C) and (B, D), while keeping a negative relationship for (C, D), we first map C and D separately then map B to “cover” or “include” C and D by assigning a larger uncertainty value (i.e., variance). Our experimental results on pg8-10 in the main paper, as well as Appendix C show the advantage of proper uncertainty estimates.
>
> Our main contribution is to train the first PrVLM (which has the advantage of correctly representing aleatoric uncertainty) comparable to the deterministic baselines (e.g., CLIP, SigLIP). As already clarified in Section A.3, our ablation study (Table C.3) empirically show that the PCME++ loss fails to be converged when it is used in a stand-alone way without any deterministic loss (e.g., CLIP loss). On the other hand, PPCL loss converges well even without any additional loss function. Our first contribution is to enable a fully end-to-end scalable pre-training of PrVLM by proposing (1) new loss functions -- PPCL and inclusion loss, (2) new [UNC] token architecture, which is extremely efficient. Table C.3 and C.6 support that our new loss function and architecture outperform the previous PrVLM design choice (PCME++ loss + CLIP loss & multi-head uncertainty estimation module). Again, previous PrVLMs were not able to pre-train (i.e., they fine-tune the pre-trained CLIP models). PCME++ shows a small-scale pre-training result, but it still needs deterministic loss. Furthermore, as we mentioned before, even if we use additional CLIP loss, PPCL-only ProLIP performs better than PCME++ (Table C.3). It shows that ProLIP has a significant novel contribution to PrVLM.
>
> Our revised paper includes the related discussion in Section A.1.

---

> > ### Author Response · Authors · 2024-11-18
> >
> > **[W2/Q1] UNC token efficiency**
> >
> > In practice, Transformer with a relatively short context length (e.g., a few hundred) and small network capacity (e.g., ViT-B) does not show a quadratic complexity on the input length. Following the reviewer’s comment, we compare three different design choices (deterministic, UNC token, and multi-head uncertainty estimate module) on five different architectures (ViT-B/32, ViT-B/16, ViT-B/16 with 768 width, ViT-L/16, and ViT-L/14). Here, we only compare image encoders because image encoders always take the same image token length, which makes it easier to compare the impact of different design choices. We report (1) the number of input image tokens for each architecture (if we choose the “[UNC] token architecture”, it will be added by one), (2) the base parameters when we use deterministic, (3) inference time for 50k ImageNet validation images (lower is faster), and (4) additional parameters compared to the deterministic baseline.
> >
> > |  | ViT-B/32 | ViT-B/16 | ViT-B/16-768 | ViT-L/16 | ViT-L/14 |
> > |---|---|---|---|---|---|
> > | # img tokens | 49 | 196 | 196 | 196 | 256 |
> > | Base param | 151.7M | 150.0M | 197.3M | 428.5M | 428.4M |
> > | Deterministic | 75.18s | 76.68s | 78.53s | 137.77s | 176.95s |
> > | UNC (proposed) | 75.24s (+0.3M) | 76.84s (+0.3M) | 78.61s (+0.6M) | 137.99s (+0.6M) | 177.82s (+0.6M) |
> > | Multi-head | 75.75s (+20.7M) | 78.18s (+20.7M) | 80.08s (+28.9M) | 146.12s (+40.0M) | 191.68s (+40.0M) |
> >
> > We have three findings from the table.
> >
> > - Even if we increase the input token length, the actual inference speed is not quadratically slow -- ViT-B/32 and ViT-B/16 have the same Transformer capability but only input lengths are different (49 vs. 196), but their inference times are 75.18s and 76.68s, which is an almost neglectable change. If we choose a larger model (e.g., ViT-L), the difference becomes larger, e.g., 137.77s vs. 176.95s, but it is still not a quadratic order.
> > - [UNC] token adds almost neglectable parameters (0.3M for B and 0.6M for L) and inference time compared to the deterministic one.
> > - Multi-head architecture needs a large number of additional parameters (e.g., 20M for B and 40M for L) and shows slower inference time, especially for a larger network (e.g., 176.95s vs. 191.68s for ViT-L/14).
> >
> > Furthermore, in practice, multi-head architecture requires more memory than [UNC] token, which makes it difficult to use a large batch size and scale up to a larger backbone. On the other hand, [UNC] token only needs almost neglectable additional parameters, inference speed and memory size, which makes it easier to scale up.
> >
> > We add the related discussion in Appendix C.3.
> >
> > **[W3] Novelty compared to PCME++**
> >
> > Continuing from the response for W2, ProLIP has a huge contribution in terms of scaling up probabilistic representations. Namely, we cannot scale up PCME++ to CLIP level training dataset. First, as shown in the previous response, ProLIP is significantly more efficient than PCME++’s architecture (multi-head architecture) in terms of inference speed, parameter size, and memory. It makes ProLIP easier to be scaled up to a larger backbone while PCME++ suffers from the scalability issue. Namely, our architectural contribution is significant compared to PCME++ in terms of scaling up.
> >
> > Second, as shown in Table C.3 of the paper, a model will not be converged if we solely use PCME++ loss. We need additional deterministic loss, such as CLIP loss or SigLIP loss for a stable convergence of PCME++ loss. Even if we use additional deterministic loss, we can find that it performs worse than a model trained solely with ProLIP’s PPCL loss. It is because as we discussed in Sec A.2 of the paper, PCME++ loss uses binary cross entropy (BCE) loss, and PPCL loss is based on log-sigmoid loss. To compute BCE loss, we have to compute the squared L2 distance between two $\mu$s. If the difference between $\mu$s is extremely small, computing $\sum_{i=1}^D ((\mu_v^i)^2 - (\mu_t^i)^2)$ will suffer from a numerical error. On the other hand, PPCL loss directly uses the cosine similarity between $\mu$s, which is more numerically stable than the squared summation. Moreover, our inclusion loss is sufficiently novel which enforces a desired behavior; which was not able to PCME++. It supports that our loss function design is crucial to scaling up probabilistic representations.
> >
> > Finally, our main contribution is to train the first PrVLM comparable to the deterministic baselines (e.g., CLIP, SigLIP). Again, previous PrVLMs were not able to pre-train (i.e., they fine-tune the pre-trained CLIP models). PCME++ shows a small-scale pre-training result, but it still needs deterministic loss. Furthermore, as we mentioned before, even if we use additional CLIP loss, PPCL-only ProLIP performs better than PCME++ (Table C.3). It shows that ProLIP has a significant novel contribution to PrVLM.

---

> > > ### Author Response · Authors · 2024-11-18
> > >
> > > **[W4] CSD efficiency**
> > >
> > > CSD does not increase computational complexity. For given extracted $\mu_v, \Sigma_v, \mu_t, \Sigma_t$, the cosine similarity (by CLIP) is computed by $\mu_v^\top \mu_t$ (assuming that $\mu$s are l2-normalized). CSD is computed by $\mu_v^\top \mu_t - \frac{1}{2}tr(\Sigma_v + \Sigma_t)$. Here, $tr(\Sigma_v + \Sigma_t)$ is simply computed by $\sum_{i=1}^D ((\sigma_v^i)^2 + (\sigma_t^i)^2)$ (because we use diagonal covariances), which is a simple summation operation. Since our architecture estimates $\mu$ and $\Sigma$ with almost neglectable inference speed (see **[W2/Q1] UNC token efficiency** for more details), the only additional computation will be the summation of $\sigma^2$. Furthermore, in practice, if we can pre-extract the uncertainty scalar $u=\sum_{i=1}^D (\sigma^i)^2$ for each instance, we can simply compute CSD by computing $\mu_v^\top \mu_t - 0.5 * (u_v + u_t)$. Interestingly, in terms of “retrieval”, this score is invariant to the uncertainty of the query. For example, assume an image-to-text retrieval scenario. In this case, $u_v$ will be the same for the same image query. Therefore, we do not need an image uncertainty value, but only text uncertainty affects the retrieval ranking. Using this property, PCME++ (Chun 2024) showed that CSD is also possible to use an efficient approximated KNN (see Table C.7 of Chun 2024 for more details).
> > >
> > > We tried to measure the inference speed difference between cosine similarity and CSD, but we observed that there is no meaningful inference speed difference between the two methods. Even more, if one really needs cosine similarity-based retrieval (e.g., due to the limitation of the existing retrieval indexing system), ProLIP also supports cosine similarity retrieval.
> > >
> > > **[Q2] Limitation**
> > >
> > > ProLIP is a PrVLM, which shares a common limitation with other PrVLMs. For example, PCME++ (Chun 2024) discussed the possible limitation of assuming embeddings as normal distributions with diagonal covariance. This is highly related to W1 by Reviewer t6Gq. In addition, if we use a different probability distribution rather than Gaussian, the closed-form solutions derived in this paper will not work anymore and a Monte-Carlo approximation will be required. In the revised paper Section D, we added a discussion related to the limitation as the reviewer’s suggestion.

---

> > > > ### Author Response · Authors · 2024-11-20
> > > > **Any follow-up question?**
> > > >
> > > > Dear Reviewer 8Wie,
> > > >
> > > > We sincerely appreciate your efforts and time for the community. As we approach the close of the author-reviewer discussion period in one week, we wonder whether the reviewer is satisfied with our response. It will be pleasurable if the reviewer can give us the reviewer's thoughts on the current revision to give us an extra valuable chance to improve our paper. We summarized our revision in the "Revision summary" comment.
> > > >
> > > > Again, we thank the reviewer's valuable commitment and their help to strengthen our submission. We will address all the raised concerns by reviewers if there remain any.

---

> > > > > ### Comment · Reviewer_8Wie · 2024-11-24
> > > > >
> > > > > Thank you for your detailed response. I will update my score.

---

> > > > > > ### Author Response · Authors · 2024-11-26
> > > > > >
> > > > > > Thank you for your update! Could you kindly share the reasons behind your decision to raise the score to 6 (marginally above the acceptance threshold) rather than 8 (accept)? If there are specific aspects we can improve, we would be happy to update our paper accordingly.

---

> > > > > > > ### Author Response · Authors · 2024-11-29
> > > > > > >
> > > > > > > Dear Reviewer 8Wie,
> > > > > > >
> > > > > > > We would like to note that the reviewer-author discussion period will be closed in three days (2nd Dec). We would like to ask the reviewer to be willing to update their score in a more positive direction from borderline (6) to accept (8). If the reviewer thinks that there still exists room for improvement, please let us know. We will do our best to improve our submission.

---

> > > > > > > > ### Author Response · Authors · 2024-12-03
> > > > > > > >
> > > > > > > > Dear Reviewer 8Wie,
> > > > > > > >
> > > > > > > > This is a gentle reminder that **only 6 hours remain** for reviewers to respond to comments. We are still awaiting your response and would greatly appreciate it if you could provide your feedback. Additionally, we kindly ask if the reviewer would consider updating your score based on our clarification.

---

### Official Review · Reviewer_RZmV · 2024-11-02

**Soundness:** 3
**Presentation:** 2
**Contribution:** 3
**Rating:** 6
**Confidence:** 2

**Summary:**

The paper proposes a probabilistic VLM model that accounts for the many-to-many relationships between images and text. This model introduces an uncertainty token to estimate uncertainty without significantly increasing the parameter count. Additionally, the inclusion loss enhances interpretability by enforcing consistency between image-text pairs and between original and masked inputs.

**Strengths:**

The paper creatively applies probabilistic modeling to VLMs to better capture the many-to-many relationships between image and text pairs. It introduces a straightforward setup by adding an uncertainty token, effectively minimizing additional parameters. The paper provides a detailed analysis of the sources and levels of uncertainty in both image and text modalities, and explores various applications of this approach.

**Weaknesses:**

The paper primarily evaluates the model's performance on single-object scenarios, as seen with ImageNet in the main results (Table 1). While Table C.1 provides metrics for datasets with longer, more complex queries, such as Flickr, ProLIP shows a significant drop compared to CLIP. Including performance metrics and a detailed analysis of datasets with complex queries involving multiple objects and interactions in the main results would offer a more comprehensive evaluation of the model's capabilities.

**Questions:**

1. In Table C.1, ProLIP underperforms compared to CLIP on several datasets, such as Flickr. Could the authors provide a detailed analysis to explore the reasons behind this discrepancy?
2. The paper suggests that images with a single object are more uncertain than those with complex scenes. Intuitively, one might expect complex scenes to have higher uncertainty due to the variety of possible descriptions. Could the observed uncertainty levels be influenced by the choice of dataset used in the analysis?

---

> ### Author Response · Authors · 2024-11-18
>
> We thank the constructive feedback from the reviewer. In the following comment and the revised paper, we address all the raised concerns by the reviewer. The overview of the revised paper is clarified in the common comment (“Revision summary”). If the reviewer still has any concerns, please let us know.
>
> Before addressing each comment, we would like to clarify that while we addressed the concern raised by Reviewer RZmV, we found that our original experiments on 1.28B seen samples had a fair comparison issue; we made a mistake on data augmentation setting (we specified that we use 0.8-1.0 crop size, but we found that CLIP and ProLIP with 1.28B seen samples were trained with different crop sizes, e.g., 0.4-1.0 not 0.8-1.0). For a fair comparison, we re-evaluated ProLIP, CLIP and SigLIP on 1.28B seen samples and here is the result (also see revised paper Table 1):
>
> | Method | # seen samples | IN | IN dist. | VTAB | Retrieval | Average |
> |--------|-------|------|------|------|------|------|
> | CLIP   | 1.28B | 67.2 | 55.1 | 56.9 | 53.4 | 57.1 |
> | SigLIP | 1.28B | 67.4 | 55.4 | 55.7 | 53.4 | 56.7 |
> | ProLIP | 1.28B | 67.8 | 55.3 | 58.5 | 53.0 | 57.9 |
>
> **[W1/Q1] Subtask performance**
>
> With the revised experimental results, we found that the difference between CLIP, SigLIP and ProLIP in Flickr (as well as MSCOCO) is almost neglectable (also see revised paper Table C.1)
>
> | Method | # seen samples | Flickr | MSCOCO |
> |--------|-------|------|------|
> | CLIP   | 1.28B | 71.09 | 45.49 |
> | SigLIP | 1.28B | 71.76 | 45.47 |
> | ProLIP | 1.28B | 71.13 | 45.73 |
>
> From this observation, we believe that the assumption made by the reviewer (ProLIP performs worse in longer, more complex queries than CLIP) would not be true.
>
> When we compare CLIP and ProLIP task-wise, we can observe that the following datasets show “significant” performance difference (> 1.0%)
>
> | Dataset        | CLIP | ProLIP | diff |
> |------------|--------|--------|--------|
> | PatchCamelyon | 56.7 | 65.8 | 9.1 |
> | Camelyon17 | 56.0 | 64.3 | 8.3 |
> | KITTI Vehicle Distance | 30.8 | 39.1 | 8.3 |
> | CLEVR Counts | 18.0 | 22.4 | 4.4 |
> | SVHN | 59.7 | 62.6 | 2.9 |
> | Stanford Cars | 83.1 | 85.7 | 2.6 |
> | MNIST | 73.0 | 74.8 | 1.8 |
> | Oxford Flowers-102 | 69.7 | 71.2 | 1.5 |
> | GTSRB | 50.3 | 51.3 | 1.1 |
> | FGVC Aircraft | 20.8 | 21.8 | 1.0|
> | iWildCam | 11.2 | 10.1 | -1.0 |
> | Food-101 | 86.6 | 85.3 | -1.3 |
> | WinoGAViL | 43.5 | 42.1 | -1.3 |
> | Pascal VOC 2007 | 81.4 | 80.0 | -1.4 |
> | FMoW | 10.6 | 9.0 | -1.6 |
> | EuroSAT | 45.1 | 40.2 | -4.9 |
>
> Overall, ProLIP performs better than CLIP more than +1.0pp in 10 over 38 datasets, while only performing worse in 6 over 38 datasets. Furthermore, ProLIP outperforms CLIP in several datasets, such as PatchCamelyon (very difficult medical images indicating the presence of metastatic tissue), Camelyon17 (similarly, medical images), KITTI Vehicle Distance (real road images where the task is to classify how a car is close in four levels) and CLEVR Counts (counting objects in the image among 3~10). It supports that ProLIP has a higher capability to understand complex scenes (e.g., road and vehicle images), multiple objects (counting task), or images (medical images) compared to deterministic embeddings. Meanwhile, ProLIP performs worse than CLIP in EuroSAT. We presume that it is because the EuroSAT images are satellite images that look highly similar to each other and “uncertain” representations compared to other benchmarks. However, when we sufficiently train probabilistic representations, the EuroSAT ZS accuracy becomes 60.9 from 40.2 (+20.7pp), while the average enhancement is only +5.4pp.
>
> In addition, we would like to emphasize that we did not selectively evaluate the mode performance in a specific scenario. We follow the evaluation protocol by DataComp [A], which evaluates models on 38 different zero-shot tasks, including classification and retrieval. This contains vast scenarios, such as image-text cross-modal retrieval (MS-COCO, Flickr) clean single object (CIFAR, Flowers, Cars, …), a somewhat noisy single object (ImageNet, ImageNet variants, ObjectNet, …) -- although we evaluate on a single label, their images are not a single object, but actually multi objects [B, C] --, counting tasks (CLEVR Counts), OCR tasks (Rendered SST2), distance estimation tasks (CLEVR distance, KITTI Vehicle Distance), texture identification tasks (Describable Textures), medical tasks (PatchCamelyon, Camelyon17), satellite images (EuroSAT) and in-the-wild image classification tasks (iWildCAM, GTSRB, …).
>
> - [A] Gadre, Samir Yitzhak, et al. "Datacomp: In search of the next generation of multimodal datasets." NeurIPS 2024.
> - [B] Beyer, Lucas, et al. "Are we done with imagenet?." arXiv preprint arXiv:2006.07159 (2020).
> - [C] Yun, Sangdoo, et al. "Re-labeling imagenet: from single to multi-labels, from global to localized labels." CVPR 2021.

---

> > ### Author Response · Authors · 2024-11-18
> >
> > **[Q2] Image uncertainty**
> >
> > An ideal PrVLM should capture three potential input uncertainties: (1) uncertainty from the text modality, (2) uncertainty from the image modality, and (3) uncertainty from the text-image cross-modality. Uni-modal uncertainty is straightforward; if an input has more detailed information (e.g., describing more detailed information in text, or capturing a very detailed and complex scene by photography), then it will have smaller uncertainty, otherwise, it will have larger uncertainty (e.g., providing very high-level caption, such as “person”, or only a part object with white background is taken by picture).
> >
> > The cross-modal uncertainty should capture “how many possible instances can be matched to this input?”. We can think this in two different viewpoints: text-to-image and image-to-text. The text-to-image relationship is straightforward. If we have a caption “photo”, then it will be matched to all photographs, and if we have a caption with a very detailed description (e.g., the full description of the hotel room), then it will be only matched to specific images. Image-to-text relationships are often determined by the dataset. For example, if we have a caption dataset exactly describing which objects are in the image, then we can think that an image with fewer objects will have larger uncertainty. However, if we consider a caption dataset where an image has multiple captions each caption focuses on completely different objects, then an image with more objects will have larger uncertainty. In other words, unlike text uncertainty originating from a text-to-image relationship, image uncertainty originating from an image-to-text relationship is highly affected by the property of the training dataset.
> >
> > In practice, because our datasets have a mixed property and their captions are somewhat noisy, our image uncertainty will have a mixed property, namely, unlike text uncertainty, there could be no strong relationship between the absolute uncertainty value and the number of objects (or complexity of images). However, empirically, we have more captions that exactly describe the scene (because of the dataset filtering strategy), therefore, a more simple image tends to have larger uncertainty and a more complex image tends to have smaller uncertainty.
> >
> > Furthermore, since there is no “uni-modal” uncertainty supervision, a PrVLM trained only with image-text relationships will have no guarantee to represent proper image uni-modal uncertainty. To enforce a proper uni-modal image uncertainty, we first propose the uni-modal uncertainty supervision by proposing the inclusion loss, namely, the original image embedding should be included by an image embedding from the masked image.
> >
> > We also clarify that the dataset used for the analysis is a very large-scale dataset with 3.5M image-text pairs (which follows the training distribution of DataComp1B). If we choose a more restricted dataset with selective images or captions, we may have a different observation, but we don’t think it truly represents the embedding space learned by ProLIP.
> >
> > Our revised paper includes the related discussion in Section A.1.

---

> > > ### Author Response · Authors · 2024-11-20
> > > **Any follow-up question?**
> > >
> > > Dear Reviewer RZmV,
> > >
> > > We sincerely appreciate your efforts and time for the community. As we approach the close of the author-reviewer discussion period in one week, we wonder whether the reviewer is satisfied with our response. It will be pleasurable if the reviewer can give us the reviewer's thoughts on the current revision to give us an extra valuable chance to improve our paper. We summarized our revision in the "Revision summary" comment.
> > >
> > > Again, we thank the reviewer's valuable commitment and their help to strengthen our submission. We will address all the raised concerns by reviewers if there remain any.

---

> > > > ### Comment · Reviewer_RZmV · 2024-11-22
> > > >
> > > > Thank you for the detailed response and clarification. Most of my questions are now resolved, and I’ve updated my score accordingly.

---

> > > > > ### Author Response · Authors · 2024-11-22
> > > > >
> > > > > Thank you for your update! Could you kindly share the reasons behind your decision to raise the score to 6 (marginally above the acceptance threshold) rather than 8 (accept)? If there are specific aspects we can improve, we would be happy to update our paper accordingly.

---

> > > > > > ### Author Response · Authors · 2024-11-29
> > > > > >
> > > > > > Dear Reviewer RZmV,
> > > > > >
> > > > > > We would like to note that the reviewer-author discussion period will be closed in three days (2nd Dec). We would like to ask the reviewer to be willing to update their score in a more positive direction from borderline (6) to accept (8). If the reviewer thinks that there still exists room for improvement, please let us know. We will do our best to improve our submission.

---

> > > > > > > ### Author Response · Authors · 2024-12-03
> > > > > > >
> > > > > > > Dear Reviewer RZmV,
> > > > > > >
> > > > > > > This is a gentle reminder that **only 6 hours remain** for reviewers to respond to comments. We are still awaiting your response and would greatly appreciate it if you could provide your feedback. Additionally, we kindly ask if the reviewer would consider updating your score based on our clarification.

---

### Official Review · Reviewer_t6Gq · 2024-11-04

**Soundness:** 3
**Presentation:** 2
**Contribution:** 3
**Rating:** 6
**Confidence:** 3

**Summary:**

The paper works on a probabilistic CLIP that can model the uncertainty of the data. The model measures the uncertainty with a multi-dimensional Gaussian distribution, and it is trained with proposed novel inclusion loss. The paper also simplified the sampled distance loss from previous paper. Given that, the paper first shows some analysis that the estimated uncertainty aligns with several intuitions. Lastly, it shows two applications of image traversals (find a more concrete caption for the original caption iteratively) and prompt enhancement for the zero-shot image classification task.

**Strengths:**

1. The paper constructs a probabilistic VLM that can capture of uncertainty from the multimodal dataset.

2. The paper proposes the inclusion loss and also provide intuitive understanding of it.

3. The paper contains interesting analysis of the uncertainty.

4. The paper proposes to utilize the uncertainty to conduct prompt rewriting.

**Weaknesses:**

1. The intuition behinds the paper is that multimodal paired data can be actually many-to-many mapping. I think that this intuition would lead to a multi-modes representation intuitively, as different text can be assigned to a single image, e.g., "A dog is walking" and "A man is looking at the building". However, the modeling in this paper is a Gaussian distribution with a single mean that can not conduct good mode coverage.

1. (Minor) The basic behinds contrastive learning is that the positive pair is sampled from the joint distribution $P(x, y)$ while the negative pairs are sampled independently $P(x)P(y)$. Given that, contrastive learning does not force a 1-1 mapping from its maths intuition. Practically, the behavior might be different from the above as model learning might force 1-1. The motivation in Abs/Intro and Sec 2.1 might need a revision. The current claim is a little bit stronger to me and this might be fixed by writing or explanations. Also, in Sec 2.2, it claims that "VL tasks suffer from uncertainty as shown in Sec 2.1". This is a little bit beyond the context of Sec 2.1, as Sec 2.1 does not have strong VL task experimental evidence. I treated it as over-claim now but I think that it can be solved by writing revision.

1. The two applications (image caption traversals and prompt enhancement) of the estimated uncertainty are not representative, and it would be better if more applications (more concretely, proof of concepts) are presented. For example, the data filtering, error estimation, etc.

**Questions:**

1. Please specify the context why $\mu_1$ and $\mu_2$ are norm-1 vectors (i.e., because CLIP normalizes them during training?)? I think that the original CLIP only normalizes when calculating the contrastive loss, maybe there is a re-parameterization here. Also please include them in text which would be friendly to reader.

2. I wonder how the estimated variance from the model is used in the classification task?

2. I appreciate the comparisons to KL divergence in Sec 3.3. The paper only measures KL(p1||p2) which has a different form to the proposed inclusion metric. How about adding a comparison to metric KL(p1 || p2) - KL(p2 || p1)?

3. The inclusion loss is defined as $L(Z1, Z2)$ = $Z1 \in Z2$. Thus, in Line 259 and Line 264, should that be $L(t, v)$ and $L(partial, all)$ instead the current $L(v, t)$ and $L(all, partial)$?

4. In Fig 5, it shows that the estimated variance of image is smaller than text. The loss defined in Eqn (6) is unsymmetrical for image and text (because of the hyperparemeters and the text VIB loss). Would that be possible that the smaller image variance is from this loss design?

4. Please specify the exact formulation (i.e., implementation) of VIB in main text or in appendix.

4. Please specify how the uncertainty information is utilized when conducting the image traversals task.

5. (An open question; no concrete answer required) What would be the golden uncertainty of a particular data in the PrVLM? I.e., the best distribution that a model should converge to?

---

> ### Author Response · Authors · 2024-11-18
>
> We thank the reviewer for their positive feedback and constructive comments. In the following comment and the revised paper, we address all the raised concerns by the reviewer. The overview of the revised paper is clarified in the common comment (“Revision summary”). If the reviewer still has any concerns, please let us know.
>
> **[Q8] What is the desired probabilistic embedding space?**
>
> Before addressing the other concerns, we would like to first clarify the purpose of our probabilistic embedding space and its desired embedding space. The main purpose of the probabilistic embedding space is to model the inherent uncertainty of inputs (i.e., aleatoric uncertainty), which is not able to be captured by deterministic embeddings regardless of the choice of the objective function. For example, consider four captions, A: “Person”, B: “A person is walking”, C: “A person is walking in the rain” and D: “A person is walking on sunshine”. Conceptually, C and D have completely different semantics hence they should be located far away. However, B can be matched to C and D, and A can be matched to B, C, and D. If we use a deterministic embedding space, we cannot find a stable solution to satisfy the condition (i.e., let d() be a distance function, then d(C, D) is large, but d(B, C) and d(B, D) are small -- similar condition can be derived by A). Furthermore, we can consider a more complicated scenario by considering a new caption E: “A person hugging a cat”; E should have large distances with B, C, and D but a small distance with A, while A should be close to B~E.
>
> On the other hand, as shown in Figure A.1 in the revised paper, a probabilistic embedding space can naturally capture the inherent uncertainty of inputs by representing the uncertainty by using variance; more uncertain inputs will be mapped to a random variable with a larger variance. Namely, if we need to make a positive relationship for (B, C) and (B, D), while keeping a negative relationship for (C, D), we first map C and D separately then map B to “cover” or “include” C and D by assigning a larger uncertainty value (i.e., variance).
>
> An ideal PrVLM should capture three potential input uncertainties: (1) uncertainty from the text modality, (2) uncertainty from the image modality, and (3) uncertainty from the text-image cross-modality. Uni-modal uncertainty is straightforward; if an input has more detailed information (e.g., describing more detailed information in text, or capturing a very detailed and complex scene by photography), then it will have smaller uncertainty, otherwise, it will have larger uncertainty (e.g., providing very high-level caption, such as “person”, or only a part object with white background is taken by picture).
>
> The cross-modal uncertainty should capture “how many possible instances can be matched to this input?”. We can think this in two different viewpoints: text-to-image and image-to-text. The text-to-image relationship is straightforward. If we have a caption “photo”, then it will be matched to all photographs, and if we have a caption with a very detailed description (e.g., the full description of the hotel room), then it will be only matched to specific images. Image-to-text relationships are often determined by the dataset. For example, if we have a caption dataset exactly describing which objects are in the image, then we can think that an image with fewer objects will have larger uncertainty. However, if we consider a caption dataset where an image has multiple captions each caption focuses on completely different objects, then an image with more objects will have larger uncertainty. In other words, unlike text uncertainty originating from a text-to-image relationship, image uncertainty originating from an image-to-text relationship is highly affected by the property of the training dataset.
>
> In practice, because our datasets have a mixed property and their captions are somewhat noisy, our image uncertainty will have a mixed property, namely, unlike text uncertainty, there could be no strong relationship between the absolute uncertainty value and the number of objects (or complexity of images). However, empirically, we have more captions that exactly describe the scene (because of the dataset filtering strategy), therefore, a more simple image tends to have larger uncertainty and a more complex image tends to have smaller uncertainty.
>
> Our revised paper includes the related discussion in Section A.1.

---

> > ### Author Response · Authors · 2024-11-18
> >
> > **[W1] Mixture of Gaussian**
> >
> > Thanks for the interesting question. Mixture of Gaussian (MoG) is an intuitive approach to tackle the multi-mode problem. However, if we have a sufficiently large dimensionality, MoG is not a mandatory option. Intuitively, we can break D-dimensional unimodal Gaussian probabilistic embedding space into multiple subspaces with smaller dimensions (e.g., D/2-dimensional space) and consider each space as a different Gaussian embedding space. More formally and rigorously, in our framework (using CSD as the probabilistic distance), using k-MoG (MoG with k number of Gaussians) is equivalent to using k^2 D-dimensional embedding space.
> >
> > Consider a probabilistic embedding with 2-MoG, namely $Z \sim \frac{1}{2}\mathcal N(\mu_1, \Sigma_1)$ with probability 0.5 and $Z \sim \frac{1}{2}\mathcal N(\mu_2, \Sigma_2)$ with probability 0.5. We can compute the expected CSD between two $Z$s (where $Z_1$ is parameterized by $\mu^1_1, \mu^1_2, \Sigma^1_1, \Sigma^1_2$ and $Z_2$ is parameterized by $\mu^2_1, \mu^2_2, \Sigma^2_1, \Sigma^2_2$) by computing
> >
> > $$\frac{1}{4} \left[ d(\mu^1_1, \mu^2_1, \Sigma^1_1, \Sigma^2_1) + d(\mu^1_1, \mu^2_2, \Sigma^1_1, \Sigma^2_2) + d(\mu^1_2, \mu^2_1, \Sigma^1_2, \Sigma^2_1) + d(\mu^1_2, \mu^2_2, \Sigma^1_2, \Sigma^2_2) \right],$$
> >
> > where $d(\cdot)$ is CSD, i.e, $d(\mu_1, \mu_2, \Sigma_1, \Sigma_2) = \| \mu_1 - \mu_2 \|_2^2 + tr(\Sigma_1 + \Sigma_2)$. To simplicity, we omit $\frac{1}{4}$ for the remaining derivation. Now, consider two virtual unimodal Gaussian embeddings
> >
> > $$W_1 \sim \mathcal N(\mu^1_1 \oplus \mu^1_1 \oplus \mu^1_2 \oplus \mu^1_2, \Sigma^1_1 \oplus \Sigma^1_1 \oplus \Sigma^1_2 \oplus \Sigma^1_2)$$
> > $$W_2 \sim \mathcal N(\mu^2_1 \oplus \mu^2_2 \oplus \mu^2_1 \oplus \mu^2_2, \Sigma^2_1 \oplus \Sigma^2_2 \oplus \Sigma^2_1 \oplus \Sigma^2_2),$$
> >
> > where $\oplus$ denotes the “concatenation” operation, i.e, $W_1$ and $W_2$ have four times larger dimensionality than $Z_1$ and $Z_2$.
> >
> > Interestingly, we can easily show that the above equation equals CSD between $W_1$ and $W_2$. Note that this derivation is invariant to the diagonal covariance, but also holds for the full covariance. In other words, using MoG is mathematically equivalent to using a larger dimensionality (as much as the square of the number of modes); therefore, if we have a sufficiently large dimensionality that can capture the ambiguity of the dataset, MoG is not a mandatory option.
> >
> > Our revised paper includes the related discussion in Section D.
> >
> > **[W2] Contrastive loss can solve many-to-many?**
> >
> > From the response for **[Q8]**, we clarify that our purpose is to capture the inherent input uncertainty that cannot be captured by a deterministic embedding space. Since this is the fundamental limitation of a deterministic space, this problem is agnostic to the choice of objective function. We partially agree that contrastive learning less penalizes plausible matching than other loss functions, such as triplet loss with hard negative mining, but it still shares the limitation of deterministic embeddings.
> >
> > We cite Appendix A.1 in Section 2.1 and 2.2. If the reviewer thinks the current version is still insufficient, please let us know.
> >
> > **[W3] Other downstream tasks**
> >
> > We do not think our applications are not representative as the reviewer's comment, but we also agree that it would be better to show more applications of the learned uncertainty. Here, we show 3 applications.
> >
> > **Dataset filtering**
> >
> > Below, we show the DataComp CLIP filtering small track (filtering 12.8B noisy web-crawled image-text pairs) by using our method and baselines provided by DataComp [A]
> >
> > - [A] Gadre, Samir Yitzhak, et al. "Datacomp: In search of the next generation of multimodal datasets." NeurIPS 2024.
> >
> > |  | Dataset size | ImageNet | ImageNet dist. shifts | VTAB | Retrieval | Average |
> > |---|---|---|---|---|---|---|
> > | No filtering | 12.8M | 0.025 | 0.033 | 0.145 | 0.114 | 0.132 |
> > | Random subset (25%) | 3.2M | 0.022 | 0.032 | 0.130 | 0.099 | 0.126 |
> > | LAION-2B filtering | 1.3M | 0.031 | 0.040 | 0.136 | 0.092 | 0.133 |
> > | English (fasttext), caption length, and image size | 3M | 0.038 | 0.043 | 0.150 | 0.118 | 0.142 |
> > | Image-based & CLIP score (L/14 30%) | 1.4M | 0.039 | 0.045 | 0.162 | 0.094 | 0.144 |
> > | CLIP L14 (20%) | 2.6M | 0.042 | 0.051 | 0.165 | 0.100 | 0.151 |
> > | ProLIP B16 (20%) | 2.3M | 0.042 | 0.047 | 0.167 | 0.117 | **0.154** |
> >
> > Here, we can observe that by using ProLIP uncertainty-aware features, we can achieve a better filtering performance compared to the other baselines. However, we would like to note that ProLIP is not specifically designed for dataset filtering; proposing a new dataset filtering method using ProLIP will be an interesting future work, but not the scope of the current paper.

---

> > > ### Author Response · Authors · 2024-11-18
> > >
> > > **Understanding image dataset**
> > >
> > > First, we emphasize that ProLIP’s image uncertainty is not the same as the “image uncertainty” of classification tasks. In classification tasks, an image has a high uncertainty if it can be matched to “multiple classes”, while the ProLIP case assigns a high uncertainty if an image can be described in “multiple different and various captions”. However, ProLIP has a different mechanism with classification uncertainty.
> > >
> > > An ideal PrVLM should capture three potential input uncertainties: (1) uncertainty from the text modality, (2) uncertainty from the image modality, and (3) uncertainty from the text-image cross-modality. Uni-modal uncertainty is straightforward; if an input has more detailed information (e.g., describing more detailed information in text, or capturing a very detailed and complex scene by photography), then it will have smaller uncertainty, otherwise, it will have larger uncertainty (e.g., providing very high-level caption, such as “person”, or only a part object with white background is taken by picture).
> > >
> > > The cross-modal uncertainty should capture “how many possible instances can be matched to this input?”. We can think this in two different viewpoints: text-to-image and image-to-text. The text-to-image relationship is straightforward. If we have a caption “photo”, then it will be matched to all photographs, and if we have a caption with a very detailed description (e.g., the full description of the hotel room), then it will be only matched to specific images. Image-to-text relationships are often determined by the dataset. For example, if we have a caption dataset exactly describing which objects are in the image, then we can think that an image with fewer objects will have larger uncertainty. However, if we consider a caption dataset where an image has multiple captions each caption focuses on completely different objects, then an image with more objects will have larger uncertainty. In other words, unlike text uncertainty originating from a text-to-image relationship, image uncertainty originating from an image-to-text relationship is highly affected by the property of the training dataset.
> > >
> > > In practice, because our datasets have a mixed property and their captions are somewhat noisy, our image uncertainty will have a mixed property, namely, unlike text uncertainty, there could be no strong relationship between the absolute uncertainty value and the number of objects (or complexity of images). However, empirically, we have more captions that exactly describe the scene (because of the dataset filtering strategy), therefore, a more simple image tends to have larger uncertainty and a more complex image tends to have smaller uncertainty.
> > >
> > > For example, assume we have an image with a white background and a very clear overall object shape. In terms of classification, it will have low uncertainty because there is no confounder to the classification. However, ProLIP will assign a high uncertainty for this case.
> > >
> > > From this, we can divide the image classification tasks into two different scenarios: (1) When all images have homogeneous backgrounds and only the quality of the image determines the classification performance, (2) When images are natural images and the task is inherently multi-object classification, but the labels are single-labeled (e.g., ImageNet [B, C])
> > >
> > > - [B] Beyer, Lucas, et al. "Are we done with imagenet?." arXiv preprint arXiv:2006.07159 (2020).
> > > - [C] Yun, Sangdoo, et al. "Re-labeling imagenet: from single to multi-labels, from global to localized labels." CVPR 2021.
> > >
> > > As the first example, we choose MNIST. Here, we observe that the learned image uncertainty and the MNIST accuracy show a strong negative correlation, (-0.98), namely, if an image is more uncertain then ProLIP tends to estimate a wrong label. It aligns with our intuition.
> > >
> > > As the second example, we choose ImageNet-1k, which shows a strong positive correlation (+0.98), i.e., a certain image tends to be wrongly classified by ProLIP. This could be counterintuitive in terms of “classification task”, but actually it is a correctly estimated value. For example, ImageNet contains various image distributions. Some images are thumbnail images with white backgrounds (high uncertainty) and some images are in-the-wild images with complex backgrounds and multiple objects (low uncertainty). In this case, a certain image (more complex image) will be a more “difficult” image.
> > >
> > > Overall, ProLIP’s image uncertainty tendency can be used for understanding the properties of the given dataset. Converting ProLIP’s image uncertainty to image classification uncertainty would be an interesting topic, but we remain this for future work.

---

> > > > ### Author Response · Authors · 2024-11-18
> > > >
> > > > **Uncertainty by image manipulation**
> > > >
> > > > We additionally show the relationship between image manipulation and uncertainty. We evaluate ImageNet 1k zero-shot accuracy by applying a center occlusion. We applied 0% to 10% occlusion ratio, and got the following results:
> > > >
> > > > | Occlusion ratio | ImageNet ZS | avg($\sigma_v$) |
> > > > |---|---|---|
> > > > | 0% | 74.6 | 0.0148 |
> > > > | 2.5% | 74.1 | 0.0149 |
> > > > | 5% | 73.8 | 0.0152 |
> > > > | 7.5% | 73.5 | 0.0153 |
> > > > | 10% | 73.2 | 0.0153 |
> > > >
> > > > We also tested optimized noise by the PGD attack [D] with sampled ImageNet (1000 images):
> > > >
> > > > | Attack | ImageNet ZS (1000 images) | avg($\sigma_v$) |
> > > > |---|---|---|
> > > > | - | 72.9 | 0.0147 |
> > > > | PGD(iter=1) | 20.0 | 0.0149 |
> > > > | PGD(iter=5) | 3.8 | 0.0167 |
> > > > | PGD(iter=10) | 2.6 | 0.0175 |
> > > > | PGD(iter=40) | 2.5 | 0.0190 |
> > > >
> > > > - [D] Mądry, Aleksander, et al. "Towards deep learning models resistant to adversarial attacks." ICLR 2018
> > > >
> > > > Here, we observe that the image uncertainty is increased by more severe manipulation. Note that as we discussed in the previous application (**Understanding image dataset**), converting ProLIP’s image uncertainty to image qualification would be an interesting topic, but we remain this for future work.
> > > >
> > > > **[Q1] normalized means**
> > > >
> > > > CLIP uses normalized vectors for both training and inference to compute cosine similarity. ProLIP also uses normalized means due to the computational advantage of normalized means. For example, normalized vectors can compute pairwise distance very efficiently by using cosine distance. This is a rule-of-thumb for retrieval models (or metric learning methods) not only for CLIP, but also for Recommender systems (e.g., Matrix Factorization), document embeddings (e.g., ColBERT) and image retrieval.
> > > >
> > > > Also, conceptually, we can estimate the Gaussian distribution by using the estimated mean and covariance (i.e., $\mathcal N(\mu, \Sigma)$). In practice, because we use the closed-form solution for calculating distance (closed-form sampled distance) and loss functions (inclusion loss, PPCL), no sampling based on the re-parameterization trick is required. We specified this in Appendix A.3
> > > >
> > > > **[Q2] Classification with variance**
> > > >
> > > > Conceptually, zero-shot classification is an image-to-text retrieval, where the database texts are the class names. CLIP also uses this approach by using cosine similarity as the distance. ProLIP uses the same concept, but we use CSD as the distance. Since CSD is computed by $\mu$ and $\Sigma$: $\mu_v^\top \mu_t - \frac{1}{2}tr(\Sigma_v + \Sigma_t)$, the estimated variances are used for computing the distance between image and text. We clarify this in Section B.2.
> > > >
> > > > **[Q3] KL(p1 || p2) - KL(p2 || p1)**
> > > >
> > > > Thanks for your suggestion. Let the proposed metric be named “DKL”. We add the related discussion in Section A.6 of the revised paper. As shown in Fig A.2, the most significant problem with DKL is that it cannot correctly represent the “inclusion” relationship. For example, consider two random variables $Z_1$ and $Z_2$ where they do not include each other. In this case, although $Z_1$ and $Z_2$ do not include each other, because $DKL(Z_1, Z_2) = -DKL(Z_1, Z_2)$, one of $DKL(Z_1, Z_2)$ or $DKL(Z_2, Z_1)$ will become positive, while the other will become negative. Namely, DKL cannot correctly capture the inclusion relationship. On the other hand, in the same scenario, our inclusion test always returns negative values as shown in Fig A.2, which correctly represents the inclusion relationship.
> > > >
> > > > **[Q4] Inclusion loss misunderstanding**
> > > >
> > > > In the answer for **[Q8]**, we clarified that an embedding with higher uncertainty will have a larger variance. Here, a “partial” instance will have higher uncertainty than “all” because the hidden part of the instance can be potentially matched to various inputs. For example, partial information “A person is walking [MASK] [MASK]” can be either “A person is walking in rain” or “A person is walking on sunshine”, therefore “all” should be included by “partial”. Similarly, we will clarify the relationship between $Z_t$ and $Z_v$ in the following response for **[Q5]**.

---

> > > > > ### Author Response · Authors · 2024-11-18
> > > > >
> > > > > **[Q5] $tr(\Sigma_v)$ vs. $tr(\Sigma_t)$**
> > > > >
> > > > > We first clarify that Eqn (6) is asymmetric due to the last term, $\mathcal L_\text{inclusion} (Z_v, Z_t)$. The other terms are symmetric for image and text. VIB loss is applied to both image and text and the hyperparameters are the same for each modality.
> > > > >
> > > > > The smaller image variance can indeed be from the asymmetric loss design due to $\mathcal L_\text{inclusion} (Z_v, Z_t)$. It enforces that image distribution is included in text distribution; which naturally implies images have smaller variance than texts. We use this asymmetric formulation from our intuition and previous observations from the other papers.
> > > > >
> > > > > First, our intuition is that while an image exhaustively captures visual information by a photographic sensor, a text description is a conscious product of the dominant concepts in the image. Namely, a text is more selective and vague, which naturally implies more uncertainty. The previous PrVLM works also assume the same scenario [E, F] Second, this assumption (text is more vague than image) is a widely used assumption in VL training. For example, Meru [G] assumes that a text “entails” an image. The following works on hyperbolic embeddings also share the same assumption [H, I]. Our work aligns with these works by assuming that text is more uncertain than image. This is clarified in L260-261 in the main paper.
> > > > >
> > > > > - [E] Chun, Sanghyuk, et al. "Probabilistic embeddings for cross-modal retrieval." CVPR 2021
> > > > > - [F] Chun, Sanghyuk. "Improved probabilistic image-text representations." ICLR 2024
> > > > > - [G] Desai, Karan, et al. "Hyperbolic image-text representations." International Conference on Machine Learning. ICML, 2023
> > > > > - [H] Alper, Morris, and Hadar Averbuch-Elor. "Emergent visual-semantic hierarchies in image-text representations.", CVPR 2024
> > > > > - [I] Kim, Wonjae, et al. "HYPE: Hyperbolic Entailment Filtering for Underspecified Images and Texts." ECCV 2024
> > > > >
> > > > > **[Q6] VIB formulation**
> > > > >
> > > > > Sorry. We clarified it in Section A.7 of the revised paper. Also, we will release the code implementation publicly which includes the implementation of VIB loss.
> > > > >
> > > > > **[Q7] Image traversal with uncertainty**
> > > > >
> > > > > The traversal task needs two information: the closed text embedding and [Root] embedding for the given image. Once the closed text embedding and [Root] embedding are chosen, we interpolate them with 50 equally spaced steps and find the closest text from the database for each interpolated caption.
> > > > >
> > > > > Uncertainty information is used for the traversal task in two perspectives. First, we use CSD to retrieve captions. As clarified in **[Q2]**, it uses uncertainty.
> > > > >
> > > > > More importantly, we estimate the [Root] embedding using uncertainty. Previous approaches use the average text embedding or null text embedding as [Root] embedding. Instead of using average or null text embedding, we propose to use uncertainty-based [Root] embedding. As clarified in Section 4.4, we first retrieve the most similar caption of the given image (this is the common protocol for this task). Then, we search the most inclusive caption of the retrieved caption in the database using the inclusion measure that we proposed. Then, we use the most inclusive caption as the [Root] embedding, which is a more plausible “root” compared to the average or null embedding.
> > > > >
> > > > > We re-clarify this in Section B.3 of the revised paper.

---

> > > > > > ### Author Response · Authors · 2024-11-20
> > > > > > **Any follow-up question?**
> > > > > >
> > > > > > Dear Reviewer t6Gq,
> > > > > >
> > > > > > We sincerely appreciate your efforts and time for the community. As we approach the close of the author-reviewer discussion period in one week, we wonder whether the reviewer is satisfied with our response. It will be pleasurable if the reviewer can give us the reviewer's thoughts on the current revision to give us an extra valuable chance to improve our paper. We summarized our revision in the "Revision summary" comment.
> > > > > >
> > > > > > Again, we thank the reviewer's valuable commitment and their help to strengthen our submission. We will address all the raised concerns by reviewers if there remain any.

---

> > > > > > > ### Comment · Reviewer_t6Gq · 2024-11-21
> > > > > > > **Towards Positive**
> > > > > > >
> > > > > > > Thanks for the detailed response and further analysis. Most of my questions are resolved, and I have updated my score towards positive. I also appreciate the open-sourcing decision from the paper, which would be broadly beneficial to the community. -- Reviewer

---

> > > > > > > > ### Author Response · Authors · 2024-11-22
> > > > > > > >
> > > > > > > > Thank you for your update! Could you kindly share the reasons behind your decision to raise the score to 6 (marginally above the acceptance threshold) rather than 8 (accept)? If there are specific aspects we can improve, we would be happy to update our paper accordingly.

---

> > > > > > > > > ### Author Response · Authors · 2024-11-29
> > > > > > > > >
> > > > > > > > > Dear Reviewer t6Gq,
> > > > > > > > >
> > > > > > > > > We would like to note that the reviewer-author discussion period will be closed in three days (2nd Dec). We would like to ask the reviewer to be willing to update their score in a more positive direction from borderline (6) to accept (8). If the reviewer thinks that there still exists room for improvement, please let us know. We will do our best to improve our submission.

---

> > > > > > > > > > ### Author Response · Authors · 2024-12-03
> > > > > > > > > >
> > > > > > > > > > Dear Reviewer t6Gq,
> > > > > > > > > >
> > > > > > > > > > This is a gentle reminder that **only 6 hours remain** for reviewers to respond to comments. We are still awaiting your response and would greatly appreciate it if you could provide your feedback. Additionally, we kindly ask if the reviewer would consider updating your score based on our clarification.

---

### Official Review · Reviewer_N1uc · 2024-11-04

**Soundness:** 3
**Presentation:** 2
**Contribution:** 3
**Rating:** 6
**Confidence:** 5

**Summary:**

This paper introduces ProLIP, a fully probabilistic approach to language-image pre-training. The authors begin by noting that CLIP's loss function oversimplifies the complex relationships between real-world images and texts. In response, ProLIP proposes a novel loss function that better captures the distributional relationships between these modalities. Extensive experiments were conducted to demonstrate the effectiveness of the proposed method.

**Strengths:**

- This paper is well-motivated, starting from the many-to-many matching relationships within a batch of images and texts. It is also well-structured and easy to follow.
- The authors provided strong mathematical support for the proposed learning objective.
- Extensive experiments were conducted to demonstrate the effectiveness of the proposed method.
- The proposed method has been proven effective on datasets containing billions of image-text pairs.

**Weaknesses:**

- The proposed method was trained only on ViT-B and ViT-L vision encoders. Scaling up to ViT-H  and comparing with other methods are important to further demonstrate the scalability of ProLIP.
- As shown in Table 1, ProLIP introduces a more complex loss design and training process compared to CLIP, yet it achieves only a 0.9% improvement in ImageNet zero-shot classification and an average gain of 0.7%. Also in Table C.1, ProLIP with ViT-L also just exhibits slight improvement over CLIP. This modest improvement may indicate scalability challenges for the proposed method.

**Questions:**

Please refer to weaknesses.

---

> ### Author Response · Authors · 2024-11-18
>
> We thank the reviewer for their positive feedback and constructive comments. In the following comment and the revised paper, we address all the raised concerns by the reviewer. The overview of the revised paper is clarified in the common comment (“Revision summary”). If the reviewer still has any concerns, please let us know.
>
> **[W1] ViT-H**
>
> Thanks for your suggestion. We indeed aspire to train a ViT-H model. However, we first clarify that we only have a limited resource which is not feasible to train ViT-H even for deterministic models. For example, training a ViT-H/14 CLIP model with 34B seen samples (achieving 78.0% ImageNet zero-shot accuracy) task 279 hours with **824 A100s** [A], while we only have 32 GPUs. In theory, we need 7184 hours, almost 300 days for training this. Note that it is not for ProLIP, but for the baseline CLIP.
>
> - [A] Cherti, Mehdi, et al. "Reproducible scaling laws for contrastive language-image learning." Proceedings of the IEEE/CVF Conference on Computer Vision and Pattern Recognition. 2023.
>
> Even for ViT-L/14 12.8B seen samples, we need almost 30 days with our resources. It is not the problem of ProLIP, but the fundamental computational limitation of CLIP models. Overall, we do not have enough resources to train ViT-H.
>
> Furthermore, we remark that ImageNet zero-shot accuracy achieved by our ViT-L/14 (79.4%) is stronger than even the g/14 model with 34B seen samples in [A]:
>
> | H/14 (13B) | g/14 (13B) | G/14 (13B) | H/14 (34B) | g/14 (34B) | G/14 (34B) |
> |--|--|--|--|--|--|
> | 75.6 | 76.7 | 78.3 | 78.0 | 79.1 | 80.5 |
>
> Note that g/14 with 13B seen samples took 137 hours with 800 A100s, therefore, we can estimate the training time of g/14 with 34B seen samples might take 358 hours. On the other hand, our ViT-L/14 fine-tuning took approximately 3 days with 32 GPUs.
>
> We are now trying to fine-tune a pre-trained deterministic ViT-H/14 CLIP model (which even still takes weeks with our resources). However, we are not certain that it is possible to find a proper fine-tuning setting that is beyond our original contribution. We will inform the reviewer when we get a new result.

---

> > ### Author Response · Authors · 2024-11-18
> >
> > **[W2] Modest improvement**
> >
> > First, we want to emphasize that +0.9% ImageNet zero-shot accuracy is not a marginal improvement. As shown in [A], it is almost close to a gap when using a larger backbone (e.g., ViT-H/14 (75.6%) $\Rightarrow$ ViT-g/14 (76.7%) when the number of seen samples is 13B). However, we also want to clarify that our purpose is not to achieve a significantly stronger ImageNet zero-shot performance, but to achieve a better understanding of the input uncertainty (i.e., aleatoric uncertainty).
> >
> > Even if we consider the improvement to be marginal, we argue that the modest improvement against CLIP and the scalability challenge of PrVLM are not at the same level. As already clarified in Section A.3, our ablation study (Table C.3) empirically shows that the PCME++ loss fails to be converged when it is used in a stand-alone way without any deterministic loss (e.g., CLIP loss). On the other hand, PPCL loss converges well even without any additional loss function. Our first contribution is to enable a fully end-to-end scalable pre-training of PrVLM by proposing (1) new loss functions -- PPCL and inclusion loss, (2) new [UNC] token architecture, which is extremely efficient. Table C.3 and C.6 support that our new loss function and architecture outperform the previous PrVLM design choice (PCME++ loss + CLIP loss & multi-head uncertainty estimation module). Our main contribution is to train the first PrVLM (which has the advantage of correctly representing aleatoric uncertainty -- please see the newly added Section A.1 for more detailed discussion) comparable to the deterministic baselines (e.g., CLIP, SigLIP). Again, previous PrVLMs were not able to pre-train (i.e., they fine-tune the pre-trained CLIP models). PCME++ shows a small-scale pre-training result, but it still needs deterministic loss. Furthermore, as we mentioned before, even if we use additional CLIP loss, PPCL-only ProLIP performs better than PCME++ (Table C.3). It shows that ProLIP has a significant novel contribution to PrVLM.
> >
> > Furthermore, the proposed loss design and training process are not specifically complex than CLIP. In terms of the training process, we use the same training process to openclip, except for the “partial” inclusion loss. Similarly, if we only consider PPCL, the loss design is not specifically complex than the original contrastive loss (CLIP) or log sigmoid loss (SigLIP). As shown in Table C.3, solely using PPCL is sufficient for achieving performance; we use additional loss functions (inclusion loss, VIB loss) to ensure the learned embedding space has our desired properties (e.g., showing inclusive relationship between text and image / partial and whole instance, and preventing variance collapse). Our purpose is to train a fully end-to-end scalable pre-training of PrVLM, not outperform CLIP with a large gap.
> >
> > We also would like to clarify that, in Table C.1., ProLIP ViT-L is not a fair comparison with other backbones. It is a fine-tuned backbone from SigLIP ViT-L (ImageNet zero-shot 80%), not fully from scratch training. Even more, we would like to emphasize that its improvement is never modest against ViT-B/16 CLIP; The fine-tuned ProLIP ViT-L achieves 79.4% ImageNet zero-shot accuracy, while CLIP ViT-B/16 achieves 67.2% in our setting (1.28B seen samples), and 73.5% by openclip (13B seen samples). Even openclip ViT-L and ViT-H achieved 75.3% and 78.0%, respectively, while our ViT-L/14 shows 79.4%, outperforming expensive ViT-H training (Remark that openclip ViT-H took 279 hours with 824 A100s). We argue that the difference is not as modest as the reviewer’s comment. Again, we clarify that searching hyperparameters for fine-tuning is out of our main contribution, hence, we did not exhaustively search the fine-tuning hyperparameters and there will be a large room for performance improvements by hyperparameter search.

---

> > > ### Author Response · Authors · 2024-11-20
> > > **Any follow-up question?**
> > >
> > > Dear Reviewer N1uc,
> > >
> > > We sincerely appreciate your efforts and time for the community. As we approach the close of the author-reviewer discussion period in one week, we wonder whether the reviewer is satisfied with our response. It will be pleasurable if the reviewer can give us the reviewer's thoughts on the current revision to give us an extra valuable chance to improve our paper. We summarized our revision in the "Revision summary" comment.
> > >
> > > Again, we thank the reviewer's valuable commitment and their help to strengthen our submission. We will address all the raised concerns by reviewers if there remain any.

---

> > > > ### Author Response · Authors · 2024-11-26
> > > >
> > > > Dear Reviewer N1uc,
> > > >
> > > > We want to bring your attention to our submission. We tried to resolve your concerns via the comment and the revision. In the revised paper Table 1 and C.1, we report the fine-tuned ViT SO400M/14 (427M parameters -- similar to ViT-L/16 but it has much more parameters for vision encoder while having a relatively small text encoder) result which is more efficient than the ViT-H model (632M parameters). In the table, we can observe that SO400M/14 shows a stronger average performance than L/16 (SO400M: 66.6 vs. L/16: 65.9). Again, our goal is not to achieve a new state-of-the-art VLM, but we aim to learn a sufficiently strong novel PrVLM. We clarify that, for the sake of simplicity, we slightly modified the original architecture (excluding the attention pooling module), which does not guarantee the same performance as the original architecture. We clarified the details of fine-tuning in Section B.2.
> > > >
> > > > We would like to ask the reviewer to be willing to update their score in a more positive direction. If the reviewer thinks that there still exists room for improvement, please let us know. We will do our best to improve our submission.

---

> > > > > ### Comment · Reviewer_N1uc · 2024-11-27
> > > > > **Comments to Authors' Responses**
> > > > >
> > > > > Thank you for your comprehensive responses. After careful consideration, I will maintain my initial scores for the following reasons. While the responses address my concerns regarding the complexity of the proposed method, and I acknowledge the challenges of training ViT-H with limited resources and time, I believe it remains unfair to compare ProLIP with previous works like OpenCLIP or [A] Cherti, Mehdi, et al. This is because ProLIP is trained on the DataComp dataset, which is significantly better than previously available datasets. Furthermore, while I agree that achieving state-of-the-art results is not the primary goal of this paper, demonstrating the scalability of ProLIP under more parameter setups is crucial. I encourage the authors to include scalability experiments in future revisions.

---

> > > > > > ### Author Response · Authors · 2024-11-28
> > > > > >
> > > > > > Dear Reviewer N1uc,
> > > > > >
> > > > > > 1. Our work is invariant to openclip and Cherti, Mehdi, et al. We did not compare our work with them (their main contributions are mainly around the scalability beyond tens of billion seen samples and very large backbone, such as ViT-H or ViT-g) and we do not think that our results are directly comparable with them. Our main contribution is to enable from-scratch training of PrVLMs, which was not possible by the previous PrVLMs (in terms of from-scratch training) and previous VLM works, including openclip and Cherti, Mehdi, et al (they are not probabilistic representations).
> > > > > > 2. We already provided the results that show by increasing the parameter size ViT-B/16 (46.44GFlops) $\rightarrow$ ViT-L/16 (136.41GFlops, fine-tuned) $\rightarrow$ ViT-SO400M/14 (233.54GFlops, fine-tuned), the overall performance is improved (63.3 $\rightarrow$ 65.9 $\rightarrow$ 66.6). ViT-H has 381.68GFlops, which is not easy to fit with our limited resources (model profiles are from the official openclip model profile document: https://github.com/mlfoundations/open_clip/blob/main/docs/model_profile.csv). We are now trying to fine-tune ViT-H, and if we have any additional results, we will inform you.
> > > > > > 3. Again, scaling up beyond ViT-H is not our main contribution, and we do think it is not fair to compare our work with big research groups with hundreds of GPUs. Our work is even larger scale than other vision language representation works. For example, we bring two recent ICML papers [B,C] as examples (there are indeed more than two papers, but we take only two in this comment). Meru [B], a hyperbolic embedding-based CLIP model, only showed their results on a limited number of seen samples (approximately 245M seen samples). Their ImageNet zero-shot accuracy is only 34.3%, while ProLIP achieves more than a double score of Meru (74.6% for ViT-B/16). Similarly, OT-CLIP [C] was only trained with approximately 90M seen samples, and achieved only 16~17% ImageNet zero-shot accuracies. Although they didn't show ViT-H results or billion-scale seen samples as ours, their works still bring value to the community by showing that a different paradigm (hyperbolic embedding) or a different optimization technique (OT) can be helpful to CLIP training. We think that our work is also a similar to [B] and [C], rather than showing ViT-H which is not accessible by most researchers (again, it takes 279 hours with 824 A100s [A]).
> > > > > >
> > > > > > - [B] Desai, Karan, et al. "Hyperbolic image-text representations." ICML 2023.
> > > > > > - [C] Shi, Liangliang, Jack Fan, and Junchi Yan. "OT-CLIP: Understanding and Generalizing CLIP via Optimal Transport." ICML 2024
> > > > > >
> > > > > > From these three reasons (**1** our work is not comparable to openclip and [A], **2** we already show that more params lead to better performance, **3** there are already many research papers not showing ViT-H appeared in top-tier ML conferences [B, C and more]), we would like to ask the reviewer to adjust their score if the reviewer's only concern is the lack of ViT-H experiment.

---

> > > > > > > ### Author Response · Authors · 2024-11-29
> > > > > > >
> > > > > > > Dear Reviewer N1uc,
> > > > > > >
> > > > > > > We are now running a 1.28B fine-tuning experiment for a ViT-H model. We expect that it will end 6 days later, i.e., after the discussion period. When we check the intermediate result, we expect that the new result will be sufficiently strong (now it shows 74.3 ImageNet top-1 accuracy, while at the same iteration, ViT-SO400M shows 72.5).
> > > > > > >
> > > > > > > Again, we would like to ask the reviewer to be willing to update their score in a more positive direction from borderline (6) to accept (8) considering our latest comment and this new result. If the reviewer thinks that there still exists room for improvement, please let us know. We will do our best to improve our submission.

---

> > > > > > > > ### Author Response · Authors · 2024-12-02
> > > > > > > >
> > > > > > > > Dear Reviewer N1uc,
> > > > > > > >
> > > > > > > > As we approach the end of the author-reviewer discussion period **today**, we kindly request the reviewer to consider updating their score from borderline (6) to a more positive direction, such as accept (8). In our latest comment, we have provided clarification regarding the ViT-H experiments, addressing the concerns raised.

---

> ### Author Response · Authors · 2024-12-03
>
> Dear Reviewer N1uc,
>
> This is a gentle reminder that **only 6 hours remain** for reviewers to respond to comments. We are still awaiting your response and would greatly appreciate it if you could provide your feedback. Additionally, we kindly ask if the reviewer would consider updating your score based on our clarification.

---

### Official Review · Reviewer_1UQo · 2024-11-07

**Soundness:** 3
**Presentation:** 2
**Contribution:** 2
**Rating:** 8
**Confidence:** 3

**Summary:**

The paper proposes a novel approach for vision-language models (VLMs) that leverages probabilistic embeddings to capture better the inherent many-to-many relationships between images and text descriptions.

In general, the contribution includes
* Novel Probabilistic Embeddings: The paper introduces a fully probabilistic VLM that assigns each image-text pair a probability distribution rather than a deterministic embedding, addressing the inherent ambiguity in real-world image-text matching. This approach shows promise in capturing a richer semantic alignment by allowing multiple captions to represent an image and vice versa.
* Efficient Uncertainty Estimation: Unlike prior probabilistic VLMs, ProLIP efficiently estimates uncertainty by introducing an “uncertainty token” ([UNC]). This no-parameter simplification is computationally efficient and effectively captures the variance within embeddings, enhancing interpretability.
* Inclusion Loss: This paper introduces a novel inclusion loss, which enforces distributional inclusion relationships, and strengthens ProLIP’s interpretability by aligning learned uncertainties with human expectations (e.g., general captions should cover more specific ones).
* Performance on Zero-Shot Classification: ProLIP demonstrates strong results in zero-shot classification tasks, such as outperforming a CLIP baseline with similar architecture. The improvement in ImageNet classification accuracy (from 74.6% to 75.8%) with the use of uncertainty highlights the approach’s practical advantages.

**Strengths:**

1. Originality
* The paper introduces a novel Probabilistic Vision-Language Model (PrVLM), named ProLIP, which represents a shift from deterministic vision-language models to probabilistic embeddings. This approach innovatively captures the natural ambiguity in image-text relationships, where multiple captions may describe a single image, and a single caption may match multiple images.
* The concept of an uncertainty token ([UNC]) is a creative contribution, allowing the model to quantify uncertainty in a computationally efficient way, without additional parameters. This method differentiates ProLIP from previous probabilistic VLMs, which typically rely on additional modules to estimate uncertainty.
* The inclusion loss is another novel component of the paper. By enforcing a distributional inclusion relationship, the loss function allows the model to create more interpretable representations that respect hierarchical relationships within image-text pairs. This focus on hierarchical embedding structures is an innovative step for VLMs.

2. Quality
* The authors have conducted a thorough and robust set of experiments to validate ProLIP's effectiveness. These include zero-shot classification tasks across multiple datasets, evaluations of ProLIP’s uncertainty estimation, and comparisons with both deterministic (e.g., CLIP) and probabilistic baselines.
* The paper is methodologically sound and clearly specifies the assumptions, architectures, and loss functions. The experiments are carefully controlled, with baseline comparisons and ablation studies that effectively illustrate ProLIP’s strengths in various contexts.

3. Clarity
* The paper is written in a clear and accessible manner. Technical terms and novel concepts (e.g., inclusion loss, uncertainty token) are well-defined, with explanations that make the content readable.
* Figures, such as the visualizations, are well-crafted and effectively illustrate key points.

4. Significance
* The introduction of probabilistic embeddings and inclusion loss contributes to the broader AI field’s understanding of how uncertainty modeling can enhance interpretability and generalization in multi-modal tasks. This may inspire further work in this direction.

**Weaknesses:**

1. Interpretability and Visualization:
*  Although the inclusion loss aligns with human intuitions, the visualization methods could be improved. For example, it would be interesting to see more direct comparisons with non-probabilistic models in terms of how well ProLIP aligns visual and textual hierarchies in practice.

2. Limited Exploration of Task-Specific Uncertainties:
*  The paper briefly touches on using uncertainty for image-to-text retrieval but could expand by exploring how uncertainty contributes to other downstream tasks. For instance, analyzing ProLIP’s impact on retrieval diversity or specificity could further highlight its advantages.

3. Human evaluation
*  While the experiments indicate that ProLIP’s learned uncertainties align with human intuitions, a concrete user study or human evaluation would reinforce these claims.

**Questions:**

1. Could you clarify the design choice of the probabilistic pairwise contrastive loss (PPCL) over simpler alternatives, like KL divergence-based objectives? Have you tested how PPCL compares with other probabilistic loss functions in terms of training stability and final performance?

2. The inclusion loss is an interesting addition. Could you explain in more detail the intuition behind using this specific formulation, and how it compares with other metrics like Wasserstein distance?

3. The paper shows that ProLIP’s uncertainty estimates align well with general human expectations, such as more uncertainty for shorter text descriptions. Did you conduct any quantitative or qualitative user studies to validate this alignment? If not, how would you suggest evaluating the alignment between model uncertainty and human perception in future work?

4. You demonstrated ProLIP’s zero-shot classification ability, but could it also support a few-shot learning approach? What further modifications would be needed to explore few-shot learning capabilities and potentially improve its applicability in low-resource settings?

---

> ### Author Response · Authors · 2024-11-18
>
> We thank the reviewer for their positive feedback and constructive comments. In the following comment and the revised paper, we address all the raised concerns by the reviewer. The overview of the revised paper is clarified in the common comment (“Revision summary”). If the reviewer still has any concerns, please let us know.
>
> **[W1] More visualization**
>
> Thanks for the suggestion. We added a new visualization result of image traversal as the reviewer suggested. As shown in Figure B.1, the [ROOT] embedding of CLIP is always closest to the “umbrella”, which makes the image traversal by CLIP inaccurate. On the other hand, the [ROOT] embedding of ProLIP correctly estimates the true hierarchy of the given image query.
>
> **[W2] How uncertainty contributes to other downstream tasks**
>
> We also agree with that it would be better to show more applications of the learned uncertainty. Here, we show 3 applications.
>
> **Dataset filtering**
>
> Below, we show the DataComp CLIP filtering small track (filtering 12.8B noisy web-crawled image-text pairs) by using our method and baselines provided by DataComp [A]
>
> - [A] Gadre, Samir Yitzhak, et al. "Datacomp: In search of the next generation of multimodal datasets." NeurIPS 2024.
>
> |  | Dataset size | ImageNet | ImageNet dist. shifts | VTAB | Retrieval | Average |
> |---|---|---|---|---|---|---|
> | No filtering | 12.8M | 0.025 | 0.033 | 0.145 | 0.114 | 0.132 |
> | Random subset (25%) | 3.2M | 0.022 | 0.032 | 0.130 | 0.099 | 0.126 |
> | LAION-2B filtering | 1.3M | 0.031 | 0.040 | 0.136 | 0.092 | 0.133 |
> | English (fasttext), caption length, and image size | 3M | 0.038 | 0.043 | 0.150 | 0.118 | 0.142 |
> | Image-based & CLIP score (L/14 30%) | 1.4M | 0.039 | 0.045 | 0.162 | 0.094 | 0.144 |
> | CLIP L14 (20%) | 2.6M | 0.042 | 0.051 | 0.165 | 0.100 | 0.151 |
> | ProLIP B16 (20%) | 2.3M | 0.042 | 0.047 | 0.167 | 0.117 | **0.154** |
>
> Here, we can observe that by using ProLIP uncertainty-aware features, we can achieve a better filtering performance compared to the other baselines. However, we would like to note that ProLIP is not specifically designed for dataset filtering; proposing a new dataset filtering method using ProLIP will be an interesting future work, but not the scope of the current paper.

---

> > ### Author Response · Authors · 2024-11-18
> >
> > **Understanding image dataset**
> >
> > First, we emphasize that ProLIP’s image uncertainty is not the same as the “image uncertainty” of classification tasks. In classification tasks, an image has a high uncertainty if it can be matched to “multiple classes”, while the ProLIP case assigns a high uncertainty if an image can be described in “multiple different and various captions”. However, ProLIP has a different mechanism with classification uncertainty.
> >
> > An ideal PrVLM should capture three potential input uncertainties: (1) uncertainty from the text modality, (2) uncertainty from the image modality, and (3) uncertainty from the text-image cross-modality. Uni-modal uncertainty is straightforward; if an input has more detailed information (e.g., describing more detailed information in text, or capturing a very detailed and complex scene by photography), then it will have smaller uncertainty, otherwise, it will have larger uncertainty (e.g., providing very high-level caption, such as “person”, or only a part object with white background is taken by picture).
> >
> > The cross-modal uncertainty should capture “how many possible instances can be matched to this input?”. We can think this in two different viewpoints: text-to-image and image-to-text. The text-to-image relationship is straightforward. If we have a caption “photo”, then it will be matched to all photographs, and if we have a caption with a very detailed description (e.g., the full description of the hotel room), then it will be only matched to specific images. Image-to-text relationships are often determined by the dataset. For example, if we have a caption dataset exactly describing which objects are in the image, then we can think that an image with fewer objects will have larger uncertainty. However, if we consider a caption dataset where an image has multiple captions each caption focuses on completely different objects, then an image with more objects will have larger uncertainty. In other words, unlike text uncertainty originating from a text-to-image relationship, image uncertainty originating from an image-to-text relationship is highly affected by the property of the training dataset.
> >
> > In practice, because our datasets have a mixed property and their captions are somewhat noisy, our image uncertainty will have a mixed property, namely, unlike text uncertainty, there could be no strong relationship between the absolute uncertainty value and the number of objects (or complexity of images). However, empirically, we have more captions that exactly describe the scene (because of the dataset filtering strategy), therefore, a more simple image tends to have larger uncertainty and a more complex image tends to have smaller uncertainty.
> >
> > For example, assume we have an image with a white background and a very clear overall object shape. In terms of classification, it will have low uncertainty because there is no confounder to the classification. However, ProLIP will assign a high uncertainty for this case.
> >
> > From this, we can divide the image classification tasks into two different scenarios: (1) When all images have homogeneous backgrounds and only the quality of the image determines the classification performance, (2) When images are natural images and the task is inherently multi-object classification, but the labels are single-labeled (e.g., ImageNet [B, C])
> >
> > - [B] Beyer, Lucas, et al. "Are we done with imagenet?." arXiv preprint arXiv:2006.07159 (2020).
> > - [C] Yun, Sangdoo, et al. "Re-labeling imagenet: from single to multi-labels, from global to localized labels." CVPR 2021.
> >
> > As the first example, we choose MNIST. Here, we observe that the learned image uncertainty and the MNIST accuracy show a strong negative correlation, (-0.98), namely, if an image is more uncertain then ProLIP tends to estimate a wrong label. It aligns with our intuition.
> >
> > As the second example, we choose ImageNet-1k, which shows a strong positive correlation (+0.98), i.e., a certain image tends to be wrongly classified by ProLIP. This could be counterintuitive in terms of “classification task”, but actually it is a correctly estimated value. For example, ImageNet contains various image distributions. Some images are thumbnail images with white backgrounds (high uncertainty) and some images are in-the-wild images with complex backgrounds and multiple objects (low uncertainty). In this case, a certain image (more complex image) will be a more “difficult” image.
> >
> > Overall, ProLIP’s image uncertainty tendency can be used for understanding the properties of the given dataset. Converting ProLIP’s image uncertainty to image classification uncertainty would be an interesting topic, but we remain this for future work.

---

> > > ### Author Response · Authors · 2024-11-18
> > >
> > > **Uncertainty by image manipulation**
> > >
> > > We additionally show the relationship between image manipulation and uncertainty. We evaluate ImageNet 1k zero-shot accuracy by applying a center occlusion. We applied 0% to 10% occlusion ratio, and got the following results:
> > >
> > > | Occlusion ratio | ImageNet ZS | avg($\sigma_v$) |
> > > |---|---|---|
> > > | 0% | 74.6 | 0.0148 |
> > > | 2.5% | 74.1 | 0.0149 |
> > > | 5% | 73.8 | 0.0152 |
> > > | 7.5% | 73.5 | 0.0153 |
> > > | 10% | 73.2 | 0.0153 |
> > >
> > > We also tested optimized noise by the PGD attack [D] with sampled ImageNet (1000 images):
> > >
> > > | Attack | ImageNet ZS (1000 images) | avg($\sigma_v$) |
> > > |---|---|---|
> > > | - | 72.9 | 0.0147 |
> > > | PGD(iter=1) | 20.0 | 0.0149 |
> > > | PGD(iter=5) | 3.8 | 0.0167 |
> > > | PGD(iter=10) | 2.6 | 0.0175 |
> > > | PGD(iter=40) | 2.5 | 0.0190 |
> > >
> > > - [D] Mądry, Aleksander, et al. "Towards deep learning models resistant to adversarial attacks." ICLR 2018
> > >
> > > Here, we observe that the image uncertainty is increased by more severe manipulation. Note that as we discussed in the previous application (**Understanding image dataset**), converting ProLIP’s image uncertainty to image qualification would be an interesting topic, but we remain this for future work.
> > >
> > > **[W3/Q3] Human evaluation**
> > >
> > > Thanks for your great idea. We strongly agree with this idea and actually have a similar result to the reviewer’s idea. The HierarCaps dataset [E] is a human-validated caption dataset where each image has four levels of captions. For example, “water sports” $\Rightarrow$ “kite surfing” $\Rightarrow$ “kite surfer on top of the board”  $\Rightarrow$ “kite surfer in the air on top of a red board”. These captions are human-validated, namely, we already have the “hierarchical perception” of humans.
> > >
> > > [E] Alper, Morris, and Hadar Averbuch-Elor. "Emergent visual-semantic hierarchies in image-text representations." ECCV 2024.
> > >
> > > In Fig7 of the main paper, we already showed that a more general caption (e.g., “water sports” in the previous example) has a larger uncertainty value and a more specific caption (e.g., “kite surfer in the air on top of a red board”) has a smaller uncertainty value. It supports that human understanding of hierarchical information is well aligned with the learned uncertainty.
> > >
> > > We can find a similar observation in the image modality. In Fig8, we showed that the masked image only containing sub-objects of the original image tends to “include” the original image. Namely, if we mask an image and retain partial information, the new embedding will include the original embedding. We do not have a concrete human study that supports “humans think a partial image is more uncertain than the whole image”, but if we accept this argument, Fig8 supports this claim.
> > >
> > > We clarify this in Section B.3 of the revised paper. If the reviewer thinks that the current result and revision are insufficient, we will conduct a human study to support the claim for the image-level uncertainty.
> > >
> > > **[Q1] PPCL design choice**
> > >
> > > PPCL design choice can be divided into two parts. The first one is to use the LogSigmoid operation, and the other is to use closed-form sampled distance (CSD) as the probabilistic distance.
> > >
> > > We ablated PPCL with binary cross entropy (BCE) loss used for PCME++ (Chun 2024). As shown in Table C.3 of the paper, a model will not be converged if we solely use PCME++ loss. We need additional deterministic loss, such as CLIP loss or SigLIP loss for a stable convergence of PCME++ loss. Even if we use additional deterministic loss, we can find that it performs worse than a model trained solely with ProLIP’s PPCL loss.
> > >
> > > The second ablation (the choice of CSD) is already done by PCME++ (Chun 2024). As shown in Table 4 of Chun 2024, CSD is the best method for probabilistic representation learning, while KL divergence, JS divergence, Wasserstein 2-distance (WD) even did not converge. This is because our desired probabilistic embeddings should have the following properties. (1) if an image-text pair is “certain”, each instance should have a small variance, (2) if an image-text pair is “uncertain”, each instance should have a large variance, and (3) the used probabilistic distance should satisfy proper properties of “distance”. As shown in Chun 2024, CSD satisfies all three criteria. KL, JS, WD, or other probabilistic “distance” do not satisfy the criterion because their purpose is to compare whether two distributions are the same or not. Namely, even if two instances have a “certain” correspondence if two instances have very similar $\mu$ and $\Sigma$, these distances cannot minimize their $\Sigma$ because the distance between two instances will be almost zero regardless of the intensity of $\Sigma$.

---

> > > > ### Author Response · Authors · 2024-11-18
> > > >
> > > > **[Q2] Inclusion loss details**
> > > >
> > > > Section 3.3 and Figure 3 explain the intuition behind the inclusion loss. In terms of the formula, our purpose is to design a measure $d(Z_1, Z_2)$ that has a high value if $Z_1$ is “included” in $Z_2$ and a low value otherwise. How can we define $Z_1$ is “included” in $Z_2$? We define this by measuring the “overlapped area” between $Z_1$ and $Z_2$, but emphasizing the area with high $p_1$. As shown in Fig3, the squared pdf of $Z_1$ emphasizes the area where $Z_1$ has high probability; by computing $\int p_1^2 p_2$, we can determine whether $Z_1$ is included in $Z_2$ or not.
> > > >
> > > > We also already compared our inclusion loss and other metrics like Bhattacharyya distance (BD), Wasserstein distance (WD) and KL divergence in Section 3.3 and Figure 3. The most significant difference between our inclusion measure and BD, WD is asymmetricity. BD and WD are symmetric, namely, $WD(Z_1, Z_2) = WD(Z_2, Z_1)$, which cannot be used for measuring “inclusion”. On the other hand our inclusion measure has asymmetricity, $inc(Z_1, Z_2) > inc(Z_2, Z_1)$ if $Z_1$ is included in $Z_2$. KL is also asymmetric, but it is for measuring the equivalence between two distances, and cannot represent “inclusiveness” as shown in Figure 3. We additionally compare our inclusion loss with the difference of KL and reverse KL in Appendix A.6. We can observe that this measure is also not proper to estimate the amount of “inclusion”.
> > > >
> > > > **[Q4] Few-shot learning**
> > > >
> > > > Thanks for the suggestion. In fact, our prompt tuning with uncertainty estimates (Section 3.4 and Table 3) already includes the concept of few-shot learning. Our method, named Bayesian Prompt Re-Weighting (BPRW), is a fully Bayesian approach to estimating the best weight for each prompt; by treating given M image embeddings as “observations”, we seek the best weight of each text prompt Gaussian distribution by maximizing posterior. For example, assume that we have a class label “Tench” and 80 prompts (e.g., a photo of Tench, a good photo of Tench, …). Our purpose is to find the weight of each text prompt to aggregate 80 prompts into one text embedding (i.e., $\mu = \sum_{i=1}^{80} w_i \mu_i$).
> > > >
> > > > This task can be either zero-shot or few-shot by the way to select K image embeddings. If we set a “few-shot” scenario, then we choose M image embeddings from ground truth “Tench” images, it will be a few-shot. If we do not use the image label, but sample image embeddings in a different way, it will be a “zero-shot” scenario. As shown in Table 3, the few-shot learning scenario significantly improves the classification performance compared to the naive average of 80 prompts (74.6 $\rightarrow$ 75.8).

---

> > > > > ### Author Response · Authors · 2024-11-20
> > > > > **Any follow-up question?**
> > > > >
> > > > > Dear Reviewer 1UQo,
> > > > >
> > > > > We sincerely appreciate your efforts and time for the community. As we approach the close of the author-reviewer discussion period in one week, we wonder whether the reviewer is satisfied with our response. It will be pleasurable if the reviewer can give us the reviewer's thoughts on the current revision to give us an extra valuable chance to improve our paper. We summarized our revision in the "Revision summary" comment.
> > > > >
> > > > > Again, we thank the reviewer's valuable commitment and their help to strengthen our submission. We will address all the raised concerns by reviewers if there remain any.

---

> > > > > > ### Author Response · Authors · 2024-11-26
> > > > > >
> > > > > > Dear Reviewer 1UQo,
> > > > > >
> > > > > > We want to bring your attention to our submission. We tried to resolve your concerns via the comment and the revision. In the revised paper Section C.6 (page 27), we clarify how the experiments on HierarCaps can be used as a proxy of human study. We provide additional experiments in C.6 accordingly.
> > > > > >
> > > > > > We also added a new visualization result in Figure B.1 following the reviewer's suggestion. Section C.8 shows that our uncertainty is also beneficial to other downstream tasks, following the reviewer's comment.
> > > > > >
> > > > > > We would like to ask the reviewer to be willing to update their score in a more positive direction. If the reviewer thinks that there still exists room for improvement, please let us know. We will do our best to improve our submission.

---

> > > > > > > ### Author Response · Authors · 2024-11-29
> > > > > > >
> > > > > > > Dear Reviewer 1UQo,
> > > > > > >
> > > > > > > We would like to note that the reviewer-author discussion period will be closed in three days (2nd Dec). As mentioned in our previous comments, we tried to resolve all your concerns and suggestions in our response and revision.
> > > > > > >
> > > > > > > Again, we would like to ask the reviewer to be willing to update their score in a more positive direction from borderline (6) to accept (8). If the reviewer thinks that there still exists room for improvement, please let us know. We will do our best to improve our submission.

---

> > > > > > > > ### Author Response · Authors · 2024-12-02
> > > > > > > >
> > > > > > > > Dear Reviewer 1UQo,
> > > > > > > >
> > > > > > > > As we approach the end of the author-reviewer discussion period **today**, we kindly request the reviewer to consider updating their score from borderline (6) to a more positive direction, such as accept (8). If the reviewer thinks that there still exists room for improvement, please let us know. We will do our best to improve our submission.

---

> > > > > > > > > ### Author Response · Authors · 2024-12-03
> > > > > > > > >
> > > > > > > > > Dear Reviewer 1UQo,
> > > > > > > > >
> > > > > > > > > This is a gentle reminder that **only 6 hours remain** for reviewers to respond to comments. We are still awaiting your response and would greatly appreciate it if you could provide your feedback. Additionally, we kindly ask if the reviewer would consider updating your score based on our revision and rebuttal comment.

---

> > > > > > > > > > ### Comment · Reviewer_1UQo · 2024-12-03
> > > > > > > > > >
> > > > > > > > > > Thanks a lot for your detailed response. My questions and concerns have been well addressed.  I have increased my score.

---

> > > > > > > > > > > ### Author Response · Authors · 2024-12-03
> > > > > > > > > > >
> > > > > > > > > > > Thanks!

---

### Author Response · Authors · 2024-11-18
**Revision summary**

We truly appreciate all the reviewers and the chairs for their commitment to the community and thoughtful reviews. We are highly encouraged that the reviewers found that the proposed ProLIP is novel (Reviewer 1UQo, t6Gq, RZmV, 8Wie), the [UNC] architecture is creative (Reviewer 1UQo, RZmV), the inclusion loss and PPCL are novel and well supported (Reviewer 1UQo, N1uc, t6Gq), our manuscript is clear, well-written and well-motivated (Reviewer 1UQo, N1uc, 8Wie) and the conducted experiments are extensive and interesting (Reviewer 1UQo, N1uc, t6Gq, RZmV). We luckily have a chance to make our work stronger, with constructive and valuable comments from the reviewers. We have addressed all the raised concerns by the reviewers in the revised paper. The revised contents are highlighted in blue (if we modify the existing text) or red (if we add new text).

Here, we would like to clarify the changes since the last submission, and how they resolve the concerns raised by the reviewers.

First, we would like to clarify that while we addressed the concern raised by Reviewer RZmV, we found that our original experiments on 1.28B seen samples had a fair comparison issue; we made a mistake on data augmentation setting (we specified that we use 0.8-1.0 crop size, but we found that CLIP and ProLIP with 1.28B seen samples were trained with different crop sizes, e.g., 0.4-1.0 not 0.8-1.0). For a fair comparison, we re-evaluated ProLIP, CLIP and SigLIP on 1.28B seen samples and here is the result (also see revised paper Table 1):

| Method | # seen samples | IN | IN dist. | VTAB | Retrieval | Average |
|--------|-------|------|------|------|------|------|
| CLIP   | 1.28B | 67.2 | 55.1 | 56.9 | 53.4 | 57.1 |
| SigLIP | 1.28B | 67.4 | 55.4 | 55.7 | 53.4 | 56.7 |
| ProLIP | 1.28B | 67.8 | 55.3 | 58.5 | 53.0 | 57.9 |

Second, we found that we used the incorrect notation for “inclusion”. It should be $\subset$, rather than $\in$. We fixed all the notations in the revision.

Finally, we named the proposed prompt re-weighting method “Bayesian Prompt Re-Weighting (BPRW)” to make our contribution clearer.

To address the reviewers’ concerns, our revision also includes:

- We slightly revise the introduction and the related work to emphasize our contribution and avoid misunderstanding (Reviewer t6Gq [W2] [Q8])
- We add additional discussion of the desired probabilistic embedding space in Appendix A.1. This section addressed concerns by Reviewer N1uc [W2], Reviewer t6Gq [W2] [Q8], Reviewer RZmV [Q2] and Reviewer 8Wie [W1].
- We clarify the questions by Reviewer t6Gq [Q1] in Section A.3 (re-parameterization and l2 normalization) and Reviewer t6Gq [Q2] in Section B.2 (classification with variance)
- Section A.3 also clarifies the novelty of ProLIP compared to PCME++ to resolve the concern by Reviewer 8Wie [W2]
- We add more discussions and formulations for our loss function in Appendix A.6 and A.7 (Reviewer 1UQo [Q2], Reviewer t6Gq [Q3] [Q6])
- We add more details of image traversal and more visualization in Appendix B.3 (Reviewer 1UQo [W1] [W3] [Q3], Reviewer t6Gq [Q7])
- We add an architecture efficiency comparison experiment in Section C.3 and Table C.6 to address Reviewer 8Wie [W2] [Q1].
- Section C.7 contains more applications of uncertainty (Reviewer 1UQo [W2] and Reviewer t6Gq [W3])
- We add more discussion and limitation related to ProLIP in Appendix D (Reviewer t6Gq [W1], Reviewer 8Wie [Q2])

---

### Author Response · Authors · 2024-12-01
**Reminder**

Dear reviewers,

Thank you for your constructive and positive feedback! We are encouraged by the positive consensus reached among the reviewers, with all scores indicating "6: marginally above the acceptance threshold".

As the discussion phase will be closed shortly (within a day), we would like to kindly remind you to acknowledge our latest responses:
- We are still awaiting Reviewer 1UQo's response. We have addressed all the concerns raised by the reviewer and updated the paper accordingly. Following the reviewer's suggestion, we add a new visualization (Figure B.1), a proxy of human study (Section C.6), and more downstream tasks of uncertainty (Section C.8)
- We clarified to Reviewer N1uc that (1) our work is not directly comparable to openclip or Cherti, Mehdi, et al (2) We have already provided results demonstrating performance improvements as parameter size increases (ViT-B/16 $\rightarrow$ ViT-L/16 $\rightarrow$ ViT-SO400M/14) (3) ViT-H experiments have not been conducted by prior vision-language representation works. In contrast, our approach operates at an even larger scale compared to other VLM works published in top-tier ML conferences, and (4) we are now running ViT-H fine-tuning, which is currently 60% complete and shows promising results (77.4% IN top-1, while SO400M at the same iteration achieved 77.3%). Note that we chose SO400M rather than ViT-H to fit the original author-reviewer discussion deadline which was planned to be closed last week.
- We are also waiting for the responses from Reviewer t6Gq, RZmV, and 8Wie. We kindly requested insights into why your scores are 6 (marginally above acceptance) rather than 8 (accept). If there are specific areas for improvement, we are eager to address them to strengthen our submission further. We would like to ask the reviewer to be willing to update their score in a more positive direction from marginally above the acceptance threshold (6) to accept (8).

Thank you once again for your valuable contributions to the community. We look forward to your positive feedback and responses.

Authors

---

### Meta-Review · Area_Chair_ESuS · 2024-12-20

**Metareview:**

The paper introduces Probabilistic Language-Image Pre-training (ProLIP), an approach to vision-language models that integrates probabilistic modelling to capture the inherent uncertainty in image-text relationships. ProLIP departs from conventional deterministic embeddings by mapping inputs as random variables, estimating uncertainty using an "uncertainty token," and enforcing an inclusion loss to maintain distributional relationships between image-text pairs and between original and masked inputs. Lastly, it shows two applications of image traversals (find a more concrete caption for the original caption iteratively) and prompt enhancement for the zero-shot image classification task. Extensive experiments were conducted to demonstrate the effectiveness of the proposed method.

Strengths:
+ This paper is well-motivated, starting from the many-to-many matching relationships within a batch of images and texts. It is also well-structured and easy to follow.
+ The paper contains a detailed analysis of the sources and levels of uncertainty. Meanwhile, the authors provided strong mathematical support for the proposed learning objective.
+ Extensive experiments were conducted to demonstrate the effectiveness of the proposed method.
+ The proposed method has been proven effective on datasets containing billions of image-text pairs.

Weaknesses:
+ More experimental results on large model sizes are expected.
+ The improvements over the baselines are limited.
+ ProLIP requires the use of CSD to perform zero-shot prediction, which increases its complexity and makes it less straightforward than CLIP.
+ Discussions about the limitations are needed.

**Additional Comments On Reviewer Discussion:**

After the rebuttal, most of the concerns have been addressed, and all reviewers have raised their ratings to positive ratings.

---

### Decision · Program_Chairs · 2025-01-22

Accept (Poster)